# FlexBCQ: Flexible Binary-coding Quantization for Large Language Models

## Abstract

How can we compress large language models without compromising accuracy? Quantization, which reduces the number of bits for representing weights, is an essential technique to utilize large language models (LLMs) in real-world applications. Specifically, binary-coding quantization (BCQ) is a promising approach since it has extensive representation space, which encompasses the representation space of uniform quantization (UQ), and fast inference speed. However, because of the lack of accurate optimization techniques, BCQ shows inferior performance compared to UQ algorithms, failing to leverage their powerful expressive power. In this paper, we propose FlexBCQ (**Flex**ible **B**inary-**C**oding **Q**uantization), an accurate optimization algorithm for BCQ. We leverage the sophisticated optimization techniques of UQ by decomposing the quantization process of BCQ into the composition of a UQ and an inner BCQ. As a result, we take advantage of both the sophisticated optimizing techniques of UQ, specifically the flexible mapping technique, and the powerful expressive capability of BCQ. Through extensive experiments, we find that FlexBCQ provides 3.24%p higher accuracy than existing UQ and BCQ algorithms on MMLU 5-shot benchmark when quantizing a Llama-3 70B model into 3 bits.

## 1 Introduction

How can we compress large language models without compromising accuracy? Reducing the size of large language models (LLMs) (Brown et al., 2020; Touvron et al., 2023; Dubey et al., 2024) is crucial for deploying LLMs in real-world applications, as their gigantic size makes deployment challenging. Quantization (Xu et al., 2018; Kwon et al., 2022; Dettmers et al., 2022; Xiao et al., 2023; Lee et al., 2023; Lin et al., 2024) is a technique used to compress LLMs by reducing the number of bits needed to represent their weights. It reduces the bit count by representing the model's weights as a smaller set of values, namely, quantization levels. For example, 3-bit quantization uses $2^3$ distinct values to represent all the weights in the model. It is essential to use a quantization scheme that aligns well with the distribution of the model's weights, such as uniform quantization (UQ) (Lee et al., 2023; Lin et al., 2024; Liu et al., 2024) and binary-coding quantization (BCQ) (Xu et al., 2018; Kwon et al., 2022), to maximize the accuracy of a quantized model.

Uniform quantization (UQ) is a scheme that has evenly-spaced quantization levels. Figure 1(a) illustrates the quantization process of Round-to-Nearest (RTN) which maps a weight $w_0$ to the nearest quantization level $q_3$. RTN provides an optimal mapping for approximating weight itself, but if we consider the distribution of inputs, other mapping strategies encompassing farther quantization levels exhibit better accuracy (Nagel et al., 2020; Lee et al., 2023). FlexRound (b) achieves outstanding performance through its flexible mapping which allows weights to explore diverse quantization levels flexibly and map the weights to the quantization levels that maximize the accuracy of quantized models. On the other hand, binary-coding quantization (BCQ) is a scheme that has non-uniform quantization levels. Figure 1(c) illustrates the quantization process of Alternating update approach (Xu et al., 2018), a representative BCQ algorithm. As shown in the figure, BCQ adapts its non-uniform quantization levels close to the weight and reduces quantization errors which represents the distance between the weight and the mapped quantization level. Additionally, a recent study (Park et al., 2024) has introduced a fast inference kernel that supports BCQ and UQ at the same speed, providing faster inference speed than conventional UQ kernels (Frantar et al., 2023; Lin et al., 2024). This advancement positions BCQ as a promising quantization scheme, excelling

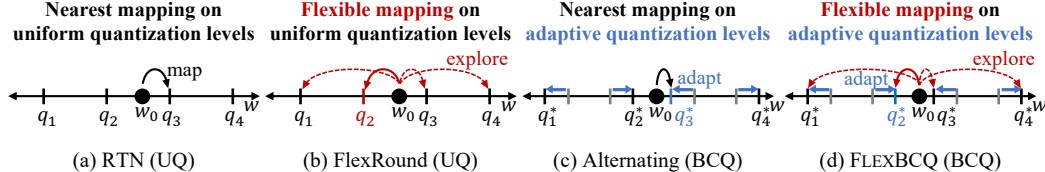

Figure 1: Comparison of quantization processes of RTN, FlexRound, Alternating, and FLEXBCQ. FLEXBCQ leverages the useful optimization technique of UQ (flexible mapping) and the powerful expressive capability of BCQ (adaptive quantization levels), resulting in effective quantization.

in expressive power and inference speed. However, because of the lack of accurate quantization algorithms designed for BCQ, BCQ provides significantly lower accuracy than UQ despite its strong expressive power.

In this paper, we propose FLEXBCQ, an accurate optimization algorithm for BCQ. We decompose BCQ's quantization process as the composition of UQ and inner BCQ to leverage the advanced optimization techniques of UQ. This approach enables flexible mapping through its UQ component and adaptive quantization levels via inner BCQ, combining the strengths of both schemes, as shown in Figure 1 (d). We propose Unified Initialization technique which integrates the initialization methods of UQ and BCQ to initialize the quantization parameters of FLEXBCQ. We optimize the quantization parameters of FLEXBCQ on a small sample dataset by reconstructing blockwise outputs of quantized models. We propose novel optimization techniques such as Gradient Filtering and Periodic Remapping to maximize the accuracy of FLEXBCQ. Finally, through Composition Theorem (Theorem 1), we prove that the decomposed quantization process is able to merge back into a single BCQ after optimization, showing that FLEXBCQ maintains the fast inference speed of BCQ. Through extensive experiments with diverse models on various benchmarks, we find that FLEXBCQ shows up to 3.24%p higher accuracy than existing UQ and BCQ algorithms on MMLU 5-shot benchmark when quantizing a Llama-3 70B model into 3 bits.

We summarize the main contributions of this paper as follows:

- **Algorithm.** We propose FLEXBCQ, an accurate BCQ algorithm which exploits both sophisticated optimization techniques of UQ algorithms and the powerful expressive capability of BCQ algorithms. We propose useful techniques for optimizing BCQ models that effectively enhance the accuracy of quantized models. To the best of our knowledge, this is the first work that transfers UQ's useful optimization technique to optimize BCQ.
- **Experiments.** We conduct exhaustive experiments to verify the performance of FLEXBCQ. FLEXBCQ shows 3.24%p higher accuracy than existing UQ and BCQ algorithms on MMLU 5-shot benchmark when quantizing a Llama-3 70B model into 3 bits.
- **Analysis.** We analyze the quantized models generated by FLEXBCQ and demonstrate that FLEXBCQ successfully takes advantage of FlexRound's flexible mapping and binary-coding quantization's adaptive quantization levels, as we intended.

The rest of this paper is organized as follows. We formally define LLM quantization problem and provide preliminaries in Section 2. We propose FLEXBCQ in Section 3 and show our experimental results in Section 4. After introducing related works in Section 5, we conclude.

## 2 PRELIMINARY

### 2.1 LLM QUANTIZATION PROBLEM

We have an accurate LLM $f$, a desired bit-width $k$, and a sample dataset $\mathbb{D}$ of input token sequences. Our goal is to find an accurate $k$-bit quantized model $\widehat{f}_{(k)}$. In this paper, we focus on uniform quantization (UQ) and binary-coding quantization (BCQ) which are supported by a fast inference kernel, LUT-GEMM Park et al. (2024). We directly compare the accuracies of UQ and BCQ algorithms since they exhibit the same inference speed with LUT-GEMM when they have the same bit width.

### 2.2 UNIFORM AND BINARY-CODING QUANTIZATIONS

In LLM quantization, we gather a small number of weights as a group and assign quantization levels for each group to maximize the accuracy of quantized models. Given a weight group $\boldsymbol{w} \in \mathbb{R}^g$ of $g$

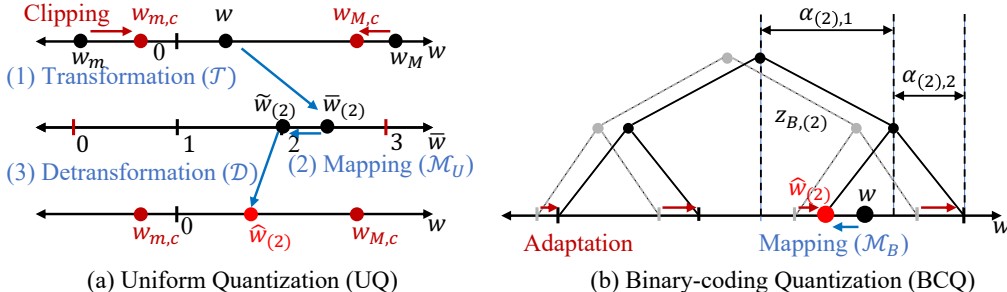

Figure 2: 2-bit quantization processes of UQ and BCQ. (a) UQ begins with clipping process, then quantize $w$ into $\widehat{w}_{(2)}$ through the sequential process of (1) Transformation ($\mathcal{T}$), (2) Mapping ($\mathcal{M}_{\mathcal{U}}$), and (3) Detransformation ($\mathcal{D}$). (b) BCQ begins with adapting its quantization levels, then quantizing $w$ into $\widehat{w}_{(2)}$ through Mapping ($\mathcal{M}_{\mathcal{B}}$) process. See Sections 2.2.1 and 2.2.2 for details.

weights and a desired bit-width $k$, quantizer $Q$ quantizes weights into $\widehat{\boldsymbol{w}}_{(k)} = Q(\boldsymbol{w}, k; \Theta)$ where $\Theta$ is a set of quantization parameters of $Q$. The quantization parameters are found through a calibration process of optimization on a sample dataset before quantizing weights. Each quantization scheme has its own quantizer and quantization parameters. Before proposing our method, we elaborate on the quantization process of UQ and BCQ as a background in Sections 2.2.1 and 2.2.2, respectively.

### 2.2.1 UNIFORM QUANTIZATION (UQ)

Uniform quantization (UQ) is a quantization scheme that has uniformly spaced quantization levels. Figure 2(a) illustrates how UQ quantizes a weight $w \in [w_m, w_M]$ into 2-bit weight $\widehat{w}_{(2)}$ where $w_m$ and $w_M$ are the minimum and maximum weights in the weight group $\boldsymbol{w}$, respectively. UQ's quantization process begins with clipping process, which determines the minimum $w_{m,c}$ and the maximum $w_{M,c}$ values of the clipped range which is the range of the values to represent after quantization. UQ's quantization parameters $\Theta_U = \{\Delta_{(k)}, z_{U,(k)}\}$ are determined based on the clipping range; $\Delta_{(k)} = (w_{M,c} - w_{m,c})/(2^k - 1)$ as a scale factor and $z_{U,(k)} = \lfloor -w_{m,c}/\Delta_{(k)} \rceil$ as a a zero-point where $\lfloor \cdot \rceil$ is a rounding function. After calibration, UQ quantizes $w$ using UQ quantizer $Q_U$ in Equation 1.

$$w \approx \widehat{w}_{(k)} = Q_U(w, k; \Theta_U) = \mathcal{D}(\mathcal{M}_U(\mathcal{T}(w; \Theta_U), k); \Theta_U) \tag{1}$$

Transformation $\mathcal{T}(w; \Theta_U)$, Mapping $\mathcal{M}_U(\bar{w}_{(k)}, k)$, Detransformation $\mathcal{D}(\widetilde{w}_{(k)}; \Theta_U)$ functions are defined as in Equations 2 to 4. $\bar{w}_{(k)}$, $\widetilde{w}_{(k)}$, and $\widehat{w}_{(k)}$ are transformed, mapped, and quantized weights in $k$ bits, respectively. Clamp$(\cdot, m, M)$ is a clamp function with min-max range $[m, M]$.

$$\mathcal{T}(w; \Theta_U) = \bar{w}_{(k)} = w/\Delta_{(k)} + z_{U,(k)} \tag{2}$$

$$\mathcal{M}_U(\bar{w}_{(k)}, k) = \widetilde{w}_{(k)} = \text{Clamp}(\lfloor \bar{w}_{(k)} \rfloor, 0, 2^k - 1) \tag{3}$$

$$\mathcal{D}(\widetilde{w}_{(k)}; \Theta_U) = \widehat{w}_{(k)} = \Delta_{(k)}(\widetilde{w}_{(k)} - z_{U,(k)}) \tag{4}$$

After quantization, we save the mapped weight $\widetilde{w}_{(k)}$ and quantization parameters in $Q_U$, then reconstruct the quantized weight $\widehat{w}$ using $\mathcal{D}(\widetilde{w}_{(k)}; \Theta_U)$ when we inference.

As described in Equation 3, the mapping process $M_U$ of UQ is a straightforward process that maps transformed weights $\bar{w}_{(k)}$ to the nearest integer. Therefore, introducing an advanced transformation process is essential to improve the accuracy of quantized models. FlexRound (Lee et al., 2023) enhances the accuracy of the quantized models by revising its transformation process as in Equation 5.

$$\mathcal{T}_{\mathcal{F}}(w; \Delta_{(k)}, s, s_r, z_{U,(k)}) = w/(\Delta_{(k)} \cdot s \cdot s_r) + z_{U,(k)} \tag{5}$$

$s$ and $s_r \in \mathbb{R}^+$ are scale factors that divide each weight before mapping; $s$ is assigned individually for each weight, while $s_r$ is shared across all weights in a row of a weight matrix. The modified transformation process enables weights to explore diverse quantization levels beyond the nearest one, and finally maps them to the quantization level that maximizes model accuracy.

### 2.2.2 BINARY-CODING QUANTIZATION (BCQ)

Binary-coding quantization (BCQ) is a non-uniform quantization scheme that has a set $\Theta_B = \{\boldsymbol{\alpha}_{(k)}, z_{B,(k)}\}$ of quantization parameters where $\boldsymbol{\alpha}_{(k)} \in \mathbb{R}^k$ is a vector of scale factors and $z_{B,(k)} \in \mathbb{R}$ is a shifting factor. $\boldsymbol{\alpha}_{(k)}$ and $z_{B,(k)}$ determine the quantization levels of BCQ by the summation and subtraction of scale factors in $\boldsymbol{\alpha}_{(k)}$ after shifting with $z_{B,(k)}$. For example, BCQ quantizer in Figure 2(b) has a set $\{z_{B,(2)} - \alpha_{(2),1} - \alpha_{(2),2}, z_{B,(2)} - \alpha_{(2),1} + \alpha_{(2),2}, z_{B,(2)} + \alpha_{(2),1} - \alpha_{(2),2}, z_{B,(2)} + \alpha_{(2),1} + \alpha_{(2),2}\}$ of quantization levels illustrated as the leaves of a binary tree. BCQ calibrates $\boldsymbol{\alpha}_{(k)}$ and $z_{B,(k)}$ on a sample dataset to adapt its quantization levels to maximize the accuracy of quantized models. After the adaption process, BCQ maps each weight to the corresponding quantization level by assigning a binary code $\boldsymbol{b}_{(k)} \in \{-1, +1\}^k$ of the quantization level to the weight. A BCQ quantizer $Q_B$ quantize a weight $w$ into $\widehat{w}_{(k)}$ as in Equation 6. $\mathcal{M}_B(w; \Theta_B)$ is a mapping function that maps weights to the nearest quantization level.

$$w \approx \widehat{w}_{(k)} = Q_B(w, k; \Theta_B) = \boldsymbol{\alpha}_{(k)}^T \boldsymbol{b}_{(k)} + z_{B,(k)}$$

$$where \;\; \boldsymbol{b}_{(k)} = \mathcal{M}_B(w; \Theta_B) = \arg\min_{\boldsymbol{b}'_{(k)}} ||w - (\boldsymbol{\alpha}_{(k)}^T \boldsymbol{b}'_{(k)} + z_{B,(k)})|| \tag{6}$$

After quantization, we save binary code $\boldsymbol{b}_{(k)}$ of each weight and quantization parameters in $Q_B$. We use Reconstruction function $\mathcal{R}_B(\boldsymbol{b}_{(k)}; \Theta_B) = \boldsymbol{\alpha}_{(k)}^T \boldsymbol{b}_{(k)} + z_{B,(k)}$ to reconstruct the quantized weight $\widehat{w}$ when we inference.

The main advantage of BCQ is its strong expressive capability; it has been proven that any UQ is representable in the form of BCQ (Appendix C in Park et al. (2024)). However, there has been limited research on optimization algorithms for BCQ, especially for LLMs. The only low-cost algorithm (Xu et al., 2018) that is applicable to LLMs with BCQ scheme does not take input distribution into account and ignores dependencies of different weight groups, resulting in low accuracy. As a result, its quantized models exhibit significantly lower accuracy when we use BCQ compared to the case when we use UQ despite its theoretical advantage.

## 3 PROPOSED METHOD

### 3.1 OVERVIEW

In this section, we propose FLEXBCQ, an accurate optimization algorithm for BCQ. The motivation behind FLEXBCQ is to leverage advanced optimization techniques designed for UQ to optimize BCQ while retaining BCQ's strong expressive capability. Before presenting our main idea, we outline key challenges that must be tackled.

**C1. Formulation.** How can we modify the quantizer of BCQ to leverage the useful optimization techniques designed for UQ without losing the strong expressive capability of BCQ?

**C2. Initialization.** How can we effectively initialize quantization parameters to accurately capture the distribution of the weights in the model?

**C3. Optimization.** How can we optimize quantization parameters on the sample dataset to maximize the accuracy of quantized models?

We address these challenges with the following main ideas:

**I1. FLEXBCQ (Section 3.2).** We decompose the quantization process of conventional BCQ into the composition of UQ and inner BCQ to utilize the optimization techniques from UQ. Decomposed BCQ takes advantage of both UQ's advanced optimization techniques and BCQ's strong expressive capability simultaneously.

**I2. Unified Initialization (Section 3.3).** We integrate the clipping range search algorithm of UQ and the alternating update algorithm of BCQ to initialize the quantization parameters of FLEXBCQ. Our initialization process is accurate by considering both clipping range and adaptive quantization levels at the same time.

**I3. Blockwise output reconstruction (Section 3.4).** We optimize the quantization parameters of FLEXBCQ by minimizing blockwise reconstruction errors. We propose novel optimization techniques including Gradient Filtering and Periodic Remapping to maximize the accuracy of the quantized models.

We carefully analyze the quantization processes of UQ and BCQ, and design FLEXBCQ to combine the strengths of both quantization schemes. Figure 3 illustrates the calibration and deployment

| | Calibration | | Deployment | |
|---|---|---|---|---|
| | Initialization | Quantization | Saving | Inference |
| (a) FlexRound (UQ) | $\boxed{C}$ | $w \xrightarrow[\text{Flex.}]{\mathcal{T}_F} \bar{w} \xrightarrow{\mathcal{M}_U} \widetilde{w} \xrightarrow{\mathcal{D}} \widehat{w}$ | $\widetilde{w} \in ([0, 2^{k-1}] \cap \mathbb{Z})^g$  $\Delta, z_U \in \mathbb{R}$ | $\widetilde{w} \xrightarrow{\mathcal{D}} \widehat{w}$ |
| (b) Alternating (BCQ) | $\boxed{A}$ | $w \xrightarrow[\text{Adapt.}]{\mathcal{M}_B} \widehat{w}$ | $\boldsymbol{B} \in \{-1,1\}^{k \times g}$  $\boldsymbol{\alpha} \in \mathbb{R}^k, z_B \in \mathbb{R}$ | $\boldsymbol{B} \xrightarrow{\mathcal{R}_B} \widehat{w}$ |
| (c) FLEXBCQ (BCQ) | $\boxed{C}\boxed{A}$ | $w \xrightarrow[\text{Flex.}]{\mathcal{T}_F} \bar{w} \xrightarrow[\text{Adapt.}]{\mathcal{M}_B} \widetilde{w} \xrightarrow{\mathcal{D}} \widehat{w}$ | $\boldsymbol{B} \in \{-1,1\}^{k \times g}$  $\boldsymbol{\alpha} \in \mathbb{R}^k, z_B \in \mathbb{R}$ | $\boldsymbol{B} \xrightarrow{\mathcal{R}_B^*} \widehat{w}$ |

$\boxed{C}$: Clipping range search"    $\boxed{A}$: Quantization level adaptation    Flex.: Flexible mapping    Adapt.: adaptive quantization levels

Figure 3: A comparison of the quantization processes of FlexRound, Alternating, and FLEXBCQ. FLEXBCQ benefits from both flexible mapping and adaptive quantization levels. $\boldsymbol{B}$ is a binary code matrix whose columns are binary codes of weights in $w$. We decompose $\widehat{w}$ in Equation 6 into $\boldsymbol{\alpha}$ and $\boldsymbol{B}$ when we save quantized weights. The notation $(k)$ for bit-width is omitted for simplicity.

phases of FlexRound (Lee et al., 2023) (UQ), Alternating (Xu et al., 2018) (BCQ), and FLEXBCQ. In calibration phases, FlexRound begins with initializing its quantization parameters through a clipping range search which finds a proper clipping range $[w_{m,c}, w_{M,c}]$. After initialization, it quantizes weights through the sequential process of transformation ($\mathcal{T}_F$), mapping ($\mathcal{M}_U$), and detransformation ($\mathcal{D}$). Flexible mapping, which is the main advantage of FlexRound, is achieved through its improved transformation process $\mathcal{T}_F$. On the other hand, Alternating initializes its quantization parameters through a quantization level adaptation process, and quantizes weights using the initialized parameters. The strength of Alternating lies in the mapping process ($\mathcal{M}_B$) to its adapted non-uniform quantization levels, which are adjusted during initialization.

FLEXBCQ integrates the strengths of both methods by first applying a transformation process $\mathcal{T}_F$ as FlexRound, followed by mapping ($\mathcal{M}_B$) to the adapted quantization levels within this transformed space ($\bar{w}$). This allows FLEXBCQ to take advantage of advanced UQ techniques, such as FlexRound's flexible mapping, while also benefiting from BCQ's adaptive quantization levels. Once calibration is completed, we quantize the pretrained model for deployments with its calibrated quantization parameters as in the "Deployment" column. FLEXBCQ shows identical saving and inference to the conventional BCQ algorithm by merging its integrated quantization process into a single BCQ quantization process ($\mathcal{R}_B^*$) based on Theorem 1. Therefore, there is no memory and latency overhead for utilizing flexible mapping in the calibration phase. We elaborate on the details of formulation, initialization, and optimization techniques of FLEXBCQ in the following sections.

## 3.2 FLEXIBLE BINARY-CODING QUANTIZATION (FLEXBCQ)

FLEXBCQ utilizes an inner BCQ in the transformed weight space $\bar{w}$ of UQ to leverage FlexRound's training techniques and BCQ's adaptive quantization levels at the same time. FlexBCQ has both FlexRound's quantization parameters $\Delta_{(k)}, z_{U,(k)}, s$, and $s_r$, as well as BCQ's quantization parameters $\boldsymbol{\alpha}_{(k)}$, and $z_{B,(k)}$ since it incoporates both BCQ and FlexRound. The quantizer $Q_F$ parameterized by $\Theta_F = \Theta_U \cup \Theta_B \cup \{s, s_r\}$ is defined in Equation 7. Equation 5.

$$w \approx \widehat{w}_{(k)} = Q_F(w, k; \Theta_F) = \mathcal{D}(\mathcal{M}_B(\mathcal{T}_F(w; \Theta_U, s, s_r); \Theta_B); \Theta_U) \tag{7}$$

As described in the Equation 7, $Q_F$ includes both $\mathcal{T}_F$ and $\mathcal{M}_B$ to leverages both FlexRound's flexible mapping and BCQ's adaptive quantization levels. After calibration, we save binary code $\boldsymbol{b}_{(k)}$ of each weight, $\Theta_B$ for BCQ's reconstruction process, and $\Theta_U$ for detransformation; we discard $s$ and $s_r$ which are used only for transformation. The Reconstruction function $R_F$ for FLEXBCQ is defined as in Equation 8.

$$\hat{w}_{(k)} = \mathcal{R}_F(\boldsymbol{b}_{(k)}; \Theta_B, \Theta_U) = \mathcal{D}(\mathcal{R}_B(\boldsymbol{b}_{(k)}; \Theta_B); \Theta_U) \tag{8}$$

As described in Equation 8, FLEXBCQ has memory and latency overhead due to $\Theta_U$ and $\mathcal{D}$ compared to the conventional BCQ's Reconstruction function $R_B(\boldsymbol{b}_{(k)}; \Theta_B)$. To address this issue, we propose Composition Theorem which integrates the expensive two-step reconstruction process of FLEXBCQ into a single process.

---

**Algorithm 1** Unified Initialization

---

**Require:** A weight group $\boldsymbol{w}$, a bit-width $k$, a number $N$ of iterations for grid search, and a number $T$ of iterations for quantization level adaptation

**Ensure:** Initialized UQ's scale factor $\Delta^*_{(k)}$, zero-point $z^*_{U,(k)}$, and a vector $\boldsymbol{\alpha}^*_{(k)}$ of BCQ's scale factors.

1: $w_m \leftarrow min(min(\boldsymbol{w}), 0)$, $w_M \leftarrow max(max(\boldsymbol{w}), 0)$
2: $z_{B,(k)} \leftarrow (2^k - 1)/2$, $s \leftarrow 1$, $s_r \leftarrow 1$, $e^* \leftarrow$ MAX_NUM
3: **for** $\gamma$ in $1/N, 2/N, ..., 1$ **do**
4:      $\Delta_{(k)} \leftarrow \gamma(w_M - w_m)/(2^k - 1)$, $z_{U,(k)} \leftarrow \lfloor -w_m/\Delta_{(k)} \rceil$
5:      $\bar{w}_{(k)} \leftarrow \mathcal{T}_F(\boldsymbol{w}; \Delta_{(k)}, z_{U,(k)})$                           ▷ Equation 5
6:      $\boldsymbol{\alpha}_{(k)} \leftarrow$ adapt-quant-level($\bar{w}_{(k)}$,$k$,$T$)       ▷ Algorithm 2 in (Xu et al., 2018)
7:      $\widehat{\boldsymbol{w}} \leftarrow Q_F(\boldsymbol{w}, k; \Theta_F)$                               ▷ Equation 7
8:      $e \leftarrow \|\boldsymbol{w} - \widehat{\boldsymbol{w}}_{(k)}\|_2^2$
9:      **if** $e < e^*$ **then**
10:          $\Delta^*_{(k)}, z^*_{U,(k)}, \boldsymbol{\alpha}^*_{(k)} \leftarrow \Delta_{(k)}, z_{U,(k)}, \boldsymbol{\alpha}_{(k)}$      ▷ Update quantization parameters
11:          $e^* \leftarrow e$                                  ▷ Update the minimum quantization error
12:      **end if**
13: **end for**

---

**Theorem 1** (Composition Theorem). *Given a BCQ Reconstruction function $\mathcal{R}_B(\boldsymbol{b}_{(k)}; \Theta_B)$ and a Detransformation function $\mathcal{D}(\widetilde{w}_{(k)}; \Theta_U)$ where $\widetilde{w}_{(k)} = \mathcal{R}_B(\boldsymbol{b}_{(k)}; \Theta_B)$. There is a BCQ Reconstruction function $\mathcal{R}^*_B(\boldsymbol{b}_{(k)}; \Theta^*_B) = \mathcal{D}(\mathcal{R}_B(\boldsymbol{b}_{(k)}; \Theta_B); \Theta_U)$ for any $\boldsymbol{b}_{(k)}$.*

*Proof.* See Appendix E.1.                                                 □

As a result, FLEXBCQ's expensive Reconstruction function $\mathcal{R}_F$ is substituted into a single BCQ's Reconstruction function $\mathcal{R}^*_B$, removing the memory and latency overhead.

### 3.3 UNIFIED INITALIZATION

After formulation, we need to optimize FLEXBCQ's quantization parameters in $\Theta_F$ to maximize the accuracy of the quantized models. We need an accurate algorithm to initialize the quantization parameters before optimization. We unify the initialization processes of UQ and BCQ since FLEXBCQ has the quantization parameters of both quantization schemes. We initialize $s$ and $s_r$ as 1 to make the flexible mapping mimic the traditional rounding-to-nearest (RTN) method at the beginning, following FlexRound (Lee et al., 2023). We initialize $z_{B,(k)} = (2^k - 1)/2$ and fix it since the transformed space, in which the inner BCQ is defined, is designed for mapping weights to the range $[0, 2^k - 1]$. We propose Unified Initialization for initializing the remaining quantization parameters $\Delta_{(k)}$, $z_{U,(k)}$, and $\boldsymbol{\alpha}_{(k)}$ as in Algorithm 1.

We precisely initialize quantization parameters through an iterative process that integrates a grid search-based clipping range search algorithm with the alternating quantization level adaptation (Algorithm 2 in (Xu et al., 2018)). In each iteration, we adjust the length $\gamma(w_M - w_m)$ of the clipping range by modifying the ratio $\gamma$ fixing the minimum value $w_m$. Within the adjusted clipping range, we compute the candidate scale factor $\Delta_{(k)}$ and candidate zero-point $z_{U,(k)}$ of UQ, followed by the flexible transformation process in Equation 5 (lines 4-5). We find a candidate vector $\boldsymbol{\alpha}_{(k)}$ of scale factors of BCQ through quantization level adaptation using the transformed weights $\bar{w}_{(k)}$ (line 6). After that, we compute the quantized weight $\widehat{w}_{(k)}$ following Equation 7 and compute the quantization error $e$ (lines 7-8). If $e$ is smaller than the minimum quantization error $e^*$ found through previous iterations, we update the quantization parameters $\Delta^*_{(k)}, z^*_{U,(k)}$, and $\boldsymbol{\alpha}^*_{(k)}$ with the candidate quantization parameters $\Delta_{(k)}, z_{U,(k)}$, and $\boldsymbol{\alpha}_{(k)}$, respectively. In this case, we also update $e^*$ with $e$ to find better quantization parameters. This process is performed independently for each weight group and each group has the precise quantization parameters after initialization.

### 3.4 BLOCKWISE OUTPUT RECONSTRUCTION

After initializing quantization parameters, we perform Blockwise Output Reconstruction process to optimize the quantization parameters in FLEXBCQ. From the bottom block to the top block,

we sequentially minimize the reconstruction loss $\mathcal{L}_i$ in Equation 9, which reduces the gap between outputs of the $i$-th blocks before and after quantization, using stochastic gradient descent. $f_i$ and $\hat{f}_i$ are the $i$-th blocks of the pretrained LLM $f$ before and after quantization, respectively. $\boldsymbol{X}_i$ and $\widehat{\boldsymbol{X}_i}$ are the inputs of $f_i$ and $\hat{f}_i$, respectively. $\boldsymbol{\theta}_i$ is the pretrained parameters in the $i$-th block of $f$.

$$\mathcal{L}_i = ||f_i(\boldsymbol{X}_i; \boldsymbol{\theta}_i) - \hat{f}_i(\widehat{\boldsymbol{X}_i}; \boldsymbol{\theta}_i, \Theta_F)||_F^2 \tag{9}$$

We optimize $\Delta_{(k)}$, $z_{U,(k)}$, $s$, $s_r$, and $\boldsymbol{\alpha}_{(k)}$ during the optimization process and do not update the pretrained model's parameters. We utilize straight-through estimator (STE) (Bengio et al., 2013) to update $\Delta_{(k)}$, $z_{U,(k)}$, $s$, $s_r$ corresponding to $\mathcal{T}_F$ since the mapping function $\mathcal{M}_B$ is not differentiable. We propose gradient filtering and periodic remapping to precisely optimize quantization parameters.

**Gradient Filtering.** Gradient filtering is a technique that filters the gradients of the weights that have large mapping errors, i.e., $|\bar{w}_{(k)} - \widetilde{w}_{(k)}|$, to stabilize the optimization process of FLEXBCQ. STE hypothesizes that the gradient of the transformed weight ($\bar{w}_{(k)}$) and mapped weight ($\widetilde{w}_{(k)}$) have the same gradients. However, if the difference between two values is significant, the hypothesis does not hold and it degrades the accuracy of the quantized models. Therefore, we set a hyperparameter $\tau$ and we filter the gradients of weights that have a mapping error larger than $\tau$; i.e. we zero-out the gradients of a weight $w$ if $|\bar{w}_{(k)} - \widetilde{w}_{(k)}| > \tau$.

**Periodic Remapping.** Periodic remapping is a technique designed to reduce the errors induced by the excessive change of mapping between weights and quantization levels. During an optimization process, $\boldsymbol{\alpha}_{(k)}$ is updated according to the mapping between weights and quantization levels, and the mapping between weights and quantization levels is changed by the update of $\boldsymbol{\alpha}_{(k)}$. The renewed mapping induces error since $\boldsymbol{\alpha}_{(k)}$ is updated according to the previous mapping and BCQ's mapping function $\mathcal{M}_B$ does not guarantee the reduction of the output reconstruction error. Therefore, we set a hyperparameter $p$ and update mapping in every $p$ steps, rather than every step.

## 4 EXPERIMENTS

We perform experiments to answer the following questions.

- **Q1. (General knowledge benchmark)** How accurate are the models quantized by FLEXBCQ on general knowledge benchmarks, e.g. MMLU?
- **Q2. (Task-specific knowledge benchmark)** How accurate are the models quantized by FLEXBCQ on task-specific knowledge benchmarks, e.g. GSM8K?
- **Q3. (Ablation study)** How do the main ideas of FLEXBCQ contribute to improving the accuracy of quantized models?
- **Q4. (Analysis)** Does FLEXBCQ effectively utilize flexible mapping and quantization levels?

### 4.1 EXPERIMENTAL SETUP

**Setup.** We use PyTroch (Paszke et al., 2019) and Transformers (Vaswani et al., 2017) libraries for implementation. We use Llama-3 8B, Llama-3 70B (Dubey et al., 2024), and Mistral 7B (Jiang et al., 2023) models for evaluating the amount of general knowledge in quantized models on MMLU (Hendrycks et al., 2021) benchmark. We use Llama-3 8B Instruct (Dubey et al., 2024) model for evaluating the amount of task-specific knowledge within quantized models on GSM8K (Cobbe et al., 2021) benchmark. We sample 128 token sequences of length 2048 from C4 (Raffel et al., 2020) and GSM8K (Cobbe et al., 2021) as a sample dataset to quantize models for general and task-specific benchmarks, respectively. We use a single A100 GPU for quantization.

**Baselines.** We directly compare the performance of FLEXBCQ with both UQ and BCQ algorithms since they have the same inference speed with the state-of-the-art inference kernel (Park et al., 2024) when they have the same bit width. We use Round-to-Nearest (RTN), OmniQuant (Shao et al., 2024), and FlexRound (Lee et al., 2023) as UQ competitors. We use Greedy and Alternating versions of algorithms in Xu et al. (2018) as BCQ competitors.

**Hyperparameters.** We use 3 and 4-bit weight-only quantization with a group size of 128. We optimize quantization parameters for 20 epochs with a batch size of 1 following OmniQuant Shao et al. (2024). We use a learning rate of 0.005 for all experiments with a cosine annealing learning rate scheduler (Loshchilov & Hutter, 2017). We set hyperparameters $N$, $T$, $p$, and $\tau$ to 50, 15, 2, and $min(\boldsymbol{\alpha}_{(k)})$, respectively. We use $p$ as 1 for Llama-3 70B model.

Table 1: Average accuracies of quantized models on 5-shot and 0-shot MMLU (Hendrycks et al., 2021) benchmarks. FLEXBCQ shows the highest or second-highest accuracy in all cases. Bold and underlined texts indicate the highest and second-highest accuracies, respectively.

| # Bits | Scheme | Method | Mistral 7B | | Llama-3 8B | | Llama-3 70B | |
|--------|--------|--------|------------|--------|------------|--------|-------------|--------|
| | | | 5-shot | 0-shot | 5-shot | 0-shot | 5-shot | 0-shot |
| | Full precision | | 62.57 | 60.22 | 65.02 | 60.52 | 78.89 | 76.15 |
| 4 | UQ | RTN | 60.91 | 57.93 | 62.11 | 57.79 | 66.10 | 69.56 |
| | | OmniQuant | 59.85 | 57.31 | 63.17 | **58.99** | 77.90 | 75.35 |
| | | FlexRound | 61.20 | 58.45 | 63.25 | 58.32 | **78.61** | **75.86** |
| | BCQ | Greedy | 24.77 | 25.45 | 24.90 | 24.51 | 22.95 | 22.95 |
| | | Alternating | 57.18 | 54.50 | 56.27 | 53.10 | 44.98 | 43.17 |
| | | FLEXBCQ | **61.38** | **59.03** | **63.81** | 58.56 | 78.36 | 75.61 |
| 3 | UQ | RTN | 53.01 | 50.69 | 38.12 | 39.50 | 46.50 | 34.82 |
| | | OmniQuant | 54.00 | 52.51 | 51.56 | 45.06 | 72.80 | 69.47 |
| | | FlexRound | 58.58 | 55.84 | 58.89 | **54.65** | 73.19 | 71.68 |
| | BCQ | Greedy | 24.70 | 24.66 | 27.15 | 26.84 | 22.95 | 22.95 |
| | | Alternating | 26.16 | 23.40 | 25.85 | 23.10 | 24.62 | 23.49 |
| | | FLEXBCQ | **59.08** | **56.53** | **59.33** | 54.49 | **76.43** | **73.14** |

Table 2: Accuracies of 3 and 4-bit quantized Llama-3 Instruct 8B models on GSM8K benchmark. FLEXBCQ outperforms all of the competitors in all cases.

| Scheme | Method | 4bit | 3bit |
|--------|--------|------|------|
| | Full precision | | 76.12 |
| UQ | RTN | 70.89 | 22.37 |
| | OmniQuant | 70.36 | 45.79 |
| | FlexRound | 73.39 | 64.67 |
| BCQ | Greedy | 0.00 | 0.00 |
| | Alternating | 56.79 | 0.08 |
| | FLEXBCQ | **75.44** | **67.22** |

Table 3: Accuracies of quantized models with and without our main ideas. We evaluate the accuracy of quantized models on MMLU 5-shot benchmark. All of our main ideas, especially gradient filtering, improve the accuracy of FLEXBCQ. We analyze the effect of unified initialization in Section 4.5.

| Method | 4bit | 3bit |
|--------|------|------|
| FLEXBCQ | **63.81** | **59.33** |
| - flexible mapping | 62.83 | 57.45 |
| - gradient filtering | *NaN* | |
| - periodic remapping | 63.70 | 58.18 |

## 4.2 GENERAL KNOWLEDGE BENCHMARK

We compare the accuracies of quantized models on 5-shot and 0-shot MMLU benchmarks to estimate the amount of general knowledge within quantized models. Table 1 summarizes the evaluation results. The results show that FLEXBCQ achieves the highest performance or the second-highest performance by a small margin in all cases. In the 3-bit quantization case of the Llama-3 70B model, FLEXBCQ outperforms the second-best model by a substantial margin of 3.24% in the 5-shot setting and 1.46% in the 0-shot setting. This demonstrates that FLEXBCQ effectively quantizes models while preserving their general knowledge. In contrast, in the 4-bit quantization results, FLEXBCQ performs 0.25% lower than FlexRound in both 0-shot and 5-shot settings. This slight drop in performance is suspected due to the FLEXBCQ's strong fitting capability, which may have caused slight overfitting to a small sample dataset.

## 4.3 TASK-SPECIFIC KNOWLEDGE BENCHMARK

We compare the accuracies of 3 and 4-bit quantized Llama-3 8B Instruct model on GSM8K, which focuses exclusively on mathematical problems. We summarize the experimental results in Table 2. The experimental results show that FLEXBCQ achieves the highest performance, outperforming the second-best competitor by 2.55% and 2.05% in the 3-bit and 4-bit experiments, respectively. Notably, while most BCQ competitors demonstrate less than 1% accuracy in almost all cases, FLEXBCQ consistently delivers significantly higher accuracy than them. In task-specific knowledge benchmarks, the distribution between sample and test datasets is more closely aligned than in general knowledge benchmarks. This suggests that FLEXBCQ 's strong fitting capability, enabled by

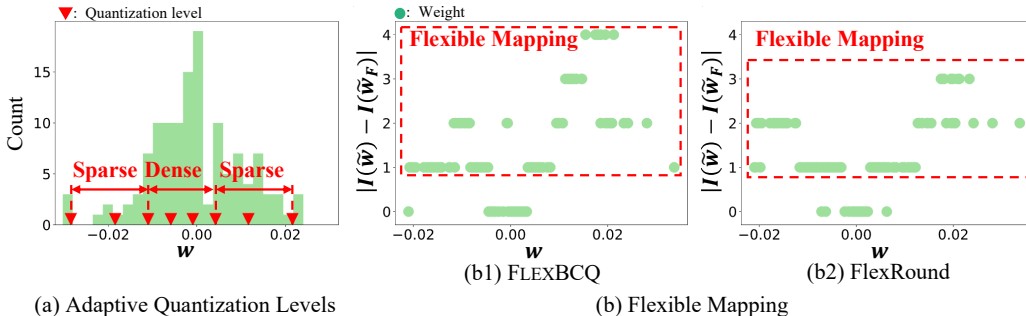

(a) Adaptive Quantization Levels  (b) Flexible Mapping

Figure 4: Visualizations of (a) the adaptive quantization levels produced by FLEXBCQ and (b) flexible mappings of weights quantized by FLEXBCQ and FlexRound (see Section 4.5 for details). FLEXBCQ successfully adapts its quantization levels to the distribution of weights and enables weights to plentifully explore flexible mappings analogous to FlexRound. $I(\widetilde{w}_F)$ and $I(\widetilde{w})$ represent the indices of the quantization levels mapped to the weights when flexible transformation is applied and not applied, respectively.

Table 4: Weight quantization errors and MMLU accuracies of the 3-bit quantized models initialized by unified initialization (unified init.) and competitors. We report the scaled quantization error by multiplying $10^6$ for simplicity. Unified initialization shows the lowest weight quantization errors in all types of layers leading to the highest accuracy. See Section 4.5 for details.

| Scheme | Initialization | Weight Quantization Error ($\times 10^6$, ↓) | | | | | | | MMLU (↑) | |
|---|---|---|---|---|---|---|---|---|---|---|
| | | Q | K | V | O | U | G | D | 5-shot | 0-shot |
| UQ | Grid Search | 16.33 | 37.67 | 2.15 | 2.28 | 5.39 | 4.35 | 4.41 | 38.12 | 39.50 |
| BCQ | Alternating | 16.88 | 38.40 | 1.82 | 2.08 | 5.00 | 4.03 | 4.05 | 25.85 | 23.10 |
| BCQ | Unified Init. | **12.57** | **28.50** | **1.61** | **1.86** | **4.46** | **3.61** | **3.63** | **39.93** | **40.51** |

its adaptive quantization levels, contributes to its superior performance. In conclusion, FLEXBCQ resolves the optimization deficiencies seen in existing BCQ algorithms, fully utilizing the expressive power that BCQ offers, resulting in a remarkable performance on a task-specific benchmark.

## 4.4 ABLATION STUDY

We conduct an ablation study to show the powerfulness of our main ideas. We summarize the result of the ablation study in Table 3. "-flexible mapping" refers to the case where only the BCQ quantization parameters are used without flexible transformation ($\mathcal{T}_F$). "-gradient filtering" denotes the result when optimization is performed using all gradients without filtering. Lastly, "-periodic remapping" refers to the case where weights are remapped to quantization levels at every step during optimization. The experimental results show that all three techniques significantly contribute to improving the accuracy of the quantized model. In particular, the results show that it is impossible to optimize the quantization parameters without applying gradient filtering technique. We suspect that this phenomenon occurs due to the weight clipping process within unified initialization. The weight clipping process generates weights with severe mapping errors which violate the hypothesis of straight through estimator (STE) that the gradient of a weight is the same as the gradient of the mapped weight. Unified initialization is an algorithm proposed to initialize FLEXBCQ, and there is no existing alternative algorithm to replace it. Therefore, we demonstrate its effectiveness in Section 4.5 by comparing it with existing methods for initializing UQ and BCQ.

## 4.5 ANALYSIS

Thus far, we have validated the superiority of FLEXBCQ and the proposed techniques based on the benchmark scores of quantized models. In this section, beyond the benchmark scores, we analyze a quantized model produced by FLEXBCQ to verify whether the intended mechanisms function as expected. Specifically, we examine whether FLEXBCQ effectively learns quantization levels adapted to the weight distribution, whether weights explore diverse quantization levels through flexible mapping, and whether unified initialization reduces weight reconstruction error more effectively than existing parameter initialization methods.

Figure 4(a) visualizes the distribution of weights within a weight group and the quantization levels of the group learned by FLEXBCQ. As shown in the figure, FLEXBCQ assigns dense quantization levels near zero, where most of the weights are concentrated, while assigning sparse quantization levels in regions where fewer weights are distributed. This demonstrates that FLEXBCQ indeed learns quantization levels adapted to the weight distribution.

Figure 4(b) shows the difference in indices of quantization levels of weights in a weight group in a quantized model, comparing cases where flexible transformation is applied versus when it is not. $I(\widetilde{w}_F)$ and $I(\widetilde{w})$ represent the indices of the quantization levels mapped to the weights when flexible transformation is applied and not applied, respectively. The difference between these indices indicates the extent of flexible mapping. We compare the results of the model quantized by FLEXBCQ (b1) and FlexRound (b2), which proposed flexible mapping. The results show that sufficient flexible mapping occurs in both methods, demonstrating that FLEXBCQ successfully exploits flexible mapping. Combining both results in Figure 4, FLEXBCQ successfully leverages both UQ's advanced optimization technique of flexible mapping and BCQ's adaptive quantization levels, as we intended.

Table 4 compares the performance of the unified initialization algorithm proposed in this paper with other initialization algorithms used in existing UQ and BCQ algorithms. We compare the pre- and post- quantization differences in weights by calculating $||w - \widehat{w}||_2^2$. We use the weights in the query (Q), key (K), value (V), out (O), up (U), gate (G), and down (D) projections within a single Transformer block for comparison. We compare the accuracy of models initialized with each initialization method to evaluate the impact of the initialization techniques on model accuracy. The experimental results show that unified initialization algorithm results in significantly lower errors than competitors in all cases, which leads to high accuracy. Notably, although Alternating algorithm (Xu et al., 2018) has the same expressive power as unified initialization, its performance is even lower than UQ, suggesting that we need an effective initialization method to fully leverage BCQ's strong expressive capability, and unified initialization fulfills this need. FLEXBCQ adopts an extensible approach that introduces and modifies the transformation function to BCQ. Unified initialization is effectively utilized to initialize the quantization parameters of techniques following this methodology.

## 5 RELATED WORK

Despite BCQ's (Kwon et al., 2022; Xu et al., 2018) strong expressive power, it has been under-investigated to uniform quantization (Lin et al., 2024; Ashkboos et al., 2024; Liu et al., 2024), which has a weaker expressive capability, mainly due to the lack of accelerated kernels for BCQ. To the best of our knowledge, there has been no research on accurately compressing LLMs using the BCQ scheme. In this context, the most important research related to BCQ is the development of the LUT-GEMM kernel (Park et al., 2024), which enables BCQ and UQ to be accelerated at the same speed. LUT-GEMM leverages the fact that BCQ's binary code is composed of only 1s and -1s, resulting in repetitive operations regardless of the weight values. LUT-GEMM enables fast general matrix multiplication (GEMM) by referencing the look-up table (LUT) constructed based on this observation. LUT-GEMM has demonstrated that it performs inference with the OPT-175B (Zhang et al., 2022) model using BCQ faster than OPTQ (Frantar et al., 2023) and AWQ (Lin et al., 2024), which support the acceleration of UQ, under small batch size constraints. Although LUT-GEMM has demonstrated excellent performance under constrained environments, subsequent works, such as You et al. (2024), indicate that fast kernels supporting BCQ are evolving. Precise BCQ algorithms, such as FLEXBCQ, are crucial since they boost the studies of fast inference kernels for BCQ.

## 6 CONCLUSION

In this paper, we propose FLEXBCQ, an accurate optimization algorithm for binary-coding quantization (BCQ). Our motivation is to integrate uniform quantization (UQ)'s advanced optimization technique into BCQ's powerful expressive capability. We find that the UQ's advanced optimization techniques stem from its transformation process and propose a novel formulation that utilizes the UQ's transformation process for BCQ without latency overhead. Combined with our effective initialization and optimization techniques, FLEXBCQ shows 3.24%p higher accuracy than existing UQ and BCQ algorithms on MMLU 5-shot benchmark when quantizing Llama-3 70B into 3 bits.

## 7 REPRODUCIBILITY STATEMENT

We report detailed hyperparameter settings including learning rate, learning rate scheduler, batch size, the number of epochs, the number of iterations for unified initialization, and the remapping period in Section 4.1, and another implementation detail regarding BCQ's mapping function in Appendix C to promote the reproducibility of the experimental results. We include the proof of Composition Theorem in Section E.1 in Appendix for completeness. We provide an in-depth analysis of quantized models generated by FLEXBCQ that can be used for examining reproduced results with figures and a table using the additional 10th page.

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

## A  TERMINOLOGY

To promote clarity, we summarize the definitions of terminologies frequently used in this paper.

**Units in LLMs.** We summarize the definitions of units in Large Language Models (LLMs) from a weight to a model. Figure 5 exhibits an example of a Transformer-based LLM with $L$ blocks.

- **Weight:** the smallest unit, representing an individual numerical weight value.
- **Weight group:** a collection of weights grouped by a specified group size, all of which share the same quantization parameters.
- **Weight matrix:** a two-dimensional matrix composed of weights, containing multiple weight groups.
- **Layer:** a component that performs affine transformations using a weight matrix and a bias vector. For example, Q, K, V, O, U, G, and D in Table 4 of the paper represent individual layers.
- **Module:** a collection of layers that performs a specific function. In Transformers, modules include Multi-Head Attention (MHA) and Multi-Layer Perceptron (MLP).
- **Block:** a fundamental unit of a Transformer, consisting of one MHA module and one MLP module.
- **Model:** a complete language model composed of multiple blocks. An LLM refers to a model.

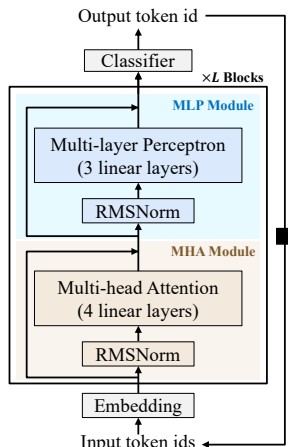

Figure 5: An illustration of a Transformer-based LLM.

**Quantization error and reconstruction error.** Assume that we have a weight matrix $W$ and we quantize $W$ into $k$-bit and obtain $\widehat{W}$. If we have input matrices $X$ and $\widehat{X}$ for $W$ and $\widehat{W}$, respectively, then the quantization error $E_{quant}$ and reconstruction error $E_{recon}$ are defined as follows:

$$E_{quant} = ||W - \widehat{W}|_F^2 \;\; and \;\; E_{recon} = ||WX - \widehat{W}\widehat{X}|_F^2$$

Reducing $E_{recon}$ provides more accurate models than when reducing $E_{quant}$ since $E_{recon}$ takes input distribution into account.

## B  SYMBOLS AND DEFINITIONS

We summarize symbols and their definitions that are frequently used in this paper in Table 5.

Table 5: Symbols and definitions.

| Symbols | Definitions |
|---|---|
| **Weights and Bit-width** | |
| $w$ | A weight |
| $\boldsymbol{w}$ | A weight group |
| $k$ | A constant for representing a bit-width of $k$ |
| $(\cdot)_{(k)}$ | A subscript for representing that $(\cdot)$'s bit-width is $k$ |
| $\widetilde{w}_{(k)}$ | A transformed weight |
| $\bar{w}_{(k)}$ | A mapped weight |
| $\widehat{w}_{(k)}$ | A quantized weight |
| $\boldsymbol{b}_{(k)}$ | A binary code of a weight |
| **Functions** | |
| $\mathcal{T}$ | A transformation function |
| $\mathcal{T}_F$ | A flexible transformation function |
| $\mathcal{M}_U$ | A mapping function of UQ (rounding and clamping) |
| $\mathcal{M}_B$ | A mapping function of BCQ (map to the nearest) |
| $\mathcal{D}$ | A detransformation function |
| $\mathcal{R}_B$ | A reconstruction function of BCQ |
| $\mathcal{R}_F$ | A reconstruction function of FLEXBCQ |
| $\mathcal{R}_B^*$ | An integrated reconstruction function of FLEXBCQ |
| **Quantizers** | |
| $Q_U$ | A UQ quantizer |
| $Q_B$ | A BCQ quantizer |
| $Q_F$ | A FLEXBCQ quantizer |
| **Quantizatin Parameters** | |
| $\Theta_U$ | A set of quantization parameters of UQ |
| $\Theta_B$ | A set of quantization parameters of BCQ |
| $\Theta_F$ | A set of quantization parameters of FLEXBCQ |
| $\Delta_{(k)}$ | A sacle facor of UQ |
| $z_{U,(k)}$ | A zoro-point of UQ |
| $s$ | An element-wise scale factor for flexible mapping |
| $s_r$ | A row-wise scale factor for flexible mapping |
| $\boldsymbol{\alpha}$ | A vector of scale factors of BCQ |
| $z_{B,(k)}$ | A shifting factor of BCQ |
| **Hyperparameterss** | |
| $N$ | The number of iterations for clipping range search |
| $T$ | The number of iterations for alternating updates |
| $p$ | A remapping period |
| $\tau$ | A threshold for Gradient Filtering |
| **Others** | |
| $f$ | A pretrained LLM |
| $\hat{f}$ | A $k$-bit quantized pretrained LLM |
| $f_i$ | The $i$-th block of $f$ |
| $X_i$ | The input matrix of $f_i$ |
| $\boldsymbol{\theta}_i$ | The parameters matrix of $f_i$ |
| $\mathcal{L}_i$ | The block reconstruction loss of $f_i$ |

# C    IMPLEMENTATION DETAILS OF FLEXBCQ

In this section, we discuss the implementation details necessary to reproduce the performance of FLEXBCQ reported in this paper. We first summarize hyperparameter settings used in the main text. Then we summarize the performance variations of FLEXBCQ, emphasizing the importance of implementing the mapping function, along with the impacts of the remapping period, Gradient Filtering threshold, optimization epochs, sample dataset, and clipping strategy.

## C.1    HYPERPARAMETER SETTINGS

Since FLEXBCQ employs a Blockwise Output Reconstruction process that requires additional hyperparameters, we aim to demonstrate that the outstanding performance of FLEXBCQ is achieved without expensive hyperparameter tuning. To this end, we fix all hyperparameters except for the Remapping Period $p$ and utilize only two combinations of hyperparameters across all cases, including the experiments presented in the Appendix, except for the sensitivity analysis. These combinations are outlined in Table 6.

Table 6: Hyperparameter settings of FLEXBCQ

| Hyperparameter | Setting |
|---|---|
| Learning rate | 0.005 |
| Clipping range search iterations ($N$) | 50 |
| Alternating update iterations ($T$) | 15 |
| Remapping Period ($p$) | 1, 2 |
| Gradient Filtering threshold ($\tau$) | $min(\boldsymbol{\alpha}_{(k)})^*$ |

\* Minimum value among $\boldsymbol{\alpha}_{(k)}$ for each weight group.

## C.2    IMPLENETATION OF BCQ'S MAPPING FUNCTION

One critical implementation detail is about the computational speed of the mapping function $\mathcal{M}_B$ of BCQ. In Xu et al.'s Algorithm 1 (Xu et al., 2018), this mapping function is implemented using a binary search tree (BST), but because of its sequential nature, it is significantly slow and makes training infeasible within a reasonable time. Therefore, in this paper, we replace this operation with the following GPU-friendly process: (1) calculate the values of all quantization levels, (2) compute the distance between each weight and every quantization level in parallel, (3) return the index of the closest quantization level for each weight, (4) mapping the weight to the corresponding quantization level found in (3). We named this process as Direct BCQ Mapping process. Table 7 compares the running time of the two mapping functions when they map all weights in the weight matrix of size $4096 \times 4096$ to corresponding quantization levels, assuming they have group sizes of 128. We use 3-bit and 4-bit quantization settings, meaning there are 8 and 16 quantization levels, respectively. We report the average running time of both functions using 100 runs.

Table 7: Running time of Xu et al's BCQ mapping function and Direct mapping function used in our paper. The ratio represents the multiple of time taken compared to Direct.

| Mapping Function | 4bit | | 3bit | |
|---|---|---|---|---|
| | Time (s) | Ratio ($\times$) | Time (s) | Ratio ($\times$) |
| BST (Xu et al., 2018) | 0.1694 | 23.62 | 0.5250 | 37.31 |
| Direct (ours) | 0.0072 | 1.00 | 0.0141 | 1.00 |

As depicted in the table, Direct mapping requires a significantly shorter time than BST-based mapping function used in Xu et al. (2018). Note that mapping functions are used in every weight matrix in every forwarding step and BST requires infeasible time for Blockwise Output reconstruction process. Therefore, the implementation detail of BCQ's mapping function is essential for implementing FLEXBCQ and reproducing the experimental results.

## C.3 SENSITIVITY ANALYSIS REGARDING $p$ AND $\tau$

Table 8: MMLU benchmark accuracies of 3-bit Llama-3 8B models quantized by FlexBCQ with different remapping period $p$. $p = 1$ represents the case that we remap weights to quantization levels in every step.

| p | 5-shot | 0-shot | Average |
|---|--------|--------|---------|
| 1 | 58.87 | 53.98 | 56.42 |
| 2 | 59.33 | 54.49 | 56.91 |
| 4 | 59.05 | 52.91 | 55.98 |
| 8 | 59.23 | 55.13 | 57.18 |
| 16 | 59.12 | 54.23 | 56.68 |

Table 9: MMLU benchmark accuracies of 3-bit Llama-3 8B models quantized by FlexBCQ with different gradient filtering threshold $\tau$. $min(\boldsymbol{\alpha}_{(k)})$ represents that $\tau$ is equal to the minimum value of $\boldsymbol{\alpha}_{(k)}$ in each weight group.

| $\tau$ | 5-shot | 0-shot | Average |
|--------|--------|--------|---------|
| 0.25 | 57.95 | 52.63 | 55.29 |
| 0.5 | 58.68 | 55.03 | 56.85 |
| 1 | 58.98 | 54.03 | 56.50 |
| $min(\boldsymbol{\alpha}_{(k)})$ | 59.33 | 54.49 | 56.91 |

**Remapping period ($p$).** Periodic Remapping is a technique that performs weight remapping not at every step during Blockwise Output Reconstruction but at intervals of $p$ steps. Table 8 presents the performance variations of FLEXBCQ across different remapping periods.

The experimental results indicate that the performance of FLEXBCQ exhibits slight variations based on the remapping period $p$. When $p = 2$, as employed in the main text, FLEXBCQ delivers strong performance. Moreover, when $p = 8$, FLEXBCQ shows even higher accuracy than the performance reported in the paper. In conclusion, using $p = 2$ without hyperparameter tuning is sufficient to achieve the excellent results proposed in the paper, except for Llama-3 70B which requires $p = 1$. Additionally, further performance gains are realized by tuning $p$ for specific configurations.

**Gradient Filtering threshold $\tau$.** We adopt a gradient filtering technique to exclude gradients that could negatively impact performance during optimization and introduce hyperparameter $\tau$ as the filtering threshold. We select $\tau$ in FLEXBCQ taking its mapping process into account.

In FLEXBCQ, the mapping process is performed in the space of transformed weights and we map each weight to its nearest quantization level. If integers in the range $[0, 2^k - 1]$ are used as quantization levels, as shown in Figure 2(a), this corresponds to uniform quantization (UQ). Alternatively, if a binary tree is used to represent non-uniform quantization levels, as shown in Figure 2(b), it corresponds to FLEXBCQ.

For simplicity, we first consider the case of uniform quantization. Weights within the clipping range $[w_{m,c}, w_{M,c}]$ are transformed to real values within $[0, 2^k - 1]$, while weights outside this range take on values beyond, e.g. $w_m$ and $w_M$. These out-of-range weights cause substantial mapping errors. In the transformed space, the interval between quantization levels in UQ is 1, and each level has a range of 0.5 on either side. Thus, it is reasonable to allow a margin of 0.5 even for the extreme levels 0 and $2^{k-1}$.

For FLEXBCQ, the value of 0.5 is replaced by the smallest scale factor $min(\boldsymbol{\alpha}_{(k)})$ in BCQ. Therefore, $\tau$ is assigned as the smallest scale factor in each weight group, automatically providing a threshold suited to the group's characteristics.

To evaluate the effectiveness of selecting $\tau$ as $min\boldsymbol{\alpha}_{(k)}$, we compare it against the UQ-based threshold of 0.5 from (2), as well as alternative thresholds of 0.25 (half of 0.5) and 1.0 (double of 0.5). The results of these experiments are presented in Table 9. As shown in the table, $min(\boldsymbol{\alpha}_{(k)})$, which is employed in our paper, shows the highest accuracy by providing the adaptive gradient filtering threshold for each weight group.

## C.4 Sensitivity Analysis Regarding Sample Dataset

In this section, we analyze the effect of selecting a sample dataset and the size of the sample dataset on the performance of FLEXBCQ. We use Llama-3 8B model for the experiments.

**Selecting Sample dataset** FLEXBCQ performs Blockwise Output Reconstruction based on SGD using a small size of a sample dataset and selecting the sample dataset has a critical impact on the performance of the quantized models. Based on previous studies (Lee et al., 2023; Frantar et al., 2023; Lee et al., 2023), we conduct experiments using the C4 dataset in the main text. In this section, we aim to examine how the performance of FLEXBCQ changes when using the SlimPajama (Soboleva et al., 2023) and FineWeb-Edu dataset (Lozhkov et al., 2024).

Table 10: MMLU benchmark accuracies of 3-bit quantized Llama-3 8B models on C4 and SimPajama datasets. Difference column indicates the amount of improvement in average accuracy over the case of using C4.

| Dataset | 5-shot | 0-shot | Average | Difference |
|---|---|---|---|---|
| C4 | 59.33 | 54.49 | 56.91 | 0 |
| SlimPajama | 59.78 | 55.92 | 57.86 | 0.94 |
| FineWeb-Edu | 59.14 | 55.67 | 57.41 | 0.50 |

Experimental results indicate that using the SlimPajama dataset and FineWeb-Edu dataset achieves an average accuracy improvement of 0.94% and 0.5%, respectively. This enhancement is attributable to the extensive preprocessing of those datasets, such as the removal of duplicates Soboleva et al. (2023). These results underscore that the performance of a quantized model is influenced by the characteristics of the sample dataset used for calibration.

**Sensitivity on the size of sample dataset.** To illustrate the effect of sample dataset size on the performance of FLEXBCQ, we quantized the Llama-3 8B model to 3 bits using sample datasets of varying sizes and evaluated its performance on the MMLU benchmark. We use sample datasets ranging in size from 32 to 256, with 128 being the specific size used in our paper. Each sample dataset comprises 2,048 tokens. We summarize the result in Table 11.

Table 11: MMLU 0-shot and 5-shot accuracies of 3-bit quantized Llama-3 8B models on various sizes of sample datasets.

| Benchmark | 32 | 64 | 128 | 256 |
|---|---|---|---|---|
| 5-shot | 58.76 | 59.36 | 59.33 | 59.66 |
| 0-shot | 54.53 | 54.70 | 54.49 | 54.63 |
| Average | 56.65 | 57.03 | 56.91 | 57.15 |

Experimental results indicate that the performance of the quantized models remains largely stable except in cases where the sample dataset size is extremely small (e.g., 32). This is attributed to the fact that even the largest sample dataset (with 256 samples), consisting of tokens represents only approximately one ten-millionth of the tokens used to train the Llama-3 8B model Dubey et al. (2024). As such, marginal increases in dataset size do not result in significant performance changes. Based on these observations, the dataset size of 128, used in our study, is appropriate for comparing the accuracy of different quantization methods.

## C.5 SENSITIVITY ANALYSIS REGARDING OPTIMIZATION EPOCHS

In our experimental setup, we adopt OmniQuant (Shao et al., 2024)'s SGD-based optimization framework and set the number of epochs to 20. Table 12 illustrates the impact of varying the number of epochs on FLEXBCQ 's performance.

Table 12: Accuracy variation of 3-bit quantizaed Llama-3 8B models using FLEXBCQ on the MMLU benchmark across different epoch settings.

| Benchmark | 10 | 20 | 30 |
|---|---|---|---|
| 5-shot | 59.17 | 59.33 | 59.23 |
| 0-shot | 54.28 | 54.49 | 53.65 |
| Average | 56.72 | 56.91 | 56.44 |

The experimental results demonstrate that FLEXBCQ's performance is not critically affected by the number of epochs, and the 20-epoch setting used in our experiments achieves the highest average accuracy. Overfitting appears to occur at approximately 30 epochs, further validating that training with 20 epochs is a proper choice.

## C.6 ANALYZING THE EFFECT OF CLIPPING STRATEGY

There are three possible strategies for clipping range search used in Unified Initialization: (a) Fixed Minimum strategy which fixes the minimum weight and adjusts only the maximum value, (b) Fixed Maximum strategy which fixes the maximum weight and adjusts only the minimum value, and (c) Balanced strategy which adjusts both the minimum and maximum values to find the clipping range. The performance of these strategies for quantizing the Llama-3 8B model into 3 bits is presented in Table 13.

Table 13: Accuracies of 3-bit quantized Llama-3 8B models with different clipping strategies on MMLU benchmark.

| Clipping Strategy | 5-shot | 0-shot | Average |
|---|---|---|---|
| Fixed Minimum | 59.33 | 54.49 | 56.91 |
| Fixed Maximum | 59.23 | 54.99 | 57.11 |
| Balanced | 58.77 | 54.59 | 56.68 |

Experimental results reveal that while all three strategies deliver comparable performance, the Fixed Minimum strategy achieves a higher score than the Balanced strategy. Therefore, Fixed Minimum, used in Algorithm 1, is a proper strategy for initialization. Furthermore, as shown in the table, using Fixed Maximum strategy provides a slight performance improvement.

## D IMPLEMENTATION DETAILS OF COMPETITORS

### D.1 OMNIQUANT (SHAO ET AL., 2024)

We use the official implementation[1] of OmiQuant. We use the hyperparameters reported in the paper and GitHub[1] for running OmniQuant.

### D.2 FLEXROUND (LEE ET AL., 2023)

We implement the FlexRound following the paper (Lee et al., 2023). Since FlexRound and FLEXBCQ have similar quantization and optimization processes, we implement FlexRound on top of our implementation of FLEXBCQ besides its UQ quantizer. We use the same hyperparameter settings, e.g. learning rate, for fair comparison.

---

[1]https://github.com/OpenGVLab/OmniQuant

### D.3 RTN

We use clipping range search with 50 iterations for implementing RTN which is the same as the number of iterations used in unified initialization. We use the same source code for clipping range search in unified quantization.

### D.4 GREEDY AND ALTERNATING (XU ET AL., 2018)

We implement Greedy and Alternating based on the paper (Xu et al., 2018) besides its BCQ mapping function as explained in Section C.2. We use alternating update with 15 iterations for implementing Alternating which is the same as the number of iterations used in unified initialization. We use the same source code for clipping range search in unified quantization.

## E DETAILS OF COMPOSITION THEOREM

Composition Theorem is an essential part of FLEXBCQ enables us to exploit flexible mapping without any memory and latency overhead compared to the conventional BCQ algorithms. We first provide a proof of Composition Theorem and perform an in-depth analysis of its effect.

### E.1 PROOF OF THEOREM 1

*Proof.* By the definitions of $\mathcal{R}(\boldsymbol{b}_{(k)}; \Theta_B)$ and $\mathcal{D}(\widetilde{w}_{(k)}; \Theta_U)$, we reduce $\mathcal{D}(Q_B(\boldsymbol{b}_{(k)}; \Theta_B); \Theta_U)$ as follows.

$$\mathcal{D}(\mathcal{R}_B(\boldsymbol{b}_{(k)}; \Theta_B); \Theta_U) = \Delta_{(k)}((\boldsymbol{\alpha}_{(k)}^T \boldsymbol{b}_{(k)} + z_{B,(k)}) - z_{U,(k)})$$
$$= (\Delta_{(k)} \boldsymbol{\alpha}_{(k)})^T \boldsymbol{b}_{(k)} + \Delta_{(k)}(z_{B,(k)} - z_{U,(k)})$$

Thus, $\boldsymbol{\alpha}_{(k)}^* = \Delta_{(k)} \boldsymbol{\alpha}_{(k)}$, $z_{B,(k)}^* = \Delta_{(k)}(z_{B,(k)} - z_{U,(k)})$, and $\Theta_B^* = \{\boldsymbol{\alpha}_{(k)}^*, z_{B,(k)}^*\}$. □

### E.2 EFFECT OF COMPOSITION THEOREM

In this section, we analyze the effect of Composition Theorem by comparing the size and latency of the quantized models when we use and do not use Composition Theorem. Assume that we have a group $\boldsymbol{w} \in \mathbb{R}^g$ of weights consists of $g$ weights, and we quantize $\boldsymbol{w}$ into $k$-bit using Alternating, FLEXBCQ without and with Composition Theorem. Then, quantization results for saving, number of bits for saving, and reconstruction process for inference for each case is summarized in Table 14 We assume we use 16 bits to save quantization parameters following our experimental setting.

Table 14: Comparison of quantized results, number of bits for saving, and reconstruction process of Alternating, FLEXBCQ without and with Composition Theorem, when quantizing a group $\boldsymbol{w} \in \mathbb{R}^g$ of weights into $k$ bits.

| Method | Composition* | Quantized Results | Bits | Reconstruction |
|---|---|---|---|---|
| Alternating | - | $\boldsymbol{B}, \boldsymbol{\alpha}_{(k)}, z_{B,(k)}$ | $gk + 16(k+1)$ | $\mathcal{R}_B(\boldsymbol{B}; \Theta_B)$ |
| FLEXBCQ | - | $\boldsymbol{B}, \boldsymbol{\alpha}_{(k)}, z_{B,(k)}, \Delta_{(k)}, z_{U,(k)}$ | $gk + 16(k+3)$ | $\mathcal{D}(\mathcal{R}_B(\boldsymbol{B}; \Theta_B); \Theta_U)$ |
| FLEXBCQ | ✓ | $\boldsymbol{B}, \boldsymbol{\alpha}_{(k)}^*, z_{B,(k)}^*$ | $gk + 16(k+1)$ | $\mathcal{R}_B^*(\boldsymbol{B}; \Theta_B)$ |

\* Whether apply Composition Theorem or not.

$\boldsymbol{B}$ is a binary code matrix containing binary weights codes in $\boldsymbol{w}$. $\boldsymbol{\alpha}_{(k)}^*$ and $z_{B,(k)}^*$ are scale factors and a shifting factor of the integrated BCQ Reconstruction function $\mathcal{R}_B^*$. the definitions of other symbols are summarized in Table 5. If we do not apply Composition Theorem, FLEXBCQ requires additional quantization parameters $\Delta_{(k)}$ and $z_{U,(k)}$ since it utilizes flexible transformation function $\mathcal{T}_\mathcal{F}$, requiring additional 32 bits per weight group. Moreover, FLEXBCQ requires a dequantization process whenever it performs inference and this slows down the inference speed of quantized models. On the other hand, if we apply Composition Theorem, the memory and latency overheads are

removed since it integrates the inefficient two-step reconstruction process of FLEXBCQ into a single BCQ's Reconstruction function $\mathcal{R}_B^*$. Therefore, Composition Theorem is essential for deploying quantized models generated by FLEXBCQ.

## F    ADDITIONAL ANALYSIS OF FLEXBCQ

In this section, we analyze the running time and memory usage of FLEXBCQ. We also analyze the patterns of flexible mapping when we quantize LLMs using FLEXBCQ.

### F.1    RUNNING TIME AND MEMORY USAGE OF FLEXBCQ

Table 15: Running times (s) of FlexRound and FLEXBCQ for quantizing Llama-3 8B into 2, 3 or 4 bits.

| Method | 4bit | 3bit | 2bit |
|--------|------|------|------|
| FlexRound | 16526 | 16523 | 16521 |
| FLEXBCQ | 36505 | 27306 | 25412 |

Table 16: Average number of bits per weight of FLEXBCQ and FlexRound. Column names are bits for saving a weight.

| Method | Scheme | 4bit | 3bit | 2bit |
|--------|--------|------|------|------|
| FlexRound | UQ | 4.25 | 3.25 | 2.25 |
| FLEXBCQ | BCQ | 4.63 | 3.50 | 2.38 |

In LLM quantization, the feasible running time and the size of the model after quantization are important factors for practical usage. In this section, we evaluate the running time and memory usage of FLEXBCQ by comparing it with FlexRound, which achieved the second-best performance in our experiments. Table 15 summarizes the comparison of running time for quantizing Llama-3 8B, and Table 16 shows the average number of bits required to store weights. In Table 16, the column headers indicate the bits needed to store a single weight without quantization parameters, and the values include the bits required for quantization parameters, providing the average number of bits needed per weight.

FLEXBCQ requires longer quantization times and increases the model size by approximately 8%. However, note that the quantization time is only incurred during the initial process and does not impact subsequent usage. Therefore, FLEXBCQ's running time is still feasible for practical usage. Moreover, we prioritize the acceleration capabilities of quantized models over memory size. Since both UQ and BCQ schemes achieve the same acceleration on state-of-the-art hardware such as LUT-GEMM, the 8% increase in memory usage is considered acceptable. For memory-constrained applications, channel-wise quantization, which minimizes memory overhead from quantization parameters become negligible, is a viable alternative. In this setting, FLEXBCQ significantly outperforms other methods by a large margin as presented in Table 22.

In conclusion, considering the outperforming performance demonstrated across various experiments in the main text and Appendix, the running time and memory usage of FLEXBCQ are both reasonable.

### F.2    PATTERNS OF FLEXIBLE MAPPING

The main idea of FLEXBCQ is to transfer the flexible mapping technique from FlexRound, a state-of-the-art UQ algorithm, to the BCQ. We examine the proportion of flexible mapping applied across all model weights to assess the effectiveness of this transfer. Tables 17, 18 and 19 illustrate the proportion of the flexibly mapped weights in the entire model, by block location, and by module type, respectively. The column names in the tables indicate the amount of changes in indices of weights' mapped quantization levels resulting from flexible mapping. We use a 3-bit quantized Llama-3 8B model for the experiment.

We recognize patterns from P1 to P4 as follows:

**P1.** Across the entire model, FLEXBCQ demonstrates a similar level of flexible mapping as FlexRound.

Table 17: Proportion of weights exhibiting flexible mapping across the entire model. Each column indicates the index change in the quantization level resulting from flexible mapping.

| Method | 0 | 1 | 2 | >2 |
|---|---|---|---|---|
| FlexRound | 95.4132 | 4.5862 | 5.00E-04 | 1.30E-05 |
| FlexBCQ | 96.0395 | 3.8619 | 9.85E-02 | 7.62E-05 |

Table 18: Proportion of weights exhibiting flexible mapping across different block positions. Each column corresponds to the index change in the quantization level resulting from flexible mapping. Bold values indicate the highest proportion within each column.

| Block Location | FlexBCQ | | | | FlexRound | | | |
|---|---|---|---|---|---|---|---|---|
| | 0 | 1 | 2 | >2 | 0 | 1 | 3 | >2 |
| Lower | **98.04** | 1.89 | 6.35E-02 | 9.44E-05 | **97.34** | 2.66 | 1.05E-03 | **4.43E-05** |
| Mid-Lower | 96.35 | 3.55 | 1.02E-01 | 1.75E-05 | 95.72 | 4.28 | 4.28E-05 | 6.25E-07 |
| Mid-Upper | 95.66 | 4.24 | 9.60E-02 | 2.24E-05 | 95.12 | 4.88 | 6.83E-05 | 8.75E-07 |
| Upper | 94.11 | **5.76** | **1.33E-01** | **1.70E-04** | 93.47 | **6.53** | **9.99E-04** | 5.75E-06 |

**P2.** The proportion of weights undergoing flexible mapping across the model is approximately 4%. This suggests that flexible mapping does not occur universally but is selectively applied to specific weights where necessary.

**P3.** In both methods, flexible mapping tends to occur more frequently in the upper layers of the model compared to the lower layers.

**P4.** In both methods, flexible mapping is generally more prevalent in MHA modules than in MLP modules for both methods.

From P1, it is clear that FLEXBCQ effectively utilizes flexible mapping across the entire model as we intended. P2, however, shows that most weights do not undergo flexible mapping, which aligns with the fact that mapping to the nearest quantization level typically minimizes quantization error. This indicates that the model leverages flexible mapping to reduce reconstruction error in specific cases. The trends observed in P3 and P4, by model location and module type, are similar to the patterns seen in modules that are effectively pruned by the module pruning algorithm (Zhong et al., 2024). When comparing these findings with P2, we interpret that more cautious flexible mapping occurs in the lower layers and MLP modules, which have a greater impact on model performance. In conclusion, we confirm that FLEXBCQ successfully utilizes flexible mapping as intended and our objective of transferring the optimization techniques of UQ algorithms to BCQ has been successfully achieved.

Table 19: Proportion of weights exhibiting flexible mapping across different module types. Each column corresponds to the index change in the quantization level resulting from flexible mapping. Bold values indicate the highest proportion within each column.

| Module Type | FlexBCQ | | | | FlexRound | | | |
|---|---|---|---|---|---|---|---|---|
| | 0 | 1 | 2 | >2 | 0 | 1 | 3 | >2 |
| MHA | 95.26 | **4.61** | **1.27E-01** | **1.31E-04** | 94.69 | **5.31** | 5.00E-04 | **1.30E-05** |
| MLP | **96.22** | 3.68 | 9.18E-02 | 6.30E-05 | **95.59** | 4.41 | **5.49E-04** | 1.30E-05 |

## G ADDITIONAL EXPERIMENTS

In the main text, we demonstrated the superiority of FLEXBCQ by comparing the accuracy of 3-bit and 4-bit quantized models on MMLU and GSM8K benchmarks under a weight group size of 128. In this section, we extend our analysis to additional experimental settings. All experiments are conducted using the Llama-3 8B. We introduce GPTQ and AWQ as additional competitors for rigorous comparison.

**Perplexity benchmarks.** The main text evaluates quantized LLM's knowledge using MMLU benchmarks, which assess general knowledge, and GSM8K, which focuses on mathematical reasoning. In this section, we further evaluate the quantized models' ability to generate texts by comparing the perplexity of quantized models on the WikiText-2 (Merity et al., 2017) and C4 (Raffel et al., 2020) benchmarks. The results are presented in the Table 20.

Table 20: Comparison of perplexities of quantized models on WikiText-2 and C4 benchmakrs. Lower perplexity indicates better performance.

| Scheme | Method | 4-bit | | 3-bit | |
|---|---|---|---|---|---|
| | | Wiki | C4 | Wiki | C4 |
| - | Full precision | 6.14 | 8.89 | 6.14 | 8.89 |
| UQ | RTN | 6.75 | 9.67 | 10.82 | 14.85 |
| | AWQ | 6.54 | 9.40 | 8.23 | 11.59 |
| | GPTQ | 6.54 | 9.36 | 9.05 | 11.70 |
| | OmniQuant | 6.69 | 9.59 | 8.82 | 12.36 |
| | FlexRound | 6.55 | 9.36 | 7.62 | 10.76 |
| BCQ | Greedy | 6.22E+04 | 2.45E+04 | 8.32E+04 | 3.81E+04 |
| | Alternating | 7.70 | 10.88 | 869.89 | 978.83 |
| | FlexBCQ | **6.46** | **9.24** | **7.42** | **10.46** |

The results demonstrate that FLEXBCQ achieves the highest performance (lowest perplexities) across all bit widths and benchmark types. This confirms that FLEXBCQ not only excels in general and task-specific knowledge retention but also outperforms in sentence generation tasks.

**2-bit quantization** While the main text focuses on relatively high bit widths (3-bit and 4-bit), this experiment evaluates the robustness of FLEXBCQ under an extremely low bit width of 2 bits. The results are summarized in Table 21.

The results show that all algorithms, except FLEXBCQ, fail to function effectively under the extreme quantization setting. On MMLU benchmark, other methods achieve accuracy near or below the random guess rate of 25% which randomly selects an answer among four choices. On GSM8K benchmark, all competitors achieve less than 1% accuracy, with perplexity values reaching tens to millions. In contrast, FLEXBCQ succeeds in retaining pretrained models' knowledge under low bit widths, outperforming the second-best competitor by 10.27%, 7.48%, and 22.14% in MMLU 5-shot, MMLU 0-shot, and GSM8K, respectively. These findings highlight FLEXBCQ 's superior performance in extremely low-bit scenarios, achieved by our main ideas.

**Channel-wise quantization** In quantization algorithms, such as UQ and BCQ, weights in the same weight group share quantization parameters. As the group size increases, quantization becomes more challenging since more weights share the same quantization parameters. We use a small group size

Table 21: Accuracies and perplexities of 2-bit quantized Llama-3 8B models using FLEXBCQ and competitors. Higher accuracy and lower perplexities represent better performance.

| Scheme | Method | Accuracy (↑) | | | Perplexity (↓) | |
| | | MMLU 5-shot | MMLU 0-shot | GSM8K | Wiki | C4 |
| --- | --- | --- | --- | --- | --- | --- |
| - | Full precision | 65.02 | 60.52 | 76.12 | 6.14 | 8.89 |
| UQ | RTN | 22.95 | 22.95 | 0.00 | 1.97E+06 | 2.44E+06 |
| | AWQ | 22.95 | 22.95 | 0.00 | 1.72E+06 | 2.13E+06 |
| | GPTQ | 25.32 | 24.43 | 0.40 | 450.59 | 254.37 |
| | OmniQuant | 25.04 | 22.91 | 0.00 | 987.10 | 1395.17 |
| | FlexRound | 24.27 | 24.97 | 0.23 | 68.54 | 66.57 |
| BCQ | Greedy | 26.76 | 26.89 | 0.00 | 4.96E+05 | 6.85E+05 |
| | Alternating | 23.59 | 22.97 | 0.00 | 7.34E+06 | 3.97E+06 |
| | FlexBCQ | **37.02** | **34.37** | **22.54** | **14.95** | **16.55** |

of 128 to minimize accuracy degradation in the main text, this experiment evaluates performance when the group size is expanded to the size of each weight matrix's input channel; we perform channel-wise quantization. The results are summarized in Table 22.

Table 22: Accuracies of Llama-3 models compressed with channel-wise quantization using FLEXBCQ and competitors.

| Scheme | Method | 4bit | | | 3bit | | |
| | | 5-shot | 0-shot | Average | 5-shot | 0-shot | Average |
| --- | --- | --- | --- | --- | --- | --- | --- |
| - | Full precision | 65.02 | 60.52 | 62.77 | 65.02 | 60.52 | 62.77 |
| UQ | RTN | 57.02 | 52.31 | 54.67 | 25.39 | 23.15 | 24.27 |
| | AWQ | 61.67 | 55.61 | 58.64 | 43.86 | 37.96 | 40.91 |
| | GPTQ | 41.62 | 41.94 | 41.78 | 25.92 | 25.98 | 25.95 |
| | OmniQuant | 58.11 | 53.86 | 55.99 | 25.35 | 28.02 | 26.69 |
| | FlexRound | 60.84 | 53.21 | 57.02 | 54.52 | 45.92 | 50.22 |
| BCQ | Greedy | 24.19 | 23.00 | 23.59 | 24.65 | 26.44 | 25.55 |
| | Alternating | 22.95 | 22.95 | 22.95 | 25.51 | 25.51 | 25.51 |
| | FlexBCQ | **62.26** | **57.01** | **59.63** | **55.73** | **51.01** | **53.37** |

As a result, all algorithms except FLEXBCQ exhibit significant performance degradation, especially in 3-bit quantization. FLEXBCQ shows 0.99% and 3.15% higher accuracy compared to the second-best algorithm in 4-bit and 3-bit quantization, respectively.

Across all experiments, FLEXBCQ demonstrates the best performance. Whether generating sentences, operating under extremely low bit widths, or managing large group sizes, FLEXBCQ consistently excels. These results highlight its capability to tackle challenging problems where other methods fail.

