# OpenReview forum: "FlexBCQ: Flexible Binary-coding Quantization for Large Language Models"
_ICLR.cc/2025/Conference — Submitted to ICLR 2025_

### Official Review · Reviewer_5RAW · 2024-11-02

**Soundness:** 2
**Presentation:** 3
**Contribution:** 2
**Rating:** 6
**Confidence:** 4

**Summary:**

This work proposes an approach to improve the performance of binary-coding quantization (BCQ) via flexible mapping of quantization levels and dedicated optimization procedure. The introduced approach is evaluated on several modern language models at 3 and 4 bit compression targets.

**Strengths:**

* FlexBCQ manages to significantly improve the performance of BCQ and becomes competitive with state-of-the-art algorithms for uniform quantization.

* This work identifies the issues with prior approaches to BCQ for LLM quantization and proposes nontrivial, yet sensible solutions, to stabilize the fine-tuning of BCQ parameters.

* The presentation quality is good. The main points, challenges and experimental results are stated clearly.

**Weaknesses:**

* Something is wrong with Mistral 7B scores. According to the table, all methods show significant performance drop relative to fp16 baselines, whereas for other models underconsideration 4 bit models are close to the original model. In addition, Mistral 7B should have inferior performance to Llama 3 8B.

**Minor**

I3 (line 224) Typo Blokwise -> Blockwise

**Questions:**

* The paper claims that GPTQ and AWQ yield significantly worse performance than OmniQuant. However, in my personal experience, GPTQ and AWQ appear to be pretty robust at 4 bit quantization and typically lose only 1-2% relative to the original model on MMLU and GSM8k. I would suggest checking experimental setup for these methods.

* MMLU and GSM8k are quite representative benchmarks. However, for a comparison with prior art, I would suggest reporting Wikitext2/C4 perplexity for quantized models (or a subset of 0-shot benchmarks from lm-eval-harness).

* What is the best value for periodic remapping? In the main experimental section, p=2 is adopted for all models except Llama-3 70B. Does even less frequent remapping help? It would be good to have a plot performance vs p for the sake of completeness.

* How was the duration of the optimization procedure chosen? 20 epochs could be excessive, as the amount of data is small and the model may start to overfit (the number of trainable parameters is small either). I would suggest adding an ablation on this.
What is the runtime of the method for the models considered?

* It would be insightful to present aggregated statistics over different weights of the value $|I(w) - I(w_f)|$ across model - i.e percentage of weights for each value.

---

> ### Author Response · Authors · 2024-11-23
>
> We sincerely appreciate your thoughtful suggestions regarding the experiments in our paper. In response, we have conducted all the proposed experiments and we are in the process of integrating the results into the revised manuscript. The updated version, including these findings, will be updated during the discussion period, and we will provide a summary of the changes upon its release.
>
> > **[R5-1]** Typos
>
> Thank you for finding typos in our paper. we revised the typos and the revised version of the paper will be uploaded soon.
> * **(Table 1)** We modify the benchmark accuracies of the fp16 baseline of Mistral 7B as follows: MMLU 5-shot: 70.48$\rightarrow$62.57  MMLU 0-shot 68.17 $\rightarrow$ 60.22. We double-checked the scores of Mistral 7B, and there are no further typos.
> * **(line 224)** Blokwise $\rightarrow$ Blockwise. We fix the typo.
>
>
> > **[R5-2]** The paper claims that GPTQ and AWQ yield significantly worse performance than OmniQuant. However, in my personal experience, GPTQ and AWQ appear to be pretty robust at 4 bit quantization and typically lose only 1-2% relative to the original model on MMLU and GSM8k. I would suggest checking experimental setup for these methods.
>
> Thank you for your valuable suggestion. In our initial experiments, we excluded GPTQ and AWQ due to their significantly lower performance compared to OmniQuant on the W2A16 and W3A16 settings (Table 1 in OmniQuant [1]). However, in light of your feedback, we have conducted additional experiments incorporating GPTQ and AWQ. These experiments were performed on the Llama-3 8B model, and the results are presented in Table R5-1.
>
> **Table R5-1.** Perplexities on WikiText-2 and C4 benchmarks of 2 to 4bit Llama-3 8B models quantized using FlexBCQ and competitors. Bold values represent the highest accuracies.
> | Scheme |       Method       | 4bit, 5-shot | 4bit, 0-shot | 3bit, 5-shot | 3bit, 0-shot | 2bit, 5-shot | 2bit, 0-shot |
> |:------:|:------------------:|:------------:|:------------:|:------------:|:------------:|:------------:|:------------:|
> |   -    |      Baseline      |     65.02    |     60.52    |     65.02    |     60.52    |     65.02    |     60.52    |
> |   UQ   |        RTN         |     62.11    |     57.79    |     38.12    |     39.50    |     22.95    |     22.95    |
> |   UQ   |        AWQ         |     62.82    |   **59.05**  |     55.36    |     51.04    |     22.95    |     22.95    |
> |   UQ   |       GPTQ         |     61.96    |     57.67    |     34.30    |     40.94    |     25.32    |     24.43    |
> |   UQ   |    OmniQuant       |     63.17    |     58.99    |     51.56    |     45.06    |     25.04    |     22.91    |
> |   UQ   |     FlexRound      |     63.25    |     58.32    |     58.89    |   **54.65**  |     24.27    |     24.97    |
> |   BCQ  |       Greedy       |     24.90    |     24.51    |     27.15    |     26.84    |     26.76    |     26.89    |
> |   BCQ  |    Alternating     |     56.27    |     53.10    |     25.85    |     23.10    |     23.59    |     22.97    |
> |  **BCQ** |   **FlexBCQ**    |   **63.81**  |   58.56  |   **59.33**  |   54.49  |   **37.02**  |   **34.37**  |
>
> The experimental results confirm that AWQ and GPTQ demonstrate respectable performance. However, in all experiments except for the 4-bit 0-shot scenario, FlexBCQ outperforms both algorithms. Notably, for 2-bit quantization, FlexBCQ achieves a performance improvement of at least 9.94% compared to both AWQ and GPTQ. These findings will be integrated into the paper to enable a more thorough and precise comparison.

---

> ### Author Response · Authors · 2024-11-23
>
> > **[R5-3]**  MMLU and GSM8k are quite representative benchmarks. However, for a comparison with prior art, I would suggest reporting Wikitext2/C4 perplexity for quantized models (or a subset of 0-shot benchmarks from lm-eval-harness).
>
> To further validate the performance of FlexBCQ across diverse scenarios, we compare the perplexity (PPL) of quantized models on the WikiText-2 and C4 benchmarks. These experiments were conducted using the Llama-3 8B model, and the results are presented in Table R5-2.
>
> **Table R5-2.** Perplexities on WikiText-2 and C4 benchmarks of 2 to 4bit Llama-3 8B models quantized using FlexBCQ and competitors.
>
> | Scheme |        Method        | 4bit, Wiki | 4bit, C4 | 3bit, Wiki | 3bit, C4 | 2bit, Wiki | 2bit, C4 |
> |:------:|:--------------------:|:----------:|:--------:|:----------:|:--------:|:----------:|:--------:|
> |   -    |    Full precision    |    6.14    |   8.89   |    6.14    |   8.89   |    6.14    |   8.89   |
> |   UQ   |         RTN          |    6.75    |   9.67   |   10.82    |  14.85   | 1.97E+06   | 2.44E+06 |
> |   UQ   |         AWQ          |    6.54    |   9.40   |    8.23    |  11.59   | 1.72E+06   | 2.13E+06 |
> |   UQ   |        GPTQ          |    6.54    |   9.36   |    9.05    |  11.70   |   450.59   |  254.37  |
> |   UQ   |      OmniQuant       |    6.69    |   9.59   |    8.82    |  12.36   |   987.10   | 1395.17  |
> |   UQ   |      FlexRound       |    6.55    |   9.36   |    7.62    |  10.76   |    68.54   |  66.57   |
> |   BCQ  |        Greedy        | 6.22E+04   | 2.45E+04 | 8.32E+04   | 3.81E+04 | 4.96E+05   | 6.85E+05 |
> |   BCQ  |     Alternating      |    7.70    |  10.88   |   869.89   |  978.83  | 7.34E+06   | 3.97E+06 |
> | **BCQ** |    **FlexBCQ**      |  **6.46**  | **9.24** |  **7.42**  | **10.46**| **14.95**  | **16.55**|
>
>
> Experimental results indicate that FlexBCQ consistently achieves the lowest perplexity (demonstrating the best performance) across all experiments compared to its competitors. In particular, for 2-bit quantization, all competing methods exhibit a perplexity of 50 or higher, reflecting substantial performance degradation, whereas FlexBCQ maintains a significantly lower perplexity of approximately 15. These results will be integrated into the paper to enable a more comprehensive and rigorous comparison.

---

> ### Author Response · Authors · 2024-11-23
>
> > **[R5-4]** What is the best value for periodic remapping? In the main experimental section, p=2 is adopted for all models except Llama-3 70B. Does even less frequent remapping help? It would be good to have a plot performance vs p for the sake of completeness.
>
> The periodic remapping technique is a technique that performs weight remapping to updated quantization levels not at every step during Blockwise Output Reconstruction, but at intervals of $p$ steps. Table R5-3 presents the performance variations of FlexBCQ across different remapping periods.
>
> **Table R5-3.** MMLU benchmark accuracies of 3-bit Llama-3 8B models quantized by FlexBCQ with different remapping period $p$.
> |   p   | MMLU 5-shot | MMLU 0-shot |  Avg.   |
> |:-----:|:-----------:|:-----------:|:-------:|
> |   1   |    58.87    |    53.98    |  56.42  |
> |   2   |    59.33    |    54.49    |  56.91  |
> |   4   |    59.05    |    52.91    |  55.98  |
> |   8   |    59.23    |    55.13    |  57.18  |
> |  16   |    59.12    |    54.23    |  56.68  |
>
> The experimental results indicate that, with the exception of $p=4$, most settings achieve comparable accuracy, and $p=2$, as employed in the paper, demonstrates strong performance. Interestingly, we observe the highest average accuracy when we use less frequent settings such as $p=8$.
>
> We recommend using $p=2$, as it was utilized to achieve the outstanding results presented in the paper for all cases except the Llama-3 70B model, where $p=1$ is used. Slight performance gains may be achieved by tuning $p$ to suit specific scenarios. We will include the plot regarding hyperparameter tuning over $p$ in the revised paper.
>
> > **[R5-5]** How was the duration of the optimization procedure chosen? 20 epochs could be excessive, as the amount of data is small and the model may start to overfit (the number of trainable parameters is small either). I would suggest adding an ablation on this. What is the runtime of the method for the models considered?
>
> **[Epochs]**
>
> In our experimental setup, we adopt OmniQuant’s SGD-based optimization framework, similar to FlexBCQ, and set the number of epochs to 20. The table below illustrates the impact of varying the number of epochs on FlexBCQ’s performance:
>
> **Table R5-4.** Accuracy variation on the MMLU benchmark for the 3-bit quantized Llama-3 8B model using FlexBCQ across different epoch settings.
> |   MMLU   |   10    |   20    |   30    |
> |:--------:|:-------:|:-------:|:-------:|
> |  5-shot  |  59.17  |  59.33  |  59.23  |
> |  0-shot  |  54.28  |  54.49  |  53.65  |
> | Average  |  56.72  |  56.91  |  56.44  |
>
> The experimental results demonstrate that the 20-epoch setting used in our experiments achieves the highest average accuracy. Overfitting appears to occur at approximately 30 epochs, further validating that training with 20 epochs is a proper choice.
>
> **[Running Time]**
>
> Before explaining about running time of FlexBCQ, we clarify some main points. Our study primarily focuses on achieving superior accuracy at a given bit-width. For LLMs, the quantization process is performed only once, and the model can be used repeatedly thereafter. Consequently, as long as the quantization process is not prohibitively expensive, we consider a model with higher accuracy to be superior.
>
> We compare the time required to quantize the Llama-3 8B model to 2, 3, and 4 bits using FlexBCQ and its main competitor, FlexRound. The comparison results are summarized in Table R5-5.
>
> **Table R5-5.** Running times in seconds for quantizing Llama-3 8B models into 2 to 4bits using FlexRound and FlexBCQ.
> |   Method    |   4-bit   |   3-bit   |   2-bit   |
> |:-----------:|:---------:|:---------:|:---------:|
> | FlexRound   |   16,526   |   16,523   |   16,523   |
> | FlexBCQ     |   36,505   |   27,306   |   25,412   |
>
> FlexBCQ employs non-uniform quantization levels, requiring weights to be mapped by identifying the nearest quantization level, rather than using the rounding function typically used in uniform quantization. This process of finding the nearest quantization level is more time-consuming than rounding, resulting in longer running times. Specifically, the GPU-unfriendly BST-based algorithm proposed by Xu et al. [1] (Algorithm 1 in Xu et al. [1]) leads to execution times that are practically infeasible to large language models.
>
> To mitigate this issue, we replace the BST-based approach with a GPU-friendly algorithm that computes the distances between each weight and all quantization levels simultaneously to identify the nearest quantization level. This replacement significantly reduces the running time while preserving FlexBCQ’s superior performance. Further exploration of methods to optimize running time without compromising performance is a promising direction for future research, which we plan to address in subsequent studies. We will include a discussion about the running time of FlexBCQ in the paper during the discussion period.

---

> ### Author Response · Authors · 2024-11-23
>
> > **[R5-6]** It would be insightful to present aggregated statistics over different weights of the value $|I(w)-I(w_f)|$ across model - i.e percentage of weights for each value.
>
> Thank you for suggesting this intriguing analytical topoc. To quantify the occurrence of flexible mapping, we analyze the percentage of flexibly mapped weights in (1) the entire model, (2) different block locations, and (3) various module types. The corresponding results are summarized in Tables R5-6, R5-7, and R5-8, respectively. Furthermore, to compare the mapping patterns of FlexBCQ and FlexRound, we compute statistics for both methods.
>
> **Table R5-6.** Proportion of weights exhibiting flexible mapping across the entire model. Each column indicates the index change in the quantization level resulting from flexible mapping.
>
> |   Method    |     0      |     1      |      2       |     >2      |
> |:-----------:|:----------:|:----------:|:------------:|:-----------:|
> | FlexRound   |   95.4132  |   4.5862   |   5.00E-04   |   1.30E-05  |
> | FlexBCQ     |   96.0395  |   3.8619   |   9.85E-02   |   7.62E-05  |
>
>
> **Table R5-7.** Proportion of weights exhibiting flexible mapping across different block positions. Each column corresponds to the index change in the quantization level resulting from flexible mapping. Bold values indicate the highest proportion within each column.
> | Block Location | FlexBCQ 0 | FlexBCQ 1 | FlexBCQ 2 | FlexBCQ >2 | FlexRound 0 | FlexRound 1 | FlexRound 2 | FlexRound >2 |
> |:--------------:|:---------:|:---------:|:---------:|:----------:|:-----------:|:-----------:|:-----------:|:------------:|
> | Lower          |  **98.04** |   1.89    |  6.35E-02  |  9.44E-05  |   **97.34**  |    2.66     |   1.05E-03  |   **4.43E-05** |
> | Mid-Lower      |   96.35    |   3.55    |  1.02E-01  |  1.75E-05  |    95.72     |    4.28     |   4.28E-05  |   6.25E-07    |
> | Mid-Upper      |   95.66    |   4.24    |  9.60E-02  |  2.24E-05  |    95.12     |    4.88     |   6.83E-05  |   8.75E-07    |
> | Upper          |   94.11    |  **5.76**  |  **1.33E-01**  |  **1.70E-04**  |    93.47     |   **6.53**  |   **9.99E-04**  |   5.75E-06    |
>
>
> **Table R5-8.** Proportion of weights exhibiting flexible mapping across different module types. Each column corresponds to the index change in the quantization level resulting from flexible mapping. Bold values indicate the highest proportion within each column.
> | Module Type | FlexBCQ 0 | FlexBCQ 1 | FlexBCQ 2 | FlexBCQ >2 | FlexRound 0 | FlexRound 1 | FlexRound 2 | FlexRound >2 |
> |:--------:|:---------:|:---------:|:---------:|:----------:|:-----------:|:-----------:|:-----------:|:------------:|
> |   MHA    |   95.26   |  **4.61** | **1.27E-01** | **1.31E-04** |    94.69     |   **5.31**   |   5.00E-04  |   **1.30E-05** |
> |   MLP    | **96.22** |   3.68    |   9.18E-02   |   6.30E-05   |  **95.59**   |    4.41     |   **5.49E-04**  |   1.30E-05 |
>
> (Continues in the next response)

---

> > ### Author Response · Authors · 2024-11-23
> >
> > The experimental results reveal the following patterns related to flexible mapping:
> >
> > **[Pattern 1]** Across the entire model, FlexBCQ demonstrates a similar level of flexible mapping as FlexRound.
> >
> > **[Pattern 2]** The proportion of weights undergoing flexible mapping across the model is approximately 4%. This suggests that flexible mapping does not occur universally but is selectively applied to specific weights where necessary.
> >
> > **[Pattern 3]** In both methods, flexible mapping tends to occur more frequently in the upper layers of the model compared to the lower layers.
> >
> > **[Pattern 4]** Similarly, flexible mapping is generally more prevalent in MHA modules than in MLP modules for both methods.
> >
> > We interpret these results as follows:
> >
> > **[Pattern 1]**
> >
> > FlexBCQ effectively utilizes the flexible mapping optimization technique from UQ as intended.
> >
> > **[Pattern 2]**
> >
> > Flexible mapping is a technique that maps weights to a relatively distant quantization level instead of the nearest one. Mapping weights to the nearest quantization level minimizes quantization error. However, when considering input distribution, dependencies among weights arise, leading to the need for flexible mapping in specific cases. This indicates that flexible mapping does not occur universally but is triggered under particular conditions that are induced by input-driven dependencies.
> >
> > **[Patterns 3-4]**
> >
> > The tendency for flexible mapping to occur more in the upper layers than in the lower layers, and more in MHA modules than in MLP modules, aligns intriguingly with patterns observed in module-pruning studies [2], where pruning is more effective in similar regions. Considering these results alongside the analysis of Pattern 2, we hypothesize that in the lower layers and MLP modules, flexible mapping is more selectively applied to ensure it is only used where absolutely necessary to maintain model performance. While the exact reason for the alignment of pruning and flexible mapping patterns remains unclear, this represents an interesting research topic.
> >
> > By collecting and analyzing various statistics on flexible mapping, we identified these four interesting patterns. We are grateful to the reviewer for proposing this fascinating analysis. We will incorporate the findings and related discussions into the revised version of the paper currently under preparation.
> >
> > **[References]**
> >
> > [1] Shao, Wenqi, et al. "Omniquant: Omnidirectionally calibrated quantization for large language models." arXiv preprint arXiv:2308.13137 (2023).
> >
> > [2] Zhong, Longguang, et al. "BlockPruner: Fine-grained Pruning for Large Language Models." arXiv preprint arXiv:2406.10594 (2024).

---

> > > ### Comment · Reviewer_5RAW · 2024-11-24
> > > **Response to rebuttal**
> > >
> > > Thanks for your response. I believe that new results significantly strengthen the paper. Therefore I have increased my score.

---

> > > > ### Author Response · Authors · 2024-11-28
> > > >
> > > > Dear Reviewer 5RAW,
> > > >
> > > > We are glad to hear that all your concerns have been addressed! In particular, we sincerely appreciate your positive evaluation based on the additional experiments of our paper. The additional experimental results, aside from those provided in our previous response, have been included in Global Response [G3] and the Appendix of the paper. We would be grateful if you could check these as well.
> > > >
> > > > Furthermore, as a result of our constructive discussion, we have incorporated the following updates into the paper:
> > > > * **[R5-1] Typos:** We revise the typos in line 211 and Table 1.
> > > > * **[R5-2] Additional competitors:** We include GPTQ and AWQ in all experiments in Section G in Appendix.
> > > > * **[R5-3] Perplexity:** We include the experimental results on perplexity benchmarks (WikiText-2 and C4) in Section G in Appendix. We further include the experimental results of 2-bit quantization and channel-wise quantization in the same section. FlexBCQ shows the best performance in every setting.
> > > > * **[R5-4] Selecting remapping period:** We provide a guideline for selecting hyperparameters of FlexBCQ not only the remapping period but also, gradient filtering threshold, sample dataset, sample size, epochs, and clipping strategies with experimental results in Section C in Appendix.
> > > > * **[R5-5] Number of epochs and running time:** We include the experimental results regarding performance variations of FlexBCQ over number of epochs in Section C.5 in Appendix. We include the discussion about running time of FlexBCQ in Section F.1 in Appendix.
> > > > * **[R5-6] Analyzing flexible mapping** We include the contents we discussed regarding flexible mapping in Section F.2 in Appendix.
> > > >
> > > > This concludes the actual updates reflected in the paper. We kindly ask you to confirm whether the points from our discussion have been properly incorporated, and if you have any additional content you would like to include, please feel free to let us know. We appreciate again for your constructive feedbacks and suggesting crucial experiments.

---

> > > > > ### Comment · Reviewer_5RAW · 2024-11-28
> > > > >
> > > > > Thanks for your response. While the results above show that FlexBCQ is a reasonable approach with good performance, for a higher score I would require a more fundamental contribution with novel insights into the problem of model compression or incorporation of previously unexplored ideas. Therefore, I'll keep my score as borderline accept.

---

### Official Review · Reviewer_mGfi · 2024-11-04

**Soundness:** 2
**Presentation:** 2
**Contribution:** 2
**Rating:** 3
**Confidence:** 3

**Summary:**

This paper introduces FLEXBCQ, a method that integrates FlexRound (UQ) and Binary-Coding Quantization (BCQ). The authors demonstrate promising performance enhancements with FLEXBCQ on the MMLU 5-shot and 0-shot tasks.

**Strengths:**

1. This paper provides a foundational overview of the UQ and BCQ methods. Although some sections could benefit from clearer explanations, the background information is useful for readers in general.

2. The integration of UQ and BCQ is clearly presented in Figure 3, offering a straightforward conceptual outline that supports the paper’s logic.

**Weaknesses:**

1. Figures 1 and 2 fail to clarify the logic and pipeline of FlexBCQ. A more abstract diagram that highlights the main innovations of FlexBCQ and outlines its functionality would be beneficial. Unfortunately, Figures 1 and 2 are quite confusing as currently presented.

2. The main purpose of Theorem 1 (Composition Theorem) is unclear. The theorem lacks subsequent analysis or illustration, making it difficult to understand its application to BCQ or FlexBCQ. The section appears to discuss results related to FlexBCQ, but the text refers to BCQ. Further clarification on the detransformation function and analysis of these results is needed to grasp their significance.

3. The empirical results presented are not compelling. According to Table 1, FlexBCQ shows only marginal improvements and, in some cases, underperforms compared to the simpler FlexRound.

4. The absence of code and insufficient implementation details in Appendix B raise concerns about reproducibility. Providing the code or more comprehensive details would significantly aid in evaluating the robustness and applicability of the proposed method.

**Questions:**

1. How does FlexBCQ perform under extreme quantization conditions, such as using only 2 bits or fewer? Is such a level of quantization feasible, and if so, what can be expected in terms of performance?

2. If I understand correctly from Figure 3, FlexBCQ combines BCQ and UQ. Could the authors clarify the key innovations behind this integration? It would be beneficial if the benefits could be explicitly demonstrated through theoretical measures, such as convergence rates or other relevant criteria. Additionally, the primary challenges of this combination are not entirely clear to me.

3. How do the running time and inference speed of FlexBCQ compare with those of baseline methods? I am particularly interested in whether this method is both time and memory efficient.

4. In addition to the 5-shot and 0-shot MMLU results, could the authors also provide perplexity scores to give a more comprehensive view of the model's performance?

---

> ### Author Response · Authors · 2024-11-23
>
> We sincerely thank you for carefully reviewing our paper and providing thoughtful comments. Your feedback has guided us in revising the text and figures, as well as conducting additional experiments, thereby improving the overall quality of our paper. Below, we address the comments you raised. Revisions related to writing and figures are currently in progress, and we will provide detailed information on these updates once they are finalized. All revisions will be completed within the discussion period.
>
> ----
>
> > **[R4-1]** Figures 1 and 2 fail to clarify the logic and pipeline of FlexBCQ. A more abstract diagram that highlights the main innovations of FlexBCQ and outlines its functionality would be beneficial. Unfortunately, Figures 1 and 2 are quite confusing as currently presented.
>
> As you suggested, we are revising Figures 1 and 2 to adopt a more abstract representation, aiming to convey their intended message to readers more intuitively. Furthermore, we are refining the captions to improve overall readability. We will provide detailed updates on these revisions once they are finalized.
>
> > **[R4-2]** The main purpose of Theorem 1 (Composition Theorem) is unclear. The theorem lacks subsequent analysis or illustration, making it difficult to understand its application to BCQ or FlexBCQ. The section appears to discuss results related to FlexBCQ, but the text refers to BCQ. Further clarification on the detransformation function and analysis of these results is needed to grasp their significance.
>
> Thank you for your comments regarding the Composition Theorem. The Composition Theorem is introduced to ensure that models quantized with FlexBCQ achieve the same inference speed and memory usage as the original BCQ without any additional overhead. To clearly illustrate this effect, we will compare the inference speed and memory usage with and without the application of the Composition Theorem.
>
> **[Background]**
>
> Before delving into the comparison, let me explain the storage and inference processes of UQ and BCQ:
>
> 1. **Uniform Quantization (UQ)**. As illustrated in Figure 3(a), each weight is stored as a $k$-bit integer $\tilde{w}$, along with quantization parameters such as UQ’s scale factor $\Delta\_{(k)}$ and zero-point $z\_{U,(k)}$, for reconstruction. During inference, the weights $\tilde{\boldsymbol{w}}$ stored as integers are detransformed using the quantization parameters before continuing with further inference.
>
> 2. **Binary-coding Quantization (BCQ)**. As illustrated in Figure 3(b), each weight is stored as a $k$-bit binary code $B$, and quantization parameters such as BCQ’s scale factors $\boldsymbol{\alpha}\_{(k)}$ and shifting factors $z_{B,(k)}$ are stored for reconstruction. During inference, these parameters are used to reconstruct the weights, after which the inference proceeds. (While the exact BCQ inference process may vary depending on the kernel, this description assumes the simplest method for explanatory purposes.)
>
> 3. **FlexBCQ**. As illustrated in Figure 3(c), FlexBCQ integrates UQ’s transformation process with BCQ’s mapping process. Each weight is stored as $k$-bit binary codes $b$, and quantization parameters include UQ’s scale factor $\Delta\_{(k)}$, zero-point  $z\_{U,(k)}$, BCQ’s scale factors $\boldsymbol{\alpha}\_{(k)}$, and shifting factors $z_{B,(k)}$. During inference, FlexBCQ reconstructs weights in two steps: (1) reconstruction using BCQ’s parameters, followed by the (2) UQ's detransformation process. This additional step increases both the number of quantization parameters stored and inference latency compared to traditional BCQ.
>
> **[The role of Composition Theorem]**
>
> To mitigate these drawbacks, **the Composition Theorem enables the combined reconstruction and detransformation processes of FlexBCQ to be streamlined into a single BCQ reconstruction process.** This eliminates additional memory overhead and ensures that FlexBCQ achieves the same inference speed as standard BCQ. Consequently, the Composition Theorem provides a fundamental theoretical foundation that allows FlexBCQ to deliver its performance improvements without introducing additional latency or memory costs.
>
> (Continued on the next response)

---

> ### Author Response · Authors · 2024-11-23
>
> We summarize the above comparisons in Table R4-1. Furthermore, we will incorporate this discussion into the paper and notify you of the updated sections once the revisions are complete.
>
> **Table R4-1.** Comparison of quantized results, the number of bits for saving, and the inference process of quantized models in UQ, BCQ, FlexBCQ, and FlexBCQ with Composition Theorem. $g$ is the group size and $k$ is the bit width.
> | Method                      | Quantized Results                                         | # of bits for saving       | Inference Process                              |
> |:---------------------------:|:-------------------------------------------------------:|:--------------------------:|:---------------------------------------------:|
> | UQ                          | $\tilde{w}$, $\Delta_{(k)}$, $z\_{U,(k)}$                 | gk + 32                   | UQ's Detransformation                         |
> | BCQ                         | $B$, $\boldsymbol{\alpha}\_{(k)}$, $z\_{B,(k)}$           | gk + 16(k+1)              | BCQ's Reconstruction                          |
> | FlexBCQ without Composition theorem | $B$, $\Delta_{(k)}$, $z\_{U,(k)}$, $\boldsymbol{\alpha}\_{(k)}$, $z_{B,(k)}$ | gk + 16(k+3)              | UQ's Detransformation after BCQ's Reconstruction |
> | FlexBCQ with Composition theorem  | $B$, $\boldsymbol{\alpha}\_{(k)}$, $z\_{B,(k)}$        | gk + 16(k+1)              | BCQ's Reconstruction                          |

---

> ### Author Response · Authors · 2024-11-23
>
> > **[R4-3]** Performance of FlexBCQ (2-bit quantization and perplexity performance)
>
> In our study, we rigorously evaluate the effectiveness of FlexBCQ using the general knowledge benchmark (MMLU) and the task-specific benchmark (GSM8K). The results demonstrate that FlexBCQ consistently outperforms other methods in most scenarios. Notably, FlexBCQ excels in challenging tasks, such as the 3-bit quantization experiments reported in Table 1 and the GSM8K benchmark in Table 2, which involves answering open-ended questions.
>
> To further substantiate the robustness of FlexBCQ, we conduct experiments on 2-bit quantization and perplexity (PPL) performance. These experiments were performed using the Llama-3 8B model.
>
> **[Extreme Quantization (2-bit)]**
>
> Tables R4-2 and R4-3 summarize the accuracy of 2-bit quantized Llama-3 8B models on MMLU and GSM8K benchmarks. As illustrated in the table, FlexBCQ exhibits significantly higher accuracy than competitors in both 5-shot and 0-shot, exhibiting 8.88%p higher average accuracy than the second-best algorithm. Furthermore, FlexBCQ is the only algorithm that shows an accuracy of over 1% among all competitors, exhibiting more than 22% higher accuracies than the second-best algorithm.
>
> **Table R4-2.** The accuracies on MMLU 5-shot and 0-shot benchmarks of 2-bit Llama-3 8B models quantized with FlexBCQ and competitors.
> | Scheme |       Method       | MMLU 5-shot | MMLU 0-shot | Average |
> |:------:|:------------------:|:-----------:|:-----------:|:-------:|
> |   UQ   |        AWQ         |    22.95    |    22.95    |  22.95  |
> |   UQ   |       GPTQ         |    25.32    |    24.43    |  24.88  |
> |   UQ   |     OmniQuant      |    25.04    |    22.91    |  23.97  |
> |   UQ   |     FlexRound      |    24.27    |    24.97    |  24.62  |
> |   UQ   |       Greedy       |    26.76    |    26.89    |  26.82  |
> |   BCQ  |    Alternating     |    23.59    |    22.97    |  23.28  |
> |   **BCQ**  |     **FlexBCQ**    |  **37.02**  |  **34.37**  | **35.70** |
>
>
> **Table R4-3.** The accuracies on GSM8K benchmark of 2-bit Llama-3 8B models quantized with FlexBCQ and competitors.
> | Scheme |        Method        | GSM8K |
> |:------:|:--------------------:|:----------:|
> |   -    |   Full precision     |    76.12   |
> |   UQ   |         RTN          |     0.00   |
> |   UQ   |         AWQ          |     0.00   |
> |   UQ   |        GPTQ          |     0.40   |
> |   UQ   |      OmniQuant       |     0.00   |
> |   UQ   |      FlexRound       |     0.23   |
> |   BCQ  |        Greedy        |     0.00   |
> |   BCQ  |     Alternating      |     0.00   |
> |   BCQ  |      **FlexBCQ**     |   **22.54**|
>
> **[PPL Scores]**
>
> We summarize the perplexities of quantized models on WikiText-2 and C4 benchmarks in Table R4-4. We use 2 to 4-bit for quantization to compare the performance of FlexBCQ and competitors exhaustively. As a result, FlexBCQ exhibits the lowest perplexities (the best performance) in all settings, especially in 2-bit constraints.
>
> Table R4-4. Comparison of perplexities on WikiText-2 and C4 benchmarks of Llama-3 models quantized using FlexBCQ and competitors under 2 to 4-bit constraints. Lower perplexities mean better performances.
> | Scheme |        Method        |   4-bit Wiki   |   4-bit C4   |   3-bit Wiki   |   3-bit C4   |   2-bit Wiki   |   2-bit C4   |
> |:------:|:--------------------:|:--------------:|:------------:|:--------------:|:------------:|:--------------:|:------------:|
> |   -    |   Full precision     |      6.14      |     8.89     |      6.14      |     8.89     |      6.14      |     8.89     |
> |   UQ   |         RTN          |      6.75      |     9.67     |     10.82      |    14.85     |   1.97E+06     |   2.44E+06   |
> |   UQ   |         AWQ          |      6.54      |     9.40     |      8.23      |    11.59     |   1.72E+06     |   2.13E+06   |
> |   UQ   |        GPTQ          |      6.54      |     9.36     |      9.05      |    11.70     |     450.59     |     254.37   |
> |   UQ   |     OmniQuant        |      6.69      |     9.59     |      8.82      |    12.36     |     987.10     |    1395.17   |
> |   UQ   |      FlexRound       |      6.55      |     9.36     |      7.62      |    10.76     |      68.54     |      66.57   |
> |   BCQ  |        Greedy        |   6.22E+04     |  2.45E+04    |   8.32E+04     |  3.81E+04    |   4.96E+05     |   6.85E+05   |
> |   BCQ  |     Alternating      |      7.70      |    10.88     |     869.89     |    978.83    |   7.34E+06     |   3.97E+06   |
> |  **BCQ** |    **FlexBCQ**    |    **6.46**    |   **9.24**   |    **7.42**    |  **10.46**   |   **14.95**    |   **16.55**  |
>
> In summary, FlexBCQ exhibits outstanding performance not only in the experimental results highlighted in the paper but also in 2-bit quantization and perplexity benchmarks.

---

> ### Author Response · Authors · 2024-11-23
>
> > **[R4-4]** The absence of code and insufficient implementation details in Appendix B raise concerns about reproducibility. Providing the code or more comprehensive details would significantly aid in evaluating the robustness and applicability of the proposed method.
>
> Regrettably, due to the current circumstances of the authors, we are unable to release the code at this time. Nevertheless, we are making every effort to eventually release the code as open-source, enabling others to benefit from the superior performance of FlexBCQ. If the paper is accepted, we aim to make the code publicly available by that time.
>
> In addition, we are updating Appendix B to include detailed information on the implementation of FlexBCQ. Given the current unavailability of the code, we will provide comprehensive details on performance variations with respect to hyperparameters to facilitate the reproduction of our results without requiring access to the code.
>
> > **[R4-5]** Key innovations of FlexBCQ.
>
> We will sequentially explain the problem FlexBCQ aims to solve and the method it employs, starting from the background of its development.
>
> 1. **Background: Lack of Optimization Algorithms for BCQ.** Although BCQ theoretically offers superior representational power, it suffers from low performance due to the lack of optimization algorithms specifically designed for it. In contrast, UQ, despite its limited representational power, achieves high performance thanks to the development of many effective optimization techniques. A representative example is the flexible mapping shown in Figure 1(a).
>
> 2. **Intuition: Transferring Optimization Techniques.** We hypothesize that transferring UQ's advanced optimization techniques to BCQ would enable BCQ to achieve superior performance compared to existing algorithms.
>
> 3. **Challenge: Difficulty in Transferring Optimization Techniques.** However, transferring UQ's optimization techniques to BCQ poses a significant challenge due to the fundamental differences in their quantization schemes. These schemes differ in both the quantization parameters they use and the methods for utilizing these parameters. Without additional ideas, transferring optimization techniques between them is infeasible.
>
> 4. **Observation: Analyzing the Characteristics of Quantization Schemes.** To address the challenge in (3), we analyze the quantization processes of UQ and BCQ. As shown in Figures 3(a) and (b), we observe that UQ's superior optimization techniques stem from its transformation process, while BCQ's representational power is derived from its mapping process to non-uniform quantization levels.
>
> 5. **Main Idea: Integrating Quantization Processes.** Based on the observations in (4), we integrate the quantization processes of UQ and BCQ. Specifically, quantization is performed by first applying the transformation process from UQ and then proceeding with BCQ's mapping process. This algorithm is named FlexBCQ, which combines UQ's optimization techniques with BCQ's representational power.
>
> 6. **Main Idea 2: Composition Theorem for Efficiency** A straightforward integration of the two quantization processes would result in an increased number of quantization parameters and a more complex inference process, leading to significant memory and latency overhead. To prevent this, we utilize FlexBCQ's integrated quantization process only during the optimization (calibration) phase. For storage, we replace the integrated process with a single BCQ representation. This replacement process is formalized in our proposed Composition Theorem, which ensures that the integrated quantization process can be simplified into a single BCQ representation.
>
> 7. **Contribution: Outstanding Performance.** The FlexBCQ algorithm proposed in (5) demonstrates superior performance across various benchmarks (Tables 1, 2, R4-2, and R4-3). Furthermore, it effectively utilizes UQ's optimization techniques and BCQ's representational power as intended, as illustrated in Figures 4(a) and 4(b).
>
> This concludes our explanation of FlexBCQ, from its background to its contributions. We will revise the paper to ensure that readers can understand the above content without requiring additional explanations. These revisions will be completed during the discussion period, and we will provide a summary of the changes once they are finalized.

---

> ### Author Response · Authors · 2024-11-23
>
> > **[R4-6]** How do the running time and inference speed of FlexBCQ compare with those of baseline methods? I am particularly interested in whether this method is both time and memory efficient.
>
> **[Running Time]**
>
> **Our study primarily focuses on achieving superior accuracy at a given bit-width.** For LLMs, the quantization process is performed only once, and the model can be used repeatedly thereafter. Consequently, as long as the quantization process is not prohibitively expensive, we consider a model with higher accuracy to be superior.
>
> We compare the time required to quantize the Llama-3 8B model to 2, 3, and 4 bits using FlexBCQ and its main competitor, FlexRound. The results of this comparison are summarized in Table R4-5.
>
> **Table R4-5.** Running times in seconds for quantizing Llama-3 8B models into 2 to 4bits using FlexRound and FlexBCQ.
> |    Method    |  4-bit  |  3-bit  |  2-bit  |
> |:------------:|:-------:|:-------:|:-------:|
> |  FlexRound   |  16526  |  16523  |  16523  |
> |  FlexBCQ     |  36505  |  27306  |  25412  |
>
> FlexBCQ employs non-uniform quantization levels, requiring weights to be mapped by identifying the nearest quantization level, rather than using the rounding function typically used in uniform quantization. This process of finding the nearest quantization level is more time-consuming than rounding, resulting in longer running times. Specifically, the GPU-unfriendly BST-based algorithm proposed by Xu et al. [1] (Algorithm 1 in Xu et al. [1]) leads to execution times that are practically infeasible to large language models.
>
> To mitigate this issue, we replace the BST-based approach with a GPU-friendly algorithm that computes the distances between each weight and all quantization levels simultaneously to identify the nearest quantization level. This replacement significantly reduces the running time while preserving FlexBCQ’s superior performance. Further exploration of methods to optimize running time without compromising performance is a promising direction for future research, which we plan to address in subsequent studies. We will include a discussion about the running time of FlexBCQ in the paper during the discussion period.
>
> **[Inference Speed]**
>
> In terms of inference speed, FlexBCQ matches the speed of conventional BCQ and benefits from acceleration via the LUT-GEMM kernel. The inference speed results for the LUT-GEMM kernel are summarized in Table R4-6.
>
> **Table R4-6.**  Latency comparison of the first FFN layer on OPT-175B model with various precision and corresponding kernel selections with m = 12288 and g = 128 on A100-80GB-GPU (Table 1 in LUT-GEMM).
> | Kernel    | Schemes | Weight  |     Input        |     Output    | Latency (ms) | Speedup ($\times$) |
> |:---------:|:------------------:|:------------------------:|:----------:|:--------:|:-------------:|:------------------:|
> | cuBLAS    |         -          |         FP16            |    FP16    |   FP16   |     0.7256    |        1.0         |
> | GPTQ      |         UQ         |         INT3            |    FP16    |   FP16   |     0.3599    |        2.0         |
> | AWQ       |         UQ         |         INT4            |    FP16    |   FP16   |     0.3238    |        2.2         |
> | LUT-GEMM  |      UQ, BCQ       |         INT4*           |    FP16    |   FP16   |     0.2688    |        2.7         |
> | LUT-GEMM  |      UQ, BCQ       |         INT3*           |    FP16    |   FP16   |     0.2250    |        3.2         |
>
>
> In summary, LUT-GEMM exhibits superior acceleration capabilities compared to existing UQ kernels such as AWQ and GPTQ. However, since LUT-GEMM delivers equivalent acceleration for both BCQ and UQ, all methods listed in Tables 1 and 2 achieve identical acceleration performance at the same bit-width.
>
> **[References]**
>
> [1] Xu, Chen, et al. "Alternating multi-bit quantization for recurrent neural networks." arXiv preprint arXiv:1802.00150 (2018).

---

> > ### Author Response · Authors · 2024-11-28
> >
> > Dear reviewer mGfi,
> >
> > We updated the revised version of our paper including all discussions to address your concerns. The overall improvements are summarized in the Global Response, and we have addressed the your feedback as outlined below.
> >
> > * **[R4-1] Figures 1 and 2:** We revise both figures to clearly convey our main idea and insert detailed captions for them.
> > * **[R4-2] Composition Theorem:** We clearly state the effect of Composition Theorem in lines 261-296. We also include detailed explanation in Section E.2 in Appendix.
> > * **[R4-3] Performance:** We summarize the outstanding performance of FlexBCQ in perplexity, 2-bit, and channel-wise quantization experiments and include such results in Section G in Appendix. Note that FlexBCQ exhibits the best performance in every setting.
> > * **[R4-4] Implementation Detail:** We elaborate on implementation detail of FlexBCQ including extensive experimental results that promotes reproducibility of FlexBCQ in Section C in Appendix (Inclduing all sections from Section C.1 to C.6).
> > * **[R4-5] Key innovation:** We improve the overall readability of the paper to make readers catch the key innovation of this paper. We clearly state that the contribution of this work is to “transfer UQ’s useful optimization techniques to optimize BCQ” in lines 85-86.
> > * **[R4-6] Running time:** We include discussion about running time and memory usage in Section F.1in Appendix.
> >
> > These are our responses to your feedback. We kindly ask that you review the responses and the revised paper. If you have any unresolved concerns, we would be happy to discuss them further.

---

### Official Review · Reviewer_HKka · 2024-11-05

**Soundness:** 3
**Presentation:** 3
**Contribution:** 3
**Rating:** 5
**Confidence:** 2

**Summary:**

This paper presents FLEXBCQ, a novel binary-coded quantization (BCQ) algorithm that enhances the compression efficiency of large language models without compromising accuracy. FLEXBCQ combines Unified Quantization (UQ) and internal BCQ, benefiting from UQ’s advanced optimization while preserving BCQ's expressive power. The paper also introduces a unified initialization method for more accurate quantization parameter initialization and an optimization technique based on block output reconstruction to improve model accuracy. Experimental results demonstrate improvements over existing UQ and BCQ algorithms on the MMLU 5-shot benchmark.

**Strengths:**

1.	This method leverages UQ's advanced optimization techniques while maintaining BCQ's strong expressive power, which shows its novelty.
2.	Its unified initialization method and block output reconstruction optimization techniques are innovative approaches, making sense for future research in the quantization field.
3.	The experimental data in the paper is comprehensive, and the visual analysis is clear and concrete.

**Weaknesses:**

1.	Some of FLEXBCQ's optimization techniques, such as gradient filtering and periodic remapping, require tuning additional hyperparameters, which could increase the complexity of usage. May authors could explore ways to automatically set these hyperparameters.
2.	The unified initialization method proposed by FLEXBCQ is specifically designed for this algorithm and does not offer alternative initialization methods, potentially limiting its applicability to other quantization algorithms.
3.	The writing needs improvement in the sequence of formula derivations and other
details, as the readability of the derivation process is poor. For example, the formulas in Eq1 and Eq4 appear somewhat disorganized and need a more concise and clear description.

**Questions:**

1.	Could you please discuss the sensitivity of tuning these hyperparameters and provide guidelines for tuning them?
2.	It may help to discuss the potential generaliiability of the unified initialization method to other quantization algorithms
3.	Are there any related works beyond UQ and BCQ that perform similar large model quantization, which could further enrich the experimental data?

---

> ### Author Response · Authors · 2024-11-23
>
> We appreciate your reviews. Below are our responses to your concerns. All of the contents in this response will be reflected in our paper, highlighted in blue color.
>
> > **[R3-1]** Guidelines for selecting hyperparameters in FlexBCQ.
>
> As you correctly noted, gradient filtering and periodic remapping introduce two additional hyperparameters (one for each), potentially leading to increased hyperparameter tuning costs. **To address this concern, we demonstrate throughout the paper that FlexBCQ achieves strong performance using only two fixed settings**, without requiring extensive hyperparameter tuning for these components. Furthermore, additional performance gains can be achieved with further tuning.
>
> To provide insights into how FlexBCQ’s performance varies with changes in hyperparameters, we conducted the following supplementary experiments.
>
>
> **[Gradient filtering]**
>
> We adopt a gradient filtering technique to exclude gradients that could negatively impact performance during optimization and introduce the hyperparameter $\tau$ as the filtering threshold. The selection of $\tau$ in FlexBCQ takes into account the mapping process and can be explained as follows:
>
> 1. **Mapping in FlexBCQ:** In FlexBCQ, the mapping process is performed in the space of transformed weights by mapping each weight to its nearest quantization level. If integers in the range $[0, 2^k-1]$ are used as quantization levels, as shown in Figure 2(a), this corresponds to uniform quantization (UQ). Alternatively, if a binary tree is used to represent non-uniform quantization levels, as shown in Figure 2(b), it corresponds to FlexBCQ.
>
> 2. **Uniform Quantization for Illustration:** For simplicity, consider the case of uniform quantization. Weights within the clipping range $[w_{m,c}, w_{M,c}]$ are transformed to real values within $[0, 2^k-1]$, while weights outside this range take on values beyond, e.g. $w_m$ and $w_M$. These out-of-range weights cause substantial mapping errors. In the transformed space, the interval between quantization levels in UQ is 1, and each level has a range of 0.5 on either side. Thus, it is reasonable to allow a margin of 0.5 even for the extreme levels $0$ and $2^{k}-1$.
>
> 3. **Generalization to FlexBCQ:** For FlexBCQ, the value of 0.5 is replaced by the smallest scale factor $min(\alpha\_{(k)})$ in BCQ. Therefore, $\tau$ is assigned as the smallest scale factor in each weight group, automatically providing a threshold suited to the group’s characteristics.
>
> To evaluate the effectiveness of the $min(\alpha\_{(k)})$ criterion described in (3), we compare it against the UQ-based threshold of 0.5 from (2), as well as alternative thresholds of 0.25 (half of 0.5) and 1.0 (double of 0.5). The results of these experiments are presented in Table R3-1.
>
> **Table R3-1.** MMLU benchmark accuracies of 3-bit Llama-3 models quantized by FlexBCQ with different gradient filtering threshold $\tau$.  $min(\alpha_{(k)})$  represents that $\tau$ is equal to the minimum value of $\alpha_{(k)}$ in each weight group.
> |       Filtering threshold ($\tau$)        | MMLU 5-shot | MMLU 0-shot |  Avg.   |
> |:----------------:|:-----------:|:-----------:|:-------:|
> |       0.25       |    57.95    |    52.63    |  55.29  |
> |       0.5        |    58.68    |    55.03    |  56.85  |
> |        1         |    58.98    |    54.03    |  56.50  |
> | $min(\alpha_{(k)})$ |    59.33    |    54.49    |  56.91  |
>
>
> As a result, $min(\alpha\_{(k)})$, which is employed in our paper, shows the highest accuracy by providing the adaptive gradient filtering threshold for each weight group. Therefore, **we recommend users to use $\tau=min(\alpha\_{(k)})$.**

---

> > ### Author Response · Authors · 2024-11-23
> >
> > **[Periodic remapping]**
> >
> > The periodic remapping technique is a technique that performs weight remapping to updated quantization levels not at every SGD update step during Blockwise Output Reconstruction but at intervals of $p$ steps. In all of our experiments, as described throughout the paper, we use $p$ values of 1 and 2. Table R3-2 presents the performance variations of FlexBCQ across different remapping periods.
> >
> > **Table R3-2.** MMLU benchmark accuracies of 3-bit Llama-3 8B models quantized by FlexBCQ with different remapping period $p$.
> > |   Remapping Period (p)   | MMLU 5-shot | MMLU 0-shot |  Avg.   |
> > |:-----:|:-----------:|:-----------:|:-------:|
> > |   1   |    58.87    |    53.98    |  56.42  |
> > |   2   |    59.33    |    54.49    |  56.91  |
> > |   4   |    59.05    |    52.91    |  55.98  |
> > |   8   |    59.23    |    55.13    |  57.18  |
> > |  16   |    59.12    |    54.23    |  56.68  |
> >
> > The experimental results indicate that the performance of FlexBCQ exhibits slight variations based on the remapping period $p$. When $p=2$, as employed in the paper, FlexBCQ achieves strong performance. Moreover, setting $p=8$ results in even higher performance than that reported in the paper. **In conclusion, using $p=2$ without hyperparameter tuning is sufficient to achieve the excellent results proposed in the paper.** Additionally, further performance gains can be realized by tuning $p$ for specific configurations.
> >
> > Therefore, we recommend to use $\tau=min(\alpha\_{(k)})$ and $p=2$ when training FlexBCQ. Discussion about hyperparameter selection will be included in the revised version of the paper, which will appear during the discussion period.

---

> ### Author Response · Authors · 2024-11-23
>
> > **[R3-2]** Potential generality of Unified Initialization.
>
> FlexBCQ is the only algorithm designed to enhance the performance of BCQ by transferring training techniques from UQ. To support this novel category of algorithms, we propose Unified Initialization algorithm. A key advantage of FlexBCQ lies in its potential to transfer not only UQ techniques such as FlexRound but also other algorithms like OPTQ and OmniQuant into the BCQ framework. Although unified initialization cannot be directly applied to the quantization parameter initialization of conventional UQ algorithms, it becomes applicable when these algorithms are adapted to BCQ through the FlexBCQ methodology.
>
> **Given the development of fast inference kernels for BCQ [1,2] and the strong performance demonstrated by FlexBCQ, transferring various UQ techniques to BCQ represents a promising research direction. In this context, unified initialization is expected to play a critical role.** We will incorporate this discussion into the paper and provide you with the exact sections where these revisions have been made.
>
> ----
>
> > **[R3-3]** Writing and readability of Mathematical expressions.
>
> We appreciate your observation regarding the need to improve the readability of the paper. To address this, we will enhance the explanations of the equations to ensure that our intended message is communicated more clearly. In addition to your specific suggestions, we are also incorporating feedback from other reviewers to comprehensively refine the manuscript. These updates are underway, and we will inform you once the revisions are complete. All updates will be finalized within the discussion period.
>
> ----
>
> > **[R3-4]** Related works beyond UQ and BCQ.
>
> Uniform Quantization (UQ) has become the most widely studied quantization format, supported by the number of publicly available kernels optimized for fast inference. Recent advancements [1,2], including LUT-GEMM[1], have demonstrated the ability to achieve faster inference speeds than existing UQ kernels at the same bit-width by leveraging the unique computational characteristics of Binary-Coding Quantization (BCQ). Given this context, our work focuses on enhancing the performance of BCQ, and we therefore restrict the scope of our paper to UQ and BCQ.
>
> Below, we provide a summary of data on the inference speed of LUT-GEMM.
>
> **Table R1-4.**  Latency comparison of the first FFN layer on OPT-175B model with various precision and corresponding kernel selections with m = 12288 and g = 128 on A100-80GB-GPU (Table 1 in LUT-GEMM [1]).
>
> | Kernel    | Schemes | Weight  |     Input        |     Output    | Latency (ms) | Speedup ($\times$) |
> |:---------:|:------------------:|:------------------------:|:----------:|:--------:|:-------------:|:------------------:|
> | cuBLAS    |         -          |         FP16            |    FP16    |   FP16   |     0.7256    |        1.0         |
> | GPTQ      |         UQ         |         INT3            |    FP16    |   FP16   |     0.3599    |        2.0         |
> | AWQ       |         UQ         |         INT4            |    FP16    |   FP16   |     0.3238    |        2.2         |
> | LUT-GEMM  |      UQ, BCQ       |         INT4*           |    FP16    |   FP16   |     0.2688    |        2.7         |
> | LUT-GEMM  |      UQ, BCQ       |         INT3*           |    FP16    |   FP16   |     0.2250    |        3.2         |
>
> One potential category to be included in this paper is rotation-related UQ techniques [3,4]. These techniques involve an additional transformation that multiplies the weight matrix with an orthonormal matrix and can also be applied to FlexBCQ; in QuaRot paper, they compare the performance of QuaRot+RTN and QuaRot+GPTQ, illustrating that QuaRot serves as an auxiliary technique (Table 1 in QuaRot). As such, our study does not include comparisons with these methods.
>
> **[References]**
>
> [1] Park, Gunho, et al. "Lut-gemm: Quantized matrix multiplication based on luts for efficient inference in large-scale generative language models." arXiv preprint arXiv:2206.09557 (2022).
>
> [2] You, Haoran, et al. "ShiftAddLLM: Accelerating Pretrained LLMs via Post-Training Multiplication-Less Reparameterization." arXiv preprint arXiv:2406.05981 (2024).
>
> [3] Ashkboos, Saleh, et al. "Quarot: Outlier-free 4-bit inference in rotated llms." arXiv preprint arXiv:2404.00456 (2024).
>
> [4] Liu, Zechun, et al. "SpinQuant--LLM quantization with learned rotations." arXiv preprint arXiv:2405.16406 (2024).

---

> ### Author Response · Authors · 2024-11-28
>
> Dear reviewer HKka,
>
> We updated the revised version of our paper including all discussions to address your concerns. The overall improvements are summarized in the Global Response, and we have addressed the your feedback as outlined below.
>
> * **[R3-1] Hyperparameter selection:** We include sensitivity analysis regarding $p$, $\tau$, sample dataset, sample size, epochs, and clipping strategies in Appendix C.3 to C.6.
> * **[R3-2] Unified Initialization:** We include the potential of unified initialization in lines 509-511.
> * **[R3-3] Readability:** We made every effort to improve the writing of the paper. The overall improvements are summarized in the Global Response [G2]. We clean the mathematical notations by removing redundant equations and introducing the notation $\Theta$ for representing a set of quantization parameters.
> * **[R3-4] Quantization schemes:** We clearly state that our focus is on UQ and BCQ algorithms which are supported by LUT-GEMM in lines 99-104.
>
> These are our responses to your feedback. We kindly ask that you review the responses and the revised paper. If you have any unresolved concerns, we would be happy to discuss them further.

---

### Official Review · Reviewer_xKLF · 2024-11-06

**Soundness:** 3
**Presentation:** 2
**Contribution:** 2
**Rating:** 5
**Confidence:** 4

**Summary:**

The paper presents FLEXBCQ (Flexible Binary Coding Quantization), a method for compressing neural networks using a UQ and an inner BCQ, allowing advantage of both the sophisticated optimizing techniques of UQ, specifically the flexible mapping technique, and the powerful expressive capability of BCQ. Especially, the approach employs unified initialization algorithm for initializaiton and apply Straight-Through Estimator (STE) for non-differetiable steps with blockwise output reconstruction error to optimize quantization accuracy per block. Compared to FlexRound and other baselines, this method demonstrates significant compression rates while maintaining model performance.

**Strengths:**

1. Effective Combination of UQ and BCQ: The integration of Uniform Quantization (UQ) and Binary Coding Quantization (BCQ) leverages the strengths of both approaches, providing flexible mapping and expressive representation capabilities. The paper demonstrates the effectiveness of this combination, especially in experiments with the Llama-3 70B model.

2. Unified Initialization and Blockwise Output Reconstruction: The unified initialization approach, combined with blockwise output reconstruction, enhances quantization accuracy by aligning each block more closely with the original model output. This method strengthens the reliability of the quantization process and contributes to maintaining model performance post-compression

**Weaknesses:**

1. Readability Issues: The reading experience could be improved; please refer to the Questions section for specific details.

2. Additional Bits for Hyperparameters: Both quantization algorithms in FLEXBCQ require additional bits to store hyperparameters. Please include the bitwidth calculations for FLEXBCQ and list them when comparing against baselines in Table 1. Compared to FlexRound and UCB, FLEXBCQ uses additional storage, which contributes to its better performance. A fair comparison with the baselines should account for these extra storage requirements.

3. Clipping Strategy in Algorithm 1: In Algorithm 1, only the minimal values are iteratively clipped, rather than clipping both minimal and maximal values. Given that outliers could also appear on the maximal side, this may not be the optimal strategy. Please provide further explanation or an ablation study to justify this design choice

**Questions:**

1. I assume that the terms weight group and blockwise refer to the same concept in the context of FLEXBCQ, correct?

2. In Section 3.3 (Line 303) and Section 3.4 (Line 324), $M_U$ is mentioned as part of the algorithm. However, it is not reflected in Equation (4) or in Algorithm 1 (after Line 279). Could you clarify this?

3. Please have more details regarding the optimization for $\delta(k), z_U(k), s, s_r$, and $\alpha(k)$ when minimizing blockwise output reconstruction error. Is this process similar to Algorithm 1, using a grid search approach?

4. Please provide more details regarding the optimization of $\Delta(k),z_U(k),s,s_r$, and $\alpha(k)$when minimizing blockwise output reconstruction error. Is this process similar to Algorithm 1, using a grid search approach?

5. Line 140: $W_{M,c} - W_{M,c}$ →  $W_{M,c} - W_{m,c}$

6. Could you provide more details in your figures to clarify their meaning? These adjustments would help clarify the figures and strengthen the visual argument for FLEXBCQ’s effectiveness.

        a. Figure 1: What do the gray bars and blue arrows represent? This figure seems to suggest that a point can be attributed to quantized values, but I'm not entirely certain. Could you explain if this interpretation is correct?

        b. Figure 4: Are 4.(a) and 4.(b) for the same weight? Are there any correspondences between their distributions? It’s unclear if Figure 4(b) alone sufficiently demonstrates that FLEXBCQ outperforms FlexRound. It is best to explain the y-axis of Fig. 4(b) in the title.

7. In lines 160-161, is $s, s_r$ for a whole weight like $W_k$ or a single weight block?

---

> ### Author Response · Authors · 2024-11-23
>
> We sincerely appreciate your meticulous review. Your detailed feedback, including insights into aspects not explicitly addressed in the paper, has greatly helped to enhance the clarity and readability of our paper. Below, we have systematically addressed the points you raised. All discussions from the discussion period will be reflected in the revised manuscript, with updates regarding readability to be included in the "Global Response" section upon completion.
>
> ----
> > **[R2-1]** Readability Issues: The reading experience could be improved; please refer to the Questions section for specific details.
>
> We greatly appreciate your insightful and detailed questions. To address your concerns, we will revise the paper to ensure that the points you raised can be understood clearly by readers without requiring additional explanation. Readability improvements are currently underway, and once the revisions are finalized, we will provide a comprehensive summary outlining how your feedback has been incorporated.
>
> ----
> > **[R2-2]** Please include the bit width calculations in Table 1. A fair comparison with the baselines should account for these extra storage requirements.
>
> Thank you for highlighting this crucial aspect. As you noted, BCQ offers higher representational capacity than UQ but requires a greater number of quantization parameters. Specifically, the average number of bits per weight for models quantized to 3 and 4 bits has been calculated for both UQ and BCQ, as summarized in Table R2-1. **BCQ is observed to utilize approximately 8% more bits than UQ. Detailed calculations supporting these results are provided at the end of this response.**
>
> **Table R2-1.** The average number of bits per weight after quantization by BCQ and UQ under 3-bit and 4-bit weight quantization settings.
>
> |  Scheme  | 4 bit |    3 bit     |
> |:--------:|:-------------------:|:-------:|
> |    UQ    |       4.25          |  3.25   |
> |   BCQ    |       4.625          |  3.50   |
>
> Regarding this observation, we would like to address the following two key points:
>
> (1) In this study, **we prioritize acceleration capabilities over model storage size.** The LUT-GEMM kernel, which outperforms existing UQ kernels such as AWQ and GPTQ in speed, ensures **identical inference speeds for BCQ and UQ**. Therefore, we believe that comparing the accuracy of UQ and BCQ algorithms under the same bit-width conditions is a fair and reasonable approach.
>
> (2) We define a "weight group" by grouping a specific number of weights and assigning quantization parameters to each group. **As the weight group size increases, the proportion of bits allocated to quantization parameters becomes smaller relative to the total storage bits.** For example, with column-wise quantization, compressing the Llama-3 8B model to 4 bits results in an average of 4.02 bits per weight, with quantization parameters contributing only 0.02 bits per weight—an almost negligible amount.
>
> **In line with this rationale, we base our comparisons on the bits used for weight compression rather than the average bits per weight.** As per your suggestion, we will add details regarding the storage usage differences between UQ and BCQ, as well as the calculation of bits for quantized models, to the Appendix. Additionally, we will reference these bit differences when discussing Table 1. These updates are currently being implemented, and we will notify you once they are complete.
>
> ****
>
> Below, we detail the process for calculating Table R2-1.
>
> **[Calculation of Table R2-1]**
>
> When a weight group $w\in \mathbb{R}^{g}$ consisting of $g$ weights is quantized into $k$-bit using either the UQ or BCQ format, the total number of bits required for storage is calculated as shown below. In these calculations, the quantization parameters ($\boldsymbol{\alpha}_{(k)}$, $z\_{B,(k)}$, $\Delta\_{(k)}$, and $z\_{U,(k)}$) are set to 16 bits, consistent with the value used in our study.
>
> (# bits for BCQ) = $gk$ (for $\boldsymbol{b}\_{(k)}$) + $k$ * 16 (for $\boldsymbol{\alpha}\_{(k)}$) + 1 * 16 (for $z\_{B,(k)}$)  = $(gk + (k+1)16)$
>
> (# bits for UQ) = $gk$ (for $\widetilde{w}$) + 16 (for $\Delta\_{(k)}$) + 16 (for $z\_{U,(k)}$) = $(gk + 2 * 16)$
>
> In this context, $\boldsymbol{b}\_{(k)}\in\\{-1,1\\}^{g\times k}$, $\boldsymbol{\alpha}_{(k)}\in\mathbb{R}^{k}$, $z\_{B,(k)}\in\mathbb{R}$ denote the binary codes for weights, BCQ’s scale factor, and BCQ’s shifting factor, respectively.
> $\widetilde{w}\in ([0,2^{k-1}]\cap\mathbb{Z})^{g}$, $\Delta\_{(k)}\in\mathbb{R}$, and $z\_{U,(k)}\in\mathbb{R}$ represent the weights mapped to integers, UQ’s scale factor, and UQ’s zero-point, respectively.
> Therefore, we have the average number of bits per weight for each quantization format as follows.
>
> (average # bits per weight for BCQ) = $(gk + (k+1)16) / g$ = $k + 16(k+1)/g$
>
> (average # bits per weight for UQ) = $(gk + 2 * 16) / g$ = $k + 32/g$

---

> ### Author Response · Authors · 2024-11-23
>
> > **[R2-3]** Clipping Strategy in Algorithm 1: In Algorithm 1, only the minimal values are iteratively clipped, rather than clipping both minimal and maximal values. Given that outliers could also appear on the maximal side, this may not be the optimal strategy. Please provide further explanation or an ablation study to justify this design choice.
>
> We sincerely appreciate your detailed review of our algorithm. Our clipping range search algorithm fixes the minimum value while proportionally adjusting the maximum value during the search process. Before we begin the explanation, we need to revise line 4 in Algorithm 1.
>
> * **(Revised in line 4)** $\Delta_{(k)} \gets (\gamma (w_M - w_m))/(2^{k}-1)$ and  $z_{U, (k)} \gets \lfloor -w_m / \Delta_{(k)} \rceil)$
>
> When we design Unified Initialization, there are three possible clipping strategies:
>
> * The Fixed Minimum method. which fixes $w_m$ and adjusts only $w_M$
> * The Fixed Maximum method. which fixes $w_M$ and adjusts only $w_m$
> * The Balanced method. which adjusts both $w_m$ and $w_M$
>
> The performance of these strategies for compressing the Llama-3 8B model to 3 bits is presented in Table R2-2. Experimental results reveal that while all three strategies deliver comparable performance, the Fixed Minimum strategy achieves the the higher score than the Balanced strategy. Therefore, Fixed Minimum, used in our paper, is a proper strategy for initialization. Furthermore, as shown in the table, you can further improve the performance of FlexBCQ if you use Fixed Maximum.
>
> **Table R2-2.** MMLU 5-shot and 0-shot benchmark scores of 3-bit quantized Llama-3 8B models with three clipping strategies.
>
> | Clipping Strategy | MMLU 5-shot | MMLU 0-shot | Average |
> |:------------------:|:-----------:|:-----------:|:-------:|
> |   Fixed Minimum    |    59.33    |    54.49    |  56.91  |
> |   Fixed Maximum    |    59.23    |    54.99    |  57.11  |
> |     Balanced       |    58.77    |    54.59    |  56.68  |
>
> As you correctly noted, the Fixed Minimum approach may result in performance degradation if extreme outliers occur at the higher end of the range. However, in large language models (LLMs), it is well established that those extreme outliers predominantly occur in activations [1-3], and extreme outliers are rare in weights.
>
> Based on the results of the R2-2 experiment, we have chosen to use the Fixed Minimum method. A discussion of this rationale will be added to Section 3.3 and Appendix B.1 (Implementation Details of FlexBCQ) of the paper. Furthermore, we will introduce an argument about clipping strategy in our source code to enable users to freely select the clipping strategies that they want to use.
>
>
> ----
>
> > **[R2-4]** I assume that the terms weight group and blockwise refer to the same concept in the context of FLEXBCQ, correct?
>
> We acknowledge that the definitions of key terms in our paper may not have been sufficiently clear, and we apologize for any confusion this may have caused. To address this, we have organized the terminology used in the paper by ascending order of scale, starting from the smallest unit, "weight," to the largest unit, "model":
>
> * **Weight:** the smallest unit, representing an individual numerical weight value.
> * **Weight group:** a collection of weights grouped by a specified group size, _all of which share the same quantization parameters._
> * **Weight matrix:** a two-dimensional matrix composed of weights, containing multiple weight groups.
> * **Layer:** a component that performs affine transformations using a weight matrix and a bias vector. For example, Q, K, V, O, U, G, and D in Table 4 of the paper represent individual layers.
> * **Module:** a collection of layers that performs a specific function. In Transformers, modules include Multi-Head Attention (MHA) and Multi-Layer Perceptron (MLP).
> * **Block:** a fundamental unit of a Transformer, consisting of one MHA module and one MLP module.
> * **Model:** a complete language model composed of multiple blocks.
>
> **It is important to note that weight groups and blocks refer to different levels of grouping, with a block encompassing numerous weight groups.** To improve clarity, we will refine the descriptions of these terms where they appear in the paper and include a comprehensive glossary in the Appendix.

---

> > ### Author Response · Authors · 2024-11-23
> >
> > > **[R2-5]** In Section 3.3 (Line 303) and Section 3.4 (Line 324), $M_U$ is mentioned as part of the algorithm. However, it is not reflected in Equation (4) or in Algorithm 1 (after Line 279). Could you clarify this?
> >
> > FlexBCQ is an algorithm that incorporates the transformation process of UQ into the existing BCQ quantizer, as described in Equation (4). This integration enables FlexBCQ to leverage UQ’s effective optimization techniques. Therefore, the mapping function used in FlexBCQ is $M_B$​, not $M_U$​, as shown in Equation (4) and Figure 3. Below, we provide our responses to the relevant comments:
> >
> > *  **Line 303** “We initialize $z\_{B,(k)} = (2^{k} − 1)/2$ and fix it since the transformed space, in which the inner BCQ is defined, is designed for mapping weights to the range [0, $2^k−1$].“
> >
> > Our intended meaning here is that the Transformation function was originally designed for UQ, mapping values to the range $[0,2^k−1]$. As depicted in Figure 2(b), BCQ exhibits symmetry around $z\_{B,(2)}​.$ To align with this symmetry, we fix $z\_{B,(2)}$​ at the center of the mapped range, $(2^{k}-1)/2$. This does not suggest that FlexBCQ uses $M_U​$. To eliminate any ambiguity, we will revise this section for clarity.
> >
> > * **Line 324** “We utilize straight-through estimator (STE) (Bengio et al., 2013) to update parameters corresponding to $\mathcal{T}_F$ since the mapping function $M_U$ is not differentiable”
> >
> > This is a typo. We will revise $M_U$ into $M_B$.
> >
> > To ensure our intentions are clearly conveyed, we will revise these sections accordingly. The updated manuscript will be uploaded during the discussion period, and we will provide a detailed explanation of how these revisions were implemented.
> >
> >
> > ----
> >
> > > **[R2-6]** Please provide more details regarding the optimization of $\Delta_{(k)}$, $z_{U,(k)}$, $s$, $s_r$, and $\alpha_{(k)}$ when minimizing blockwise output reconstruction error. Is this process similar to Algorithm 1, using a grid search approach?
> >
> > We update the quantization parameters of FlexBCQ through the blockwise output reconstruction process where the block indicates a composition of an MHA and MLP modules; Llama-3 8B model has 32 blocks. The quantization parameters of weight groups in the same block are updated simultaneously **using stochastic gradient descent (SGD) with our proposed two optimization techniques (Gradient Filtering and Periodic Remapping).**
> >
> > The details of our block reconstruction process are as follows:
> >
> > * **Objective function:** $||B\_i(X\_i)  - \widehat{B}\_i(\widehat{X}\_i)||\_F^2$
> > * **Optimization target:** quantization parameters of FlexBCQ
> >
> >     * (UQ’s scale factor $\Delta_{(k)}$,
> >       UQ’s zero-point $z_{U,(k)}$,
> >        scale factors $s$ and $s_r$ for flexible mapping, and
> >        BCQ’s scale factor $\boldsymbol{\alpha}\_{(k)}$)
> > * **Optimization algorithm:** stochastic gradient descent (SGD)
> > * **Additional optimization techniques:** Gradient Filtering and Periodic Remapping
> > * **Optimization strategy:** sequential blockwise optimization from the bottom block to the top block
> >
> > $B\_i$ and $\widehat{B}\_i$ are the functions for the $i$-th unquantized and quantized blocks, respectively.
> > $\boldsymbol{X}\_i$ and $\widehat{\boldsymbol{X}}\_i$ are the inputs for the $i$-th unquantized and quantized blocks, respectively.
> > We will elaborate more details about Blockwise output reconstruction in Section 3.4.
> >
> >
> > > **[R2-7]** Line 140: $w\_{M,c} - w\_{M,c} \rightarrow w\_{M,c} - w\_{m,c}$
> >
> > We appreciate your detailed review! We fix the typo and the fixed version will be available soon.

---

> ### Author Response · Authors · 2024-11-23
>
> > **[R2-8]** Figure 1: What do the gray bars and blue arrows represent? This figure seems to suggest that a point can be attributed to quantized values, but I'm not entirely certain. Could you explain if this interpretation is correct?
>
> In Figure 1. (b) and (c), we aim to represent that quantization levels (black bars) of BCQ are adaptive to the weights. Grey bars represent the uniformly distributed quantization levels, which are represented as black bars in (a). Blue arrows mean that BCQ moves the uniformly distributed quantization levels (grey bars) to become closer to the weights (circles).
>
> ----
> > **[R2-9]** (1) Are 4.(a) and 4.(b) for the same weight? (2) Are there any correspondences between their distributions? (3) It’s unclear if Figure 4(b) alone sufficiently demonstrates that FLEXBCQ outperforms FlexRound. (4) It is best to explain the y-axis of Fig. 4(b) in the title.
>
> Our objective in Figure 4 is to illustrate that FlexBCQ successfully utilizes (1) adaptive quantization levels aligned with the weight distribution and (2) the flexible mapping optimization technique from UQ, as we intended. The overall contribution of (1) is highlighted in Table 4, while the contribution of (2) is examined through the ablation study in Table 3. **Thus, Figure 4 provides a case study to visually confirm that (1) and (2) are operating as designed.**
>
> In particular, the goal of Figure 4(b) is not to demonstrate that FlexBCQ is superior to FlexRound. Instead, it aims to show that FlexBCQ effectively utilizes the flexible mapping technique introduced in FlexRound to the same degree as FlexRound itself.
>
> Building on the explanation above, we have addressed your questions as follows.
>
> **q1.** Are 4.(a) and 4.(b) for the same weight?
>
> Figures 4(a) and 4(b) depict visualizations derived from the same weight matrix, specifically the up-projection weight matrix in the second block of Llama-3 8B, quantized to 3 bits. These figures highlight two weight groups within the matrix. While FlexBCQ successfully leverages both flexible mapping and adaptive quantization levels, not all weight groups are equally effective for visualization. Hence, we selected an arbitrary weight matrix and identified the weight groups that best visually demonstrate the impact of these techniques.
>
> It is important to note that this does not imply that flexible mapping and adaptive quantization are confined to the selected weight group. Their contributions to overall model performance can be observed in Tables 3 and 4, respectively.
>
> **q2.** Are there any correspondences between their distributions?
>
> Thank you for providing an insightful analytical perspective. As you suggested, comparing the trends between effective adaptive quantization levels and frequent flexible mapping yields meaningful insights. However, since both techniques are optimized with the shared objective of reconstructing the outputs of Transformer blocks, disentangling their individual contributions and analyzing their trends independently poses significant challenges.
>
> If we hypothesize that the effectiveness of adaptive quantization levels can be measured by minimizing weight-level quantization error, we can indirectly analyze its trend. Since flexible mapping aims to reduce block-level output reconstruction error rather than weight-level quantization error, these two factors are likely to exhibit a negative correlation.
>
> **q3.** It’s unclear if Figure 4(b) alone sufficiently demonstrates that FLEXBCQ outperforms FlexRound.
>
> **The purpose of Figure 4(b) is not to demonstrate that FlexBCQ outperforms FlexRound, but rather to show that FlexBCQ effectively applies flexible mapping to the same extent as FlexRound.** The figure illustrates that FlexBCQ achieves a comparable amount of flexible mapping, indicating its ability to leverage this optimization technique adequately.
>
> To ensure that flexible mapping is not perceived as being confined to a single weight group, we conducted a statistical analysis of the extent of flexible mapping across all weight groups within the model.
>
> **Table R2-3.** Comparison of the proportion (%) of weights undergoing flexible mapping in models quantized using FlexBCQ and FlexRound. The title of each column indicates the differences in the mapped indices between weights with flexible mapping and those without.
> |   Method   |     0      |     1      |      2       |     >2      |
> |:----------:|:----------:|:----------:|:------------:|:-----------:|
> | FlexRound  |  95.4132   |   4.5862   |    0.0005    |  1.30E-05   |
> | FlexBCQ    |  96.0395   |   3.8619   |    0.0985    |  7.62E-05   |
>
>
> The results of this analysis confirm that FlexBCQ effectively utilizes flexible mapping not only in the weight group depicted in Figure 4(b) but also consistently throughout the entire model.
>
> **q4.** It is best to explain the y-axis of Fig. 4(b) in the title.
>
> We agree with your suggestion. We will explain the meaning of y-axis in the title of Fig. 4(b).

---

> ### Author Response · Authors · 2024-11-23
>
> > **[R2-10]** In lines 160-161, is $s$, $s_r$ for a whole weight or a single weight block?
>
> $s$ and $s_r \in \mathbb{R}^{+}$ are scale factors that divide each weight before mapping to enable flexible mappings. **$s$ is assigned individually for each weight, while $s_r$ is assigned for each row in the weight matrix, i.e. all weights in a row share the same $s_r$.** If we have a weight matrix $\boldsymbol{W}\in\mathbb{R}^{3\times 4}$, then we have twelve $s$ and three $s_r$. Note that our main idea is to adapt FlexRound, which is designed for UQ, to BCQ. Therefore, using $s$ and $s_r$ are the ideas of FlexRound and we follow their settings. We will clarify about $s$ and $s_r$ in lines 158-160.
>
> ----
>
> These are our responses to your review. We sincerely appreciate your thoughtful feedback, which has brought to light several key aspects that were not sufficiently addressed in our original submission. We are actively working to incorporate your suggestions into our paper and will notify you as soon as the revisions are finalized.
>
>
> **[References]**
>
> [1] Xiao, Guangxuan, et al. "Smoothquant: Accurate and efficient post-training quantization for large language models." International Conference on Machine Learning. PMLR, 2023.
>
> [2] Ashkboos, Saleh, et al. "Quarot: Outlier-free 4-bit inference in rotated llms." arXiv preprint arXiv:2404.00456 (2024).
>
> [3] Lin, Haokun, et al. "Duquant: Distributing outliers via dual transformation makes stronger quantized llms." The Thirty-eighth Annual Conference on Neural Information Processing Systems. 2024.

---

> > ### Author Response · Authors · 2024-11-28
> >
> > Dear reviewer xKLF,
> >
> > We updated the revised version of our paper including all discussions to address your concerns. The overall improvements are summarized in the Global Response, and we have addressed your feedback as outlined below.
> >
> > * **[R2-1] Readability:** (Overall) We made every effort to improve the writing of the paper. The overall improvements are summarized in the Global Response [G2].
> > * **[R2-2] Memory usage:** (Appendix F.1)  Include discussion of memory usage of FlexBCQ. We clearly state that we directly compare the accuracy of the quantized models since UQ and BCQ algorithms have the same inference speed with LUT-GEMM when they have the same bit width in Lines 101-104, instead of including average bit widths in Table 1.
> > * **[R2-3] Clipping strategy:** (Appendix C.6) We include the discussion about clipping strategies.
> > * **[R2-4] Terminology:** (Appendix A)  We include a section about summarizing terminologies.
> > * **[R2-5] Mapping function** (Line 332, Section 2.2.1) We revise the typo and improve the readability of Section 2.2.1 which describes the Tranformation function.
> > * **[R2-6] Blockwise Output Recon:** (Lines 319-331) We add more detailed explanation about Blockwise Output Reconstruction process
> > * **[R2-7] typo:** (Line 137) Revise the typo regarding $\Delta_{(k)}$
> > * **[R2-8] Figure 1:** (Lines 41-51) We added the baseline Round-to-Nearest (RTN) method and revised the caption accordingly to effectively convey the main idea of the paper. We also revise the explanation about Figure 1 in lines 41-51.
> > * **[R2-9] Flexible mapping:** (Appendix F.2) We include the analysis regarding flexible mapping and explain y-axis of Figure 4(b) in its title.
> > * **[R2-10] $s$ and $s_r$:** (Lines 159-160) We modify the explanation about $s$ and $s_r$ in more detail.
> >
> > These are our responses to your feedback. We kindly ask that you review the responses and the revised paper. If you have any unresolved concerns, we would be happy to discuss them further.

---

### Official Review · Reviewer_y6hi · 2024-11-06

**Soundness:** 3
**Presentation:** 2
**Contribution:** 3
**Rating:** 6
**Confidence:** 4

**Summary:**

This work proposes a combination of two popular techniques: FlexRound for uniform quantization (UQ), and Alternating for Binary-Coded Quantization (BCQ). In UQ, the quantization points are uniformly spaced within a pre-specified range, and the scalar value being quantized, is scaled, by a parameter called "scale" and shifted by a parameter called "zero-point" (to make it zero-centered). For UQ, prior work FlexRound introduces two additional scaling parameters ($s$ and $s_r$ in the paper) that are learnable -- which is shown to improve the performance of vanilla UQ.

On the other hand, BCQ expresses a scalar as sums and differences of a fixed set of scale factors. In theory, BCQ has a larger representation space, and it can be shown that UQ is a special case of BCQ. However, in practice, BCQ is not seen to perform well for LLM quantization practices, primarily due to the difficulty in optimizing its hyperparameters.

This work proposes a combination of both UQ and BCQ steps for quantization, and proposes FlexBCQ -- which has a unified initialization strategy that combines the initialization of both UQ and BCQ. In FlexBCQ, BCQ is applied after an application of FlexRound's transformation (for UQ). The hyper-parameters in FlexBCQ are optimized using a small calibration dataset, and some results show that they better aligns with the distribution of the weights (It is essential to use a quantization scheme that aligns well with the distribution of model weights). Eventually, the combined quantization process does not add any additional memory or latency overhead.

**Strengths:**

The combination of UQ and BCQ is pretty neat, and also seems to be promising based on the set of experiments evaluated on in the paper. LLM compression is an important topic, and the contribution of this work is useful in light of the fact that custom kernels such as LUT-GEMM exist, that leverage BCQs for faster throughput.

The experiments are done on a reasonably wide set of models and tasks. Moreover, additional studies that show the alignment of the (learned) quantization levels with the distribution of weights is also highly appreciable. For the most part, the paper is generally well-written.

**Weaknesses:**

I have some concerns which are mentioned below:

1. It is claimed that prior work on BCQ (Xu et. al., 2018) independently updates $\alpha_{(k)}$ and $\mathbf{b}_{(k)}$ are updated independently. Are there any ablation results that show how it performs for LLM quantization?

2. It might be worth considering ablations on the choice of calibration dataset, especially for Llama-3 models. Can you provide some justification as to how these datasets were chosen? Usually, for compression, subsets of the pre-training datasets are presumed to be good choices as calibration dataset. So was there any particular reason why C4 was chosen and not RedPajama (which is the pre-training dataset for OpenLlama)? Also, any ablations on the size of the pre-training dataset?

3. Is it possible to provide a comparison of the average number of bits per parameter used by FlexBCQ, along with the other strategies it is being benchmarked against?

4. Is it possible to use the kernels from LUT-GEMM to see an actual improvement in inference throughput?

I would be more than happy to reconsider my score contingent on other reviews, and the authors' rebuttal.

**Questions:**

In addition to the concerns mentioned above in the "Weaknesses" section, I have some more queries, and would be grateful if the authors could address them:

1. I initially had about why despite its greater expressive power, BCQ significantly underperforms compared to UQ, which has limited expressive capabilities for LLM compression (as mentioned in lines 67-68)? This was perhaps answered in 199-200 -- Is this a hypothesis and are there concrete datapoints to support this? I might have missed this in the reading of the paper.

2. There is a potential typo in the definition of $\Delta_{(k)}$. Perhaps it should be $(w_{M,c} - w_{m,c})/2^{k-1}$?

3. In Line 195 - 196, please point to a more precise Theorem/Lemma in Park et. al. for reference.

4. How is $\boldsymbol{\alpha}_{(k)}$ chosen? Is it optimized for each group of weights separately?

5. How is the blockwise output reconstruction done? For each weight matrix independently, or for each entire decoder layer at once?

6. The gradient filtering section can benefit from a little more explanation. Does it mean that when the mapping error exceeds epsilon, the gradients are zeroed out?

7. Please specify all the learnable parameters in FlexBCQ explicitly in a sentence somewhere.

8. The LLM quantization problem formalization in the beginning of Sec. 2.1 needs more description.

**Details Of Ethics Concerns:**

None needed.

---

> ### Author Response · Authors · 2024-11-21
>
> We appreciate your thoughtful and constructive review. We answer your reviews below.
> All modifications will be reflected in our paper during the discussion period.
>
> >**[R1-1]** It is claimed that prior work on BCQ (Xu et. al., 2018) independently updates
> $\boldsymbol{\alpha}\_{(k)}$ and $\boldsymbol{b}\_{(k)}$ are updated independently. Are there any ablation results that show how it performs for LLM quantization?
>
> We experiment to compare the effect of updating $\boldsymbol{\alpha}\_{(k)}$ and $\boldsymbol{b}\_{(k)}$ to clarify which one has the greater impact on reducing the quantization error. Before we explain the experimental results, we clarify the updating process of Xu et al. [1]. Xu et al. begins with initializing $\boldsymbol{\alpha}\_{(k)}$and $\boldsymbol{b}\_{(k)}$ using Greedy algorithm (Equations (2) to (4) in Xu et al. [1], which is noted in Greedy in Table 1 in our paper]. After that, they alternatingly update $\boldsymbol{\alpha}\_{(k)}$ and $\boldsymbol{b}\_{(k)}$ by repeating the following two steps (Algorithm 2 in Xu et al. [1]).
>
> [Step 1] Update $\boldsymbol{\alpha}\_{(k)}$ while keeping $\boldsymbol{b}\_{(k)}$ fixed
>
> [Step 2] Update $\boldsymbol{b}\_{(k)}$ while keeping $\boldsymbol{\alpha}\_{(k)}$ fixed
>
> **In our statement that “$\boldsymbol{\alpha}\_{(k)}$ and $\boldsymbol{b}\_{(k)}$ are updated independently”, we intended to convey that one is fixed while updating the other.**
> We measure the quantization error at each updating step to evaluate the impact of updating $\boldsymbol{\alpha}\_{(k)}$ and $\boldsymbol{b}\_{(k)}$. We summarize the experimental results in Table R1-1 and a detailed experimental setup is as follows:
>
> **[Setup]**
>
> * The number of alternating updates: 15 (same as the paper)
> * Bit-width: 3
> * Target weight: query projection’s weight in the first decoder block of Llama-3 8B
> * Quantization error: $||\boldsymbol{W}-\widehat{\boldsymbol{W}}\_{(3)}||_F^2 \times 10^6$
>     * $\boldsymbol{W}$: a pretrained weight, $\widehat{\boldsymbol{W}}\_{(3)}$: 3-bit quantized weight
> * Reduction: (quantization error of the previous step) - (quantization error of the current step)
> * Step 0 indicates the quantization error after Greedy initialization
>
> **Table R1-1.** Quantization error in each update step over 15 iterations of alternating updates.
>
> | Step | Update $\boldsymbol{\alpha}\_{(k)}$ | Reduction | Update $\boldsymbol{b}\_{(k)}$ | Reduction |
> |:------:|:---------------------------------:|:-----------:|:-----------------------------:|:-----------:|
> | 0    | 64.75                          |           |                             |           |
> | 1    | 49.19                          | 15.56     | 39.55                      | 9.64      |
> | 2    | 31.56                          | 7.99      | 27.93                      | 3.63      |
> | 3    | 24.94                          | 3.00      | 23.18                      | 1.76      |
> | 4    | 21.53                          | 1.65      | 20.57                      | 0.95      |
> | 5    | 19.62                          | 0.95      | 19.10                      | 0.53      |
> | 6    | 18.54                          | 0.56      | 18.24                      | 0.30      |
> | 7    | 17.89                          | 0.34      | 17.71                      | 0.18      |
> | 8    | 17.49                          | 0.22      | 17.38                      | 0.11      |
> | 9    | 17.25                          | 0.13      | 17.18                      | 0.07      |
> | 10   | 17.10                          | 0.09      | 17.06                      | 0.04      |
> | 11   | 17.00                          | 0.05      | 16.98                      | 0.02      |
> | 12   | 16.95                          | 0.03      | 16.93                      | 0.01      |
> | 13   | 16.91                          | 0.02      | 16.91                      | 0.01      |
> | 14   | 16.89                          | 0.01      | 16.89                      | 0.01      |
> | 15   | 16.88                          | 0.01      | 16.88                      | 0.00      |
> | **Avg.** | -                              | **2.04**     | -                          | **1.15**      |
>
> As a result,  **updating $\boldsymbol{\alpha}\_{(k)}$ reduces more quantization errors than updating $\boldsymbol{b}\_{(k)}$, showing the importance of finding proper quantization levels.** Figure 4 (a) in our paper shows that FlexBCQ finds the proper quantization levels during optimization, resulting in high accuracy.

---

> ### Author Response · Authors · 2024-11-21
>
> > **[R1-2]** It might be worth considering ablations on the choice of calibration dataset, especially for Llama-3 models. Can you provide some justification as to how these datasets were chosen? Usually, for compression, subsets of the pre-training datasets are presumed to be good choices as calibration dataset. So was there any particular reason why C4 was chosen and not RedPajama (which is the pre-training dataset for OpenLlama)? Also, any ablations on the size of the pre-training dataset?
>
> For smaller language models such as ResNet or BERT, it is standard practice to perform compression using a subset (or entire) of either the pretrained dataset [2] or the fine-tuning dataset [3], as you noted, and to evaluate the performance of the compressed models on the same dataset. **However, for large language models (LLMs), which are designed to generalize well to unseen datasets, data not used during training can still be effectively leveraged for quantization.** Moreover, evaluations are typically conducted on datasets that were excluded from training, such as the MMLU benchmark or GSM8K.
>
> The table below summarizes the evaluation datasets used in major LLM quantization studies. As highlighted, **the C4 dataset is the most commonly utilized and representative dataset in previous research.** For FlexBCQ, we employed the C4 dataset to ensure consistency and fairness in comparisons with existing studies.
>
> **Table R1-2.** Sample datasets used in previous LLM quantization algorithms.
> | Method          |    GPTQ    |    AWQ     |    OmniQuant     |  FlexRound  |
> |:---------------:|:----------:|:----------:|:----------------:|:-----------:|
> | Sample Dataset  |   C4 [4]   |   Pile [5] | WikiText2 [6], C4|     C4      |
>
>
> To illustrate the effect of sample dataset size on the performance of a quantized model, we quantized the Llama-3 8B model to 3 bits using sample datasets of varying sizes and evaluated its performance on the MMLU benchmark. We use sample datasets ranging in size from 32 to 256, with 128 being the specific size used in our paper. Each sample dataset comprises 2,048 tokens. We summarize the result in Table R1-3.
>
> **Table R1-3.** MMLU 0-shot and 5-shot accuracies of 3-bit quantized models
> |  Benchmark |   32    |   64    |   128   |   256   |
> |:----:|:-------:|:-------:|:-------:|:-------:|
> | MMLU 5-shot     |  58.76  |  59.36  |  59.33  |  59.66  |
> | MMLU 0-shot     |  54.53  |  54.70  |  54.49  |  54.63  |
>
> **Experimental results indicate that the performance of the quantized model remains largely stable except in cases where the sample dataset size is extremely small (e.g., 32).** This is attributed to the fact that even the largest sample dataset (with 256 samples), consisting of $5.24 \times 10^5$  tokens, represents only approximately one ten-millionth of the $1.5 \times 10^{13}$ tokens used to train the Llama-3 8B model [7]. As such, marginal increases in dataset size do not result in significant performance changes. Based on these observations, the dataset size of 128, used in our study, is appropriate for comparing the accuracy of different quantization methods.
>
> We will explain the reason for selecting sample datasets in Section 4.1 (Experimental Setup). Additionally, we will include a discussion about the impact of sample dataset size on performance in Appendix.

---

> ### Author Response · Authors · 2024-11-21
>
> > **[R1-3]** Is it possible to provide a comparison of the average number of bits per parameter used by FlexBCQ, along with the other strategies it is being benchmarked against?
>
> In the rightmost column of Figure 3 in our paper, we summarize the values to store after quantization through UQ and BCQ formats, and FlexBCQ stores identical values to BCQ according to the Composition Theorem (Theorem 1 in our paper).
>
> When a weight group $w\in \mathbb{R}^{g}$ consisting of $g$ weights is quantized into $k$-bit using either the UQ or BCQ format, the total number of bits required for storage is calculated as shown below. In these calculations, the quantization parameters ($\boldsymbol{\alpha}_{(k)}$, $z\_{B,(k)}$, $\Delta\_{(k)}$, and $z\_{U,(k)}$) are set to 16 bits, consistent with the value used in our study.
>
> (# bits for BCQ) = $gk$ (for $\boldsymbol{b}\_{(k)}$) + $k$ * 16 (for $\boldsymbol{\alpha}\_{(k)}$) + 1 * 16 (for $z\_{B,(k)}$)  = $(gk + (k+1)16)$
>
> (# bits for UQ) = $gk$ (for $\widetilde{w}$) + 16 (for $\Delta\_{(k)}$) + 16 (for $z\_{U,(k)}$) = $(gk + 2 * 16)$
>
> In this context, $\boldsymbol{b}\_{(k)}\in\\{-1,1\\}^{g\times k}$, $\boldsymbol{\alpha}_{(k)}\in\mathbb{R}^{k}$, $z\_{B,(k)}\in\mathbb{R}$ denote the binary codes for weights, BCQ’s scale factor, and BCQ’s shifting factor, respectively.
> $\widetilde{w}\in ([0,2^{k-1}]\cap\mathbb{Z})^{g}$, $\Delta\_{(k)}\in\mathbb{R}$, and $z\_{U,(k)}\in\mathbb{R}$ represent the weights mapped to integers, UQ’s scale factor, and UQ’s zero-point, respectively.
> Therefore, we have the average number of bits per weight for each quantization format as follows.
>
> (average # bits per weight for BCQ) = $(gk + (k+1)16) / g$ = $k + (k+1)16/g$
>
> (average # bits per weight for UQ) = $(gk + 2 * 16) / g$ = $k + 32/g$
>
> Based on the results above, the average number of bits per weight for UQ and BCQ at the 3-bit and 4-bit levels is summarized in the table below.
>
> **Table R2-1.** The average number of bits per weight after quantization by BCQ and UQ when the bit width of weights is 3 or 4 bit.
> | Scheme       |        4 bit         |  3 bit  |
> |:------:|:--------------------:|:-------:|
> |   UQ   |        4.25          |  3.25   |
> |   BCQ  |        4.63          |  3.50   |
>
> BCQ requires approximately 8% more bits compared to UQ. However, as derived above, the average number of bits per weight for BCQ is given by $k + (k+1)/g​$. **This indicates that the additional bit usage diminishes significantly as the group size $g$ increases.** For example, if a single column, with 4096 weights, of the weight matrix in the Llama-3 8B model is treated as one group, the average number of bits per weight for 4-bit quantization is approximately 4.02. In this scenario, the bit consumption for the quantization parameters becomes negligible.
>
> Moreover, in the context of recent studies on LLM compression [8], acceleration capabilities are prioritized over memory consumption. **Notably, when utilizing LUT-GEMM kernels, BCQ achieves the same computational speed as UQ despite its higher average bit count.** Therefore, we directly compare the accuracy of the quantized models in BCQ and UQ formats denoted in Tables 1 and 2 without losing fairness.
>
> Discussion about average bits per weight of UQ and BCQ will be included in the Appendix during the discussion period.

---

> ### Author Response · Authors · 2024-11-21
>
> > **[R1-4]** Is it possible to use the kernels from LUT-GEMM to see an actual improvement in inference throughput?
>
> The official implementation of LUT-GEMM [9] kernel is open-sourced in GitHub [10].
> We reorganize some useful results in LUT-GEMM as follows to show the practical effectiveness of BCQ-quantized models.
>
> **Table R1-4.**  Latency comparison of the first FFN layer on OPT-175B model with various precision and corresponding kernel selections with m = 12288 and g = 128 on A100-80GB-GPU (Table 1 in LUT-GEMM).
>
> | Kernel    | Schemes | Weight  |     Input        |     Output    | Latency (ms) | Speedup ($\times$) |
> |:---------:|:------------------:|:------------------------:|:----------:|:--------:|:-------------:|:------------------:|
> | cuBLAS    |         -          |         FP16            |    FP16    |   FP16   |     0.7256    |        1.0         |
> | GPTQ      |         UQ         |         INT3            |    FP16    |   FP16   |     0.3599    |        2.0         |
> | AWQ       |         UQ         |         INT4            |    FP16    |   FP16   |     0.3238    |        2.2         |
> | LUT-GEMM  |      UQ, BCQ       |         INT4*           |    FP16    |   FP16   |     0.2688    |        2.7         |
> | LUT-GEMM  |      UQ, BCQ       |         INT3*           |    FP16    |   FP16   |     0.2250    |        3.2         |
>
>
>
> \* LUT-GEMM takes the same time for computing UQ and BCQ format values if they have the same bit-width
>
>
> **Table R1-5.** End-to-end latency per token for Llama family models (Table 7 in LUT-GEMM).
>
> |     Model     |   Kernel   | bit-width (k) | Group size (g) | Latency (ms) |
> |:-------------:|:----------:|:-------------:|:--------------:|:------------:|
> |   Llama-7B    |   cuBLAS   |      16       |       -        |      10      |
> |               | LUT-GEMM   |       4       |      128       |      6.1     |
> |               | LUT-GEMM   |       3       |      128       |      5.5     |
> |   Llama-13B   |   cuBLAS   |      16       |       -        |     18.2     |
> |               | LUT-GEMM   |       4       |      128       |     10.4     |
> |               | LUT-GEMM   |       3       |      128       |      9.3     |
> |   Llama-30B   |   cuBLAS   |      16       |       -        |     43.6     |
> |               | LUT-GEMM   |       4       |      128       |     20.7     |
> |               | LUT-GEMM   |       3       |      128       |     18.1     |
> |   Llama-65B   |   cuBLAS   |      16       |       -        |      OOM     |
> |               | LUT-GEMM   |       4       |      128       |     35.7     |
> |               | LUT-GEMM   |       3       |      128       |     31.3     |
>
> Table R1-4 exhibits that LUT-GEMM outperforms conventional kernels that support only UQ kernels and Table R1-5 demonstrates that LUT-GEMM provides end-to-end inference acceleration for diverse sizes of Llama models. **Therefore, BCQ format, which is supported by LUT-GEMM, is practically valuable.**
>
> We will include the discussion about the inference speed of LUT-GEMM in Appendix for completeness.

---

> ### Author Response · Authors · 2024-11-21
>
> > **[R1-5]** I initially had about why despite its greater expressive power, BCQ significantly underperforms compared to UQ, which has limited expressive capabilities for LLM compression (as mentioned in lines 67-68)? This was perhaps answered in 199-200 -- Is this a hypothesis and are there concrete datapoints to support this? I might have missed this in the reading of the paper.
>
> Historically, the absence of efficient kernels for BCQ has limited studies regarding accurate BCQ algorithms for LLMs. Consequently, recent studies such as GPTQ, AWQ, and OmniQuant have focused on UQ, leveraging publicly available acceleration kernels. To the best of our knowledge, there have been no BCQ-based quantization techniques specifically developed for LLMs.
>
> The existing BCQ algorithm applicable to billion-scale LLMs is Xu et al. (2018). However, this algorithm has notable limitations
>
> 1. **Neglecting input distribution.** The quantization process does not account for the input distribution, potentially leading to suboptimal performance.
> 2. **Independently updating weight groups.** Each weight group is updated independently, disregarding interdependencies among weights, which can adversely affect model accuracy.
>
> These shortcomings are evident in Equations (2) through (5) of Xu et al.'s paper, where the absence of input considerations and the isolated treatment of weight groups are apparent.
>
> In this context, the development of LUT-GEMM, a kernel capable of accelerating BCQ to match the speed of UQ while surpassing existing UQ acceleration kernels, marks a significant advancement; LUT-GEMM, which is published in ICLR’24, enables efficient BCQ inference for LLMs. We propose FlexBCQ, the first BCQ quantization algorithm tailored for LLMs, to leverage the effectiveness of LUT-GEMM. FlexBCQ offers several key improvements:
>
> 1. **Considering input distribution.** By accounting for input distribution during the quantization process, FlexBCQ effectively reduces reconstruction error.
> 2. **Reflecting dependencies between weight groups.** The algorithm simultaneously updates the quantization parameters in different weight groups, considering the dependencies between them.
> 3. **Integration of UQ Training Techniques.** FlexBCQ incorporates useful training methodologies traditionally used for UQ, e.g., flexible mapping, further improving its effectiveness.
>
> As a result, FlexBCQ demonstrates superior performance, outperforming the approach by Xu et al. by up to 67%, as evidenced in Tables 1 and 2 of our paper.
>
> We will revise lines 199-200 to clarify the limitations of Xu et al as follows.
> * **Before update:** The only low-cost algorithm that is applicable to LLM (Xu et al. 2018) with BCQ scheme independently updates alpha and b, while ignoring the dependencies between them.
> * **After update:** The only low-cost algorithm that is applicable to LLM (Xu et al. 2018) with BCQ scheme does not take input distribution into account and ignores dependencies of different weight groups, resulting in low accuracy.
>
> ----
>
> > **[R1-6]** There is a potential typo in the definition of $\Delta_{(k)}$. Perhaps it should be $(w_{M,c}-w_{m,c})/2^{k-1}$?
>
> We appreciate your detailed review! We fix the typo and the fixed version will be available soon.
>
> ----
>
> > **[R1-7]** In Line 195 - 196, please point to a more precise Theorem/Lemma in Park et. al. for reference.
>
> "Appendix C: Conversion of Uniform Quantization Into BCQ” in LUT-GEMM [9] contains the conversion rules for transforming UQ into BCQ. This clarification will be reflected in the revised version, which will be available for review shortly.

---

> ### Author Response · Authors · 2024-11-21
>
> > **[R1-8]** How is $\boldsymbol{\alpha}\_{(k)}$ chosen? How is the blockwise output reconstruction done?
>
> We update the quantization parameters of FlexBCQ through the blockwise output reconstruction process where the block indicates a composition of an MHA and MLP modules; Llama-3 8B model has 32 blocks. The quantization parameters of weight groups in the same block are updated simultaneously using stochastic gradient descent (SGD) with our proposed two optimization techniques (Gradient Filtering and Periodic Remapping).
>
> The details of our block reconstruction process are as follows:
>
> * **Objective function:** $||B\_i(X\_i)  - \widehat{B}\_i(\widehat{X}\_i)||\_F^2$
> * **Optimization target:** quantization parameters of FlexBCQ
>
>     * (UQ’s scale factor $\Delta_{(k)}$,
>       UQ’s zero-point $z_{U,(k)}$,
>        scale factors $s$ and $s_r$ for flexible mapping, and
>        BCQ’s scale factor $\boldsymbol{\alpha}\_{(k)}$)
> * **Optimization algorithm:** stochastic gradient descent (SGD)
> * **Additional optimization techniques:** Gradient Filtering and Periodic Remapping
> * **Optimization strategy:** sequential blockwise optimization from the bottom block to the top block
>
> $B\_i$ and $\widehat{B}\_i$ are the functions for the $i$-th unquantized and quantized blocks, respectively.
> $\boldsymbol{X}\_i$ and $\widehat{\boldsymbol{X}}\_i$ are the inputs for the $i$-th unquantized and quantized blocks, respectively.
> We will elaborate more details about Blockwise output reconstruction in Section 3.4.
>
> ----
>
> > **[R1-9]** The gradient filtering section can benefit from a little more explanation. Does it mean that when the mapping error exceeds epsilon, the gradients are zeroed out?
>
> You are correct. When the difference between the transformed weight $\bar{w}\_{(k)}$ and **the mapped weight $\widetilde{w}\_{(k)}$ exceeds the gradient filtering threshold $\tau$, i.e. $|\bar{w}\_{(k)}-\widetilde{w}\_{(k)}| > \tau$, the gradient with respect to w is set to zero.** Importantly, this process does not affect forward propagation.
>
> We have explicitly clarified in lines 333-334 that the filtered gradient becomes zero. This clarification will be reflected in the revised version, which will be available for review shortly.
>
> ----
>
> > **[R1-10]** Please specify all the learnable parameters in FlexBCQ explicitly in a sentence somewhere.
>
> **FlexBCQ contains both UQ’s and BCQ’s quantization parameters** to leverage both UQ’s training technique (flexible mapping) and adaptive quantization levels.
> Therefore, FlexBCQ has UQ’s scale factor $\Delta_{(k)}$, UQ’s zero-point $z_{U,(k)}$, scale factors $s$ and $s_r$ for flexible mapping, BCQ’s scale factor $\boldsymbol{\alpha}\_{(k)}$, and BCQ’s shifting factor $z_{B,(k)}$. **We update all quantization parameters of FlexBCQ except for $z_B$**. See response [R1-8] for details about updating the quantization parameters of FlexBCQ. We will specify the learnable parameters of FlexBCQ at the beginning of Section 3.4 (Blockwise Output Reconstruction) before explaining the optimization process.
>
>
> ----
> > **[R1-11]** The LLM quantization problem formalization in the beginning of Sec. 2.1 needs more description.
>
> We modify Section 2.1 (LLM Quantization Problem) to provide more details about our problem. We specify that we are focusing on improving the accuracy of the quantized models in UQ and BCQ format since there is an efficient kernel (LUT-GEMM) that runs UQ and BCQ at the same speed. The following are the problem definitions before and after revision.
>
> * **[Before]**
> We have an accurate LLM $f$, a desired bit-width $k$, and a sample dataset $\mathcal{D}$ of input token sequences. Our goal is to find an accurate $k$-bit quantized model $\hat{f}$
>
> * **[After]**
> We have an accurate LLM $f$, a desired bit-width $k$, and a sample dataset $\mathcal{D}$ of input token sequences. Our goal is to find an accurate $k$-bit quantized model $\hat{f}$.
> **In this paper, we focus on UQ and BCQ schemes. We directly compare the accuracies of UQ and BCQ algorithms since there is an effective kernel, LUT-GEMM, which runs UQ and BCQ at the same inference speed when they have the same bit-width.**
>
> ----
> These are our responses to your review.
> We hope our responses resolve your concerns.
> If you have any further inquiries or require additional clarifications, please feel free to ask us.
> Thanks,

---

> ### Author Response · Authors · 2024-11-21
>
> **[Reference]**
>
> [1] Xu, Chen, et al. "Alternating multi-bit quantization for recurrent neural networks." arXiv preprint arXiv:1802.00150 (2018).
>
> [2] Han, Song, Huizi Mao, and William J. Dally. "Deep compression: Compressing deep neural networks with pruning, trained quantization and huffman coding." arXiv preprint arXiv:1510.00149 (2015).
>
> [3] Park, Seungcheol, Hojun Choi, and U. Kang. "Accurate Retraining-free Pruning for Pretrained Encoder-based Language Models." The Twelfth International Conference on Learning Representations. 2024.
>
> [4] Raffel, Colin, et al. "Exploring the limits of transfer learning with a unified text-to-text transformer." Journal of machine learning research 21.140 (2020): 1-67.
>
> [5] Gao, Leo, et al. "The pile: An 800gb dataset of diverse text for language modeling." arXiv preprint arXiv:2101.00027 (2020).
>
> [6] Merity, Stephen, et al. "Pointer sentinel mixture models." arXiv preprint arXiv:1609.07843 (2016).
>
> [7] https://ai.meta.com/blog/meta-llama-3/
>
> [8] Song, Jiwon, et al. "SLEB: Streamlining LLMs through Redundancy Verification and Elimination of Transformer Blocks." arXiv preprint arXiv:2402.09025 (2024).
>
> [9] Park, Gunho, et al. "Lut-gemm: Quantized matrix multiplication based on luts for efficient inference in large-scale generative language models." arXiv preprint arXiv:2206.09557 (2022).
>
> [10] https://github.com/naver-aics/lut-gemm

---

> > ### Comment · Reviewer_y6hi · 2024-11-26
> > **Response to authors rebuttal**
> >
> > Dear authors,
> >
> > Thank you for your response. I believe most of my concerns have been addressed. Appreciate the additional experiments!
> >
> > However, I'm finding it a little difficult to agree with your claims regarding the choice of calibration set (ref: https://openreview.net/forum?id=dZ3cI69BE8&noteId=XuMaVwBCSk). Can you please cite a source to justify your choice of C4? You claim that *"However, for large language models (LLMs), which are designed to generalize well to unseen datasets, data not used during training can still be effectively leveraged for quantization"* -- emphasis here is on **can**. I'm not saying that C4 is not a reasonable choice -- my suggestion is merely to point out that the choice of calibration dataset does indeed matter, even for LLMs, and it is likely that RedPajama or FineWeb might give better results in terms of accuracy. For instance, Llama 3 family of models is trained on a significant fraction of multi-lingual data (see: https://arxiv.org/abs/2407.21783) -- and calibration dataset does indeed require some amount of multi-lingual data to minimize the accuracy degradation. In fact, recent knowledge distillation methods (eg: https://arxiv.org/abs/2305.17888) propose using synthetic data generated from the uncompressed model for compression;, with the major motivation being the fact that pre-training datasets are often not public. It will be appreciated if this is pointed out as a limitation/future work for this paper.
> >
> > Additionally, as also pointed out by other reviewers, the writing can be vastly improved -- so please take care of that. Contingent on the fact that the authors take into account all the suggestions in the review period, I'd be inclined to recommend acceptance of the paper.

---

> > > ### Author Response · Authors · 2024-11-28
> > >
> > > Dear Reviewer y6hi,
> > >
> > > We are pleased to hear that most of the your concerns have been addressed through our responses. However, it seems that concerns regarding the paper’s writing and the calibration set remain, thus we provide additional clarification on these points.
> > >
> > > ----
> > >
> > > **[Writing]**
> > >
> > > First, we made every effort to improve the writing of the paper. The overall improvements are summarized in the Global Response [G2], and we have addressed the your feedback as outlined below.
> > >
> > > * **[R1-2] Appendix C.4:**  Include experimental results regarding diverse calibration set and dataset sizes
> > > * **[R1-3] Appendix F.1:**  Include discussion of memory usage of FlexBCQ
> > > * **[R1-5] Lines 184-186:** Clarified the limitations of Xu et al. more explicitly.
> > > * **[R1-6] Line 137:** Revise the typo regarding $\Delta_{(k)}$
> > > * **[R1-7] Line 183:** Specify the citation for the statement
> > > * **[R1-8] Lines 319-331:** We add more detailed explanation about Blockwise Output Reconstruction process
> > > * **[R1-9] Line 339-340:** Clearly state that the filtered gradient are zeroed out
> > > * **[R1-10] Lines 330-331:** Specify the learnable parameters in FlexBCQ
> > > * **[R1-11] Lines 101-104:** We clearly state that we directly compare the accuracy of the quantized models since UQ and BCQ algorithms have the same inference speed with LUT-GEMM when they have the same bit width.
> > >
> > > ----
> > >
> > > **[Regarding Calibration Set]**
> > >
> > > We use C4 dataset as the calibration set since it is also used in previous works such as GPTQ [1], OmniQuant [2], and FlexRound [3] (see Table R1-2). We use it for a fair comparison with these methods. Therefore, the justification for this choice comes from the papers proposing these techniques, which are listed in [References]. However, as you pointed out, the calibration set plays a crucial role in informing the model about the input distribution, and selecting a good calibration set is quite important.
> > >
> > > We agree with your point, and to accurately analyze the impact of the calibration set, we conduct experiments using the SlimPajama [4] dataset and the FineWeb-Edu [5] dataset to observe the performance changes when quantization is applied. The SlimPajama dataset is a cleaned and deduplicated version of RedPjama [6], and FineWeb-Edu is a dataset created by filtering educational data from FineWeb [7]. We use FineWeb-Edu since it is related to our benchmarks, MMLU and GSM8K, regarding education. We use the Llama-3 8B model for experiments, and the results are presented in Table R1-6.
> > >
> > > **[Table R1-6]** Accuracies of 3-bit quantized Llama-3 8B models on MMLU benchmarks using FlexBCQ and competitor. We report average accuracies of MMLU 5-shot and 0-shot benchmarks. Diff. column indicates the amount of improvements in average accuracies over the results on C4.
> > > | Scheme |   Method  |   C4  | SlimPajama | Diff. | FineWeb-Edu | Diff. |
> > > |:------:|:---------:|:-----:|:----------:|:-----:|:-----------:|:-----:|
> > > |   UQ   |    AWQ    | 52.38 |    52.61   |  0.23 |    51.31    | -1.07 |
> > > |   UQ   | OmniQuant | 49.10 |    48.57   | -0.53 |    49.055   | -0.04 |
> > > |   UQ   | FlexRound | 56.77 |    56.00   | -0.77 |    56.53    | -0.24 |
> > > |   BCQ  |  FlexBCQ  | 56.91 |    57.85   |  0.94 |    57.41    |  0.50 |
> > >
> > > The experimental results show that FlexBCQ achieves performance improvements of 0.94% and 0.50% in average accuracy on the SlimPajama and FineWeb datasets, respectively. However, other techniques exhibit accuracy drops of up to 1.07%. From these results, we conclude that while the importance of the dataset itself is evident, each algorithm has a dataset that is more suitable for it. FlexBCQ, however, is less affected by this and provides good results across diverse datasets.
> > >
> > > These are our responses to your second feedback and if you have further concerns, let us know.
> > >
> > > **[References]**
> > >
> > > [1] Frantar, Elias, et al. "Gptq: Accurate post-training quantization for generative pre-trained transformers." arXiv preprint arXiv:2210.17323 (2022).
> > >
> > > [2] Shao, Wenqi, et al. "Omniquant: Omnidirectionally calibrated quantization for large language models." arXiv preprint arXiv:2308.13137 (2023).
> > >
> > > [3] Lee, Jung Hyun, et al. "Flexround: Learnable rounding based on element-wise division for post-training quantization." International Conference on Machine Learning. PMLR, 2023.
> > >
> > > [4] https://cerebras.ai/blog/slimpajama-a-627b-token-cleaned-and-deduplicated-version-of-redpajama
> > >
> > > [5] https://huggingface.co/datasets/HuggingFaceFW/fineweb-edu
> > >
> > > [6] https://github.com/togethercomputer/RedPajama-Data
> > >
> > > [7] https://huggingface.co/datasets/HuggingFaceFW/fineweb

---

### Author Response · Authors · 2024-11-21

**[G2] Improving Readability of the Paper**

Many reviewers provided feedback on the readability of the paper. In response, we made extensive revisions during the discussion period to improve clarity and flow. Below are the key changes, highlighted in blue in the revised paper.

* **Figure 1.** We added the baseline Round-to-Nearest (RTN) method and revised the caption accordingly to effectively convey the main idea of the paper.
* **Figure 2.** We revised the figure to improve clarity and add a more detailed caption.
* **Figure 3.** We added a “Deployment” column and simplified the figure.
* **Figure 4.** We adjusted the color of Figure 4 to improve readability.
* **Section 2.1**
    * (Lines 101-104) We clearly state that we directly compare the accuracy of the quantized models since UQ and BCQ algorithms have the same inference speed with LUT-GEMM when they have the same bit width.
* **Section 2.2**
    * (Equation (1) and (6)) We introduce the notation $\Theta$ as a set of quantization parameters to simplify the equations.
    * (Lines 147-148, 192-194) We add descriptions of the inference process of each quantization scheme to clearly deliver our main idea.
* **Section 3.2**
    * (Equation (7)) We eliminated the repetitive formulas in Equation (7) and streamlined it.
    * (Lines 258- 292) We clarified the effect of the Composition Theorem.
* **Section 3.4**
    * (Lines 319-331) We add a more detailed explanation of Blockwise Output Reconstruction process
* **Appendix**
    * We included all topics discussed, including terminology, symbols, implementation details, the effect of Composition Theorem, additional analysis of FlexBCQ, and additional experiments.
* All typos found in the discussion period were corrected.

---

> ### Author Response · Authors · 2024-11-28
>
> **[G3] Exhaustive Verification of FlexBCQ**
>
> Many reviewers suggested additional experiments.
> We conduct all suggested experiments and summarize some key results below.
> Remarkably, FlexBCQ outperforms in every case. We appreciate all the reviewers for their excellent experimental suggestions, and we trust these results further demonstrate FlexBCQ's effectiveness. All experiments were conducted using Llama-3 8B.
>
> **Table G-1.** Perplexities of quantized models.
> |  UQ |       UQ       | 4bit, Wiki ($\downarrow$) | 4bit, C4 ($\downarrow$) | 3bit, Wiki ($\downarrow$) | 3bit, C4 ($\downarrow$) |
> |:---:|:--------------:|:----------:|:--------:|:----------:|:--------:|
> |  -  | Full precision |    6.14    |   8.89   |    6.14    |   8.89   |
> |  UQ |       RTN      |    6.75    |   9.67   |    10.82   |   14.85  |
> |  UQ |       AWQ      |    6.54    |   9.40   |    8.23    |   11.59  |
> |  UQ |      GPTQ      |    6.54    |   9.36   |    9.05    |   11.70  |
> |  UQ |    OmniQuant   |    6.69    |   9.59   |    8.82    |   12.36  |
> |  UQ |    FlexRound   |    6.55    |   9.36   |    7.62    |   10.76  |
> | BCQ |     Greedy     |  6.22E+04  | 2.45E+04 |  8.32E+04  | 3.81E+04 |
> | BCQ |   Alternating  |    7.70    |   10.88  |   869.89   |  978.83  |
> | BCQ |     FlexBCQ    |    **6.46**    |   **9.24**   |    **7.42**    |   **10.46**  |
>
> **Table G-2.** 2-bit quantization results.
> | Scheme |     Method     | 5-shot ($\uparrow$) | 0-shot ($\uparrow$) | GSM8K ($\uparrow$) |   Wiki ($\downarrow$)  |    C4 ($\downarrow$)    |
> |:------:|:--------------:|:------:|:------:|:-----:|:--------:|:--------:|
> |    -   | Full precision |  65.02 |  60.52 | 76.12 |   6.14   |   8.89   |
> |   UQ   |       RTN      |  22.95 |  22.95 |  0.00 | 1.97E+06 | 2.44E+06 |
> |   UQ   |       AWQ      |  22.95 |  22.95 |  0.00 | 1.72E+06 | 2.13E+06 |
> |   UQ   |      GPTQ      |  25.32 |  24.43 |  0.40 |  450.59  |  254.37  |
> |   UQ   |    OmniQuant   |  25.04 |  22.91 |  0.00 |  987.10  |  1395.17 |
> |   UQ   |    FlexRound   |  24.27 |  24.97 |  0.23 |   68.54  |   66.57  |
> |   BCQ  |     Greedy     |  26.76 |  26.89 |  0.00 | 4.96E+05 | 6.85E+05 |
> |   BCQ  |   Alternating  |  23.59 |  22.97 |  0.00 | 7.34E+06 | 3.97E+06 |
> |   BCQ  |     FlexBCQ    |  **37.02** |  **34.37** | **22.54** |   **14.95**  |   **16.55**  |
>
>
> **Table G-3.** Channel-wise quantization results.
> | Scheme |    Method   | 4bit, 5-shot ($\uparrow$) | 4bit, 0-shot ($\uparrow$) | Average ($\uparrow$) | 3bit, 5-shot ($\uparrow$) | 3bit, 0-shot ($\uparrow$) | Average ($\uparrow$) |
> |:------:|:-----------:|:------------:|:------------:|:-------:|:------------:|:------------:|:-------:|
> |   UQ   |     RTN     |     57.02    |     52.31    |  54.67  |     25.39    |     23.15    |  24.27  |
> |   UQ   |     AWQ     |     61.67    |     55.61    |  58.64  |     43.86    |     37.96    |  40.91  |
> |   UQ   |     GPTQ    |     41.62    |     41.94    |  41.78  |     25.92    |     25.98    |  25.95  |
> |   UQ   |  OmniQuant  |     58.11    |     53.86    |  55.99  |     25.35    |     28.02    |  26.69  |
> |   UQ   |  FlexRound  |     60.84    |     53.21    |  57.02  |     54.52    |     45.92    |  50.22  |
> |   BCQ  |    Greedy   |     24.19    |     23.00    |  23.59  |     24.65    |     26.44    |  25.55  |
> |   BCQ  | Alternating |     22.95    |     22.95    |  22.95  |     25.51    |     25.51    |  25.51  |
> |   BCQ  |   FlexBCQ   |     **62.26**    |     **57.01**    |  **59.63**  |     **55.73**    |     **51.01**    |  **53.37**  |
>
> * 5-shot and 0-shot represent MMLU 5-shot and MMLU 0-shot benchmarks, respectively.
> * Higher accuracies on MMLU and GSM8K benchmarks represents better performances
> * Lower perplexities on WikiText-2 (Wiki) and C4 benchmarks represents better performance

---

> ### Author Response · Authors · 2024-11-28
>
> Dear Reviewers,
>
> We have updated the revised version of the paper to reflect all of your comments. We greatly appreciate your constructive feedback, and we believe the quality of the paper has been significantly enhanced, particularly in terms of readability, verification, and analysis. We sincerely thank you for your efforts and kindly ask that you review the updated version to ensure all of your suggestions have been adequately addressed. Should you have any additional points or concerns, we would be happy to address them.
>
> Sincerely,
>
> The Authors

---

> ### Author Response · Authors · 2024-12-02
>
> Dear reviewers,
>
> We have carefully incorporated your valuable feedback and summarized the updates in individual comments.
> The revisions made to the paper based on your comments are summarized in the Global Response [G2]. In [G3], we have presented compelling additional experimental results that highlight the superiority of FlexBCQ. We kindly ask you to review these updates before the discussion period concludes.
>
> Sincerely,
>
> The Authors

---

### Author Response · Authors · 2024-11-28
**Global Response**

Dear all reviewers,

On behalf of all authors, we would like to express our heartfelt gratitude to all reviewers for their thoughtful and constructive reviews. Receiving such detailed and high-quality feedback is a rare and valuable opportunity, and we deeply appreciate the time and effort you have dedicated to reviewing our paper. We think that the quality of our paper is quite improved by rigorously revalidating our work and enhancing readability while answering your reviews.

Before addressing individual reviews, we would first like to summarize and respond to frequently mentioned comments, categorized as follows:

**[G1] Clarifying the Contributions of FlexBCQ**

**[G2] Improving Readability of the Paper (updated)**

**[G3] Exhaustive Verification of FlexBCQ (updated)**

All revisions mentioned in the reviews are incorporated into the updated PDF, highlighted in blue.

---

**[G1] Clarifying the Contributions of FlexBCQ**

We are pleased to see that all reviewers seem to agree that FlexBCQ is novel and non-trivial. However, it appears that some aspects of the paper’s contributions may not have been communicated clearly. To address this, we provide a concise summary of the Background, Goal, and Contributions of FlexBCQ.

**[Background]**

The key motivations behind FlexBCQ are as follows:

* **BCQ’s Theoretical Potential.** Binary-coding quantization (BCQ) theoretically offers higher representational power than uniform quantization (UQ) [1]. However, due to the lack of effective optimization techniques, BCQ has so far demonstrated even lower performance than UQ, which itself has limited representational power.
* **Emerging Efficient BCQ Kernels.** Despite BCQ’s theoretical advantages, recent advancements have introduced efficient BCQ kernels (e.g., LUT-GEMM [1]) that replace repetitive computations with table lookups. These kernels guarantee faster inference speeds than traditional UQ kernels (e.g., AWQ [2], GPTQ [3]) in certain scenarios and achieve the same inference speed as UQ when using the same bit-width.

**[Goal]**

The primary objective of FlexBCQ is to harness BCQ’s higher representational capacity to achieve superior accuracy compared to both existing BCQ and UQ-based quantization algorithms. **We assume that BCQ and UQ have identical inference speeds when using the same bit-width [1] and mainly compare the accuracies of the quantized models.**

**[Contributions]**

The contributions of FlexBCQ are summarized as follows:

* **Novel Optimization Algorithm for BCQ.** We propose a novel method to integrate UQ’s useful optimization techniques for BCQ, along with a new initialization strategy and optimization techniques tailored for BCQ. (Section 3)

* **Experimental Validation.** Through extensive experiments encompassing both general and task-specific benchmarks, we demonstrate that FlexBCQ achieves consistently superior performance compared to existing UQ and BCQ algorithms. (Tables 1–2)

* **Effective Utilization of BCQ and UQ Features.** We show that models quantized using FlexBCQ effectively leverage UQ’s flexible mapping techniques and BCQ’s adaptive quantization levels, as demonstrated in Figure 4.

* **First Application of UQ Techniques to BCQ.** This is the first work to adapt UQ optimization algorithms for BCQ. Our methodology has the potential to extend other UQ-based approaches, such as OmniQuant, GPTQ, and AWQ, to BCQ, leading to performance improvements. For example, Table 4 highlights a Unified Initialization method, which applies UQ’s grid search (RTN) to BCQ and achieves notable performance gains. This is one of our promising future works.

We hope this summary clarifies FlexBCQ’s contributions and sets the foundation for further fruitful discussions. Thank you once again for your valuable feedback.

**[Reference]**

[1] Park, Gunho, et al. "Lut-gemm: Quantized matrix multiplication based on luts for efficient inference in large-scale generative language models." arXiv preprint arXiv:2206.09557 (2022)

[2] Lin, Ji, et al. "AWQ: Activation-aware Weight Quantization for On-Device LLM Compression and Acceleration." Proceedings of Machine Learning and Systems 6 (2024): 87-100.

[3] Frantar, Elias, et al. "Gptq: Accurate post-training quantization for generative pre-trained transformers." arXiv preprint arXiv:2210.17323 (2022).

---

### Meta-Review · Area_Chair_DUcs · 2024-12-20

**Metareview:**

Dear Authors,

Thank you for your valuable contribution to ICLR and the ML community. Your submitted paper has undergone a rigorous review process, and I have carefully read and considered the feedback provided by the reviewers.

This work proposes an approach to improve the performance of quantization by integrating FlexRound and Binary-Coding Quantization. The approach is evaluated on several language models.

The paper received borderline final review scores (6,5,5,3). Reviewers pointed out critical issues including (i) readability and clarity issues (ii) issues in the presentation of theoretical results (iii) limited novelty of the proposed method. Thank you for providing a detailed rebuttal and updating the paper. The updated paper clarified certain points and provide additional evaluations. However, the rebuttal was not convincing enough for three reviewers to increase their scores beyond borderline-reject/reject.

Given the current form of the paper and the reviewer discussion, I regret to inform you that I am unable to recommend the acceptance of the paper for publication at ICLR. I want to emphasize that this decision should not be viewed as a discouragement. In fact, the reviewers and I believe that your work has valuable insights and, with further development and refinement, can make a meaningful impact on the field.

I encourage you to carefully address the feedback provided by the reviewers and consider resubmitting the paper. Please use the comments and suggestions in the reviews to improve and refine your work.

Best,
AC

**Additional Comments On Reviewer Discussion:**

Reviewers pointed out a series of issues including (i) readability and clarity issues (ii) issues in the presentation of theoretical results (iii) limited novelty of the proposed method. The authors provided a detailed rebuttal and updated the paper, which clarified certain points and provided additional evaluations. However, the rebuttal was not convincing enough for three reviewers to increase their scores beyond borderline-reject/reject.

---

### Decision · Program_Chairs · 2025-01-22

Reject